# Non-Asymptotic Analysis of a UCB-based
# Top Two Algorithm

**Marc Jourdan**
`marc.jourdan@inria.fr`

**Rémy Degenne**
`remy.degenne@inria.fr`

[1] Univ. Lille, CNRS, Inria, Centrale Lille, UMR 9189-CRIStAL, F-59000 Lille, France

## Abstract

A Top Two sampling rule for bandit identification is a method which selects the next arm to sample from among two candidate arms, a *leader* and a *challenger*. Due to their simplicity and good empirical performance, they have received increased attention in recent years. However, for fixed-confidence best arm identification, theoretical guarantees for Top Two methods have only been obtained in the asymptotic regime, when the error level vanishes. In this paper, we derive the first non-asymptotic upper bound on the expected sample complexity of a Top Two algorithm, which holds for any error level. Our analysis highlights sufficient properties for a regret minimization algorithm to be used as leader. These properties are satisfied by the UCB algorithm, and our proposed UCB-based Top Two algorithm simultaneously enjoys non-asymptotic guarantees and competitive empirical performance.

## 1   Introduction

Faced with a collection of items ("arms") with unknown probability distributions, a question that arises in many applications is to find the distribution with the largest mean, which is referred to as the best arm. Different approaches have been considered depending on the data collection process. Sequential hypothesis testing [10, 37] encompasses situations where there is no control on the collected samples. Experimental design [7, 35] aims at choosing the data collection scheme a priori. In the multi-armed bandit [4, 20] and the ranking and selection [18] literature, an algorithm chooses sequentially the distribution from which it will collect an additional sample based on past data.

In order to have theoretical guarantees for this identification problem, one should adopt a statistical model on the underlying distributions. While parametric models are reasonable for applications such as A/B testing [28], they are unrealistic in other fields such as agriculture [21]. Despite the restricted scope of its applications, studying the identification task for Gaussian distributions is a natural first step. Hopefully the insights gained will then be generalized to wider classes of distributions.

In the fixed confidence identification problem, an algorithm aims at identifying the best arm with an error of at most $\delta \in (0, 1)$ while using as few samples as possible. Since each sample has a cost, those algorithms should provide an upper bound on the expected number of samples used by the algorithm before stopping. For those guarantees to be useful in practice, they should hold for any $\delta$, which is referred to as the non-asymptotic (or moderate) regime. In contrast, the asymptotic regime considers vanishing error level, i.e. $\delta \to 0$. For Gaussian distributions, Top Two algorithms [36, 39]

37th Conference on Neural Information Processing Systems (NeurIPS 2023).

have only been studied in the asymptotic regime. We show the first non-asymptotic guaranty for a Top Two algorithm holding for any instance with a unique best arm.

## 1.1 Setting and related work

A Gaussian bandit problem is described by $K$ arms whose probability distributions belongs to the set $\mathcal{D}$ of Gaussian distributions with known variance $\sigma^2$. By rescaling, we assume $\sigma_i^2 = 1$ for all $i \in [K]$. Since an element of $\mathcal{D}$ is uniquely characterized by its mean, the vector $\mu \in \mathbb{R}^K$ refers to a $K$-arms Gaussian bandit. Let $\triangle_K \subseteq \mathbb{R}^K$ be the $(K-1)$-dimensional simplex.

A best arm identification (BAI) algorithm aims at identifying an arm with highest mean parameter, i.e. an arm belonging to the set $i^\star(\mu) = \arg\max_{i \in [K]} \mu_i$. At each time $n \in \mathbb{N}$, the algorithm (1) chooses an arm $I_n$ based on previous observations, (2) observes a sample $X_{n,I_n} \sim \mathcal{N}(\mu_{I_n}, 1)$, and (3) decides whether it should stop and return an arm $\hat{i}_n$ or continue sampling. We consider the *fixed confidence* identification setting, in which the probability of error of an algorithm is required to be less than a given $\delta \in (0,1)$ on all instances $\mu$. The *sample complexity* of an algorithm corresponds to its stopping time $\tau_\delta$, which counts the number of rounds before termination. An algorithm is said to be $\delta$-correct on $\mathcal{D}^K$ if $\mathbb{P}_\mu(\tau_\delta < +\infty, \hat{i}_{\tau_\delta} \notin i^\star(\mu)) \leq \delta$ for all $\mu \in \mathcal{D}^K$ [14]. We aim at designing $\delta$-correct algorithms minimizing $\mathbb{E}[\tau_\delta]$.

As done in all the literature on fixed-confidence BAI, we assume that there is a unique best arm and we denote it by $i^\star(\mu)$ or $i^\star$ when $\mu$ is clear from the context. To ensure $\delta$-correctness on $\mathcal{D}^K$, an algorithm has to be able to distinguish the unknown $\mu$ from any instance having a different best arm, hence it needs to estimates the gaps between arms. Lemma 1.1 gives a lower bound on the expected sample complexity which is known to be tight in the asymptotic regime, i.e. when $\delta$ goes to zero.

**Lemma 1.1** ([17]). *An algorithm which is $\delta$-correct on all problems in $\mathcal{D}^K$ satisfies that for all $\mu \in \mathbb{R}^K$, $\mathbb{E}_\mu[\tau_\delta] \geq T^\star(\mu) \log(1/(2.4\delta))$ where $T^\star(\mu) = \min_{\beta \in (0,1)} T_\beta^\star(\mu)$ and, for all $\beta \in (0,1)$,*

$$
T_\beta^\star(\mu)^{-1} := \max_{w \in \triangle_K : w_{i^\star(\mu)} = \beta} \min_{i \neq i^\star(\mu)} \frac{(\mu_{i^\star(\mu)} - \mu_i)^2}{2(1/\beta + 1/w_i)} .
$$

When considering the sub-class of algorithms allocating a fraction $\beta$ of their sample to the best arm, we obtain a lower bound as in Lemma 1.1 with $T_\beta^\star(\mu)$ instead of $T^\star(\mu)$. An algorithm is said to be asymptotically optimal (resp. $\beta$-optimal) if its sample complexity matches that lower bound asymptotically, that is if $\limsup_{\delta \to 0} \mathbb{E}_\mu[\tau_\delta]/\log(1/\delta) \leq T^\star(\mu)$ (resp. $T_\beta^\star(\mu)$). [38] showed the worst-case inequality $T_{1/2}^\star(\mu) \leq 2T^\star(\mu)$ for any single-parameter exponential families. Therefore, the expected sample complexity of an asymptotically $\beta$-optimal algorithm with $\beta = 1/2$ is at worst twice higher than that of any asymptotically optimal algorithm. Leveraging the symmetry of Gaussian distributions, a tighter worst-case inequality can be derived (Lemma C.6). The allocations $w^\star(\mu)$ and $w_\beta^\star(\mu)$ realizing $T^\star(\mu)$ and $T_\beta^\star(\mu)$ are known to be unique, and satisfy $\min_{i \in [K]} \min\{w^\star(\mu)_i, w_\beta^\star(\mu)_i\} > 0$. [6] showed that $2 \leq w^\star(\mu)_{i^\star}^{-1} \leq \sqrt{K-1} + 1$ for Gaussian distributions (Lemma C.4).

**Related work** The first BAI algorithms were introduced and studied under the assumption that the observation have bounded support, with a known upper bound [15, 24, 16, 19]. The sample complexity bounds proved for these algorithms scale as the sum of squared inverse gap, i.e. $H(\mu) := 2\Delta_{\min}^{-2} + \sum_{i \neq i^\star} 2(\mu_{i^\star} - \mu_i)^{-2}$ where $\Delta_{\min} := \min_{i \neq i^\star(\mu)}(\mu_{i^\star} - \mu_i)$, which satisfies $H(\mu) \leq T^\star(\mu) \leq 2H(\mu)$ [17]. Following their work, a rich literature designed asymptotically optimal algorithms in the fixed-confidence setting for parametric distributions, such as single-parameter exponential families, and non-parametric distributions such as bounded ones. Those algorithms build on two main ideas. The Tracking approach computes at each round the optimal allocation for the empirical estimator, and then tracks it [17]. To achieve lower computational cost, Game-based algorithms [12] view $T^\star(\mu)^{-1}$ as a min-max game between the learner and the nature, and design saddle-point algorithms to solve it sequentially.

Top Two algorithms arose as an identification strategy based on the Thompson Sampling algorithm for regret minimization [41]: [38] introduced Top Two Probability Sampling (TTPS) and Top Two Thompson Sampling (TTTS). Adopting a Bayesian viewpoint, Russo studied the convergence rate of the posterior probability that $i^\star$ is not the best arm, under some conditions on the prior. For Gaussian bandits, other Bayesian Top Two algorithms with frequentist components have been shown to be

asymptotically $\beta$-optimal: Top Two Expected Improvement (TTEI, [36]) and Top Two Transportation Cost (T3C, [39]). [21] introduces fully frequentist Top Two algorithms. Their analysis proves asymptotic $\beta$-optimality for several Top Two algorithms and distribution classes, beyond Gaussian. [34] provides guarantees for single-parameter exponential families, at the price of adding forced exploration. [44] proposes an algorithm to tackle the top-$k$ identification problem and introduces information-directed selection (IDS) to choose $\beta$ in an adaptive manner, which differs from the one proposed in [33]. In addition to their success in the fixed-confidence setting, Top Two algorithms have also been studied for fixed-budget problems [2], in which guarantees on the error probability should be given after $T$ samples. While existing Top Two sampling rules differ by how they choose the leader and the challenger, they all sample the leader with probability $\beta$. By design, Top Two algorithms with a fixed $\beta$ can reach $\beta$-optimality at best, and cannot be optimal on all instances $\mu$.

**Shortcomings of the asymptotic regime**  While the literature provides a detailed understanding of the asymptotic regime, many interesting questions are unanswered in the non-asymptotic regime. Recent works [9, 40, 32, 31] have shown that the sample complexity is affected by strong moderate confidence terms (independent of $\delta$). The analysis of [21] applies to their $\beta$-EB-TC algorithm whose empirical stopping times is order of magnitude larger than its competitors for $\delta = 0.01$. Since the proof of asymptotic $\beta$-optimality hides design flaws, non-asymptotic guarantees should be derived to understand which Top Two algorithms will perform well in practice for any reasonable choice of $\delta$.

## 1.2 Contributions

Our main contribution is to propose the first non-asymptotic analysis of Top Two algorithms. We identify sufficient properties of the leader (seen as a regret-minimization algorithm) for it to hold. This solves two open problems: obtaining an upper bound which (1) is non-asymptotic (Theorem 2.4 holds for any $\delta$) and (2) holds for all instances having a unique best arm (i.e. sub-optimal arms can have the same mean, which was not allowed in the analyzes of existing Top Two algorithms). As a consequence, we propose the TTUCB (Top Two UCB) algorithm which builds on the UCB algorithm.

By using tracking instead of sampling to choose between the leader and the challenger, TTUCB is the first Top Two algorithm which is asymptotically $\beta$-optimal (Theorem 2.3) and has non-asymptotic guarantees (Theorem 2.4). Our experiments reveal that TTUCB performs on par with existing Top Two algorithms, which are only proven to be asymptotically $\beta$-optimal, even for large sets of arms. Numerically, we show that considering adaptive proportions compared to a fixed $\beta = 1/2$ yields a significant speed-up on hard instances, and to a moderate improvement on random instances.

## 2 UCB-based Top Two algorithm

We propose a fully deterministic Top Two algorithm based on UCB [5], named TTUCB and detailed in Algorithm 1. We prove a non-asymptotic upper bound on the expected sample complexity holding for any instance having a unique best arm.

**Stopping and recommendation rules**  The $\sigma$-algebra $\mathcal{F}_n := \sigma(\{I_t, X_{t,I_t}\}_{t \in [n-1]})$ encompasses all the information available to the agent before time $n$. Let $N_{n,i} := \sum_{t \in [n-1]} \mathbb{1}(I_t = i)$ be the number of pulls of arm $i$ before time $n$, and its empirical mean by $\mu_{n,i} := \frac{1}{N_{n,i}} \sum_{t \in [n-1]} X_{t,I_t} \mathbb{1}(I_t = i)$.

The algorithm stops as soon as the generalized likelihood ratio exceeds a threshold $c(n-1, \delta)$, i.e.

$$\min_{i \neq \hat{i}_n} \frac{\mu_{n,\hat{i}_n} - \mu_{n,i}}{\sqrt{1/N_{n,\hat{i}_n} + 1/N_{n,i}}} \geq \sqrt{2c(n-1,\delta)}, \tag{1}$$

where we recommend $\hat{i}_n = \arg\max_{i \in [K]} \mu_{n,i}$ at time $n$. Lemma 2.1 provides an explicit threshold ensuring $\delta$-correctness, which relies on concentration inequalities derived in [27].

**Lemma 2.1.** *Let $\mathcal{C}_G$ defined in* (16) *s.t.* $\mathcal{C}_G(x) \approx x + \log(x)$. *Given any sampling rule, taking*

$$c(n,\delta) = 2\mathcal{C}_G(\log((K-1)/\delta)/2) + 4\log(4 + \log(n/2)) \tag{2}$$

*in the stopping rule* (1) *ensures $\delta$-correct for Gaussian distributions.*

**Algorithm 1** TTUCB

---

**Input:** $(\beta, \delta) \in (0,1)^2$, threshold $c : \mathbb{N} \times (0,1) \to \mathbb{R}^+$ and function $g : \mathbb{N} \to \mathbb{R}^+$.
Pull once each arm $i \in [K]$;
**for** $n > K$ **do**

    Set $\hat{\imath}_n = \arg\max_{i \in [K]} \mu_{n,i}$;

    **If** $\min_{i \neq \hat{\imath}_n} \frac{\mu_{n,\hat{\imath}_n} - \mu_{n,i}}{\sqrt{1/N_{n,\hat{\imath}_n} + 1/N_{n,i}}} \geq \sqrt{2c(n-1, \delta)}$ **then** return $\hat{\imath}_n$, **else**;

    Set $B_n^{\mathrm{UCB}} = \arg\max_{i \in [K]} \left\{ \mu_{n,i} + \sqrt{\frac{g(n)}{N_{n,i}}} \right\}$ and $C_n^{\mathrm{TC}} = \arg\min_{i \neq B_n^{\mathrm{UCB}}} \frac{(\mu_{n,B_n^{\mathrm{UCB}}} - \mu_{n,i})_+}{\sqrt{1/N_{n,B_n^{\mathrm{UCB}}} + 1/N_{n,i}}}$;

    Observe $X_{n,I_n}$ by pulling $I_n = B_n^{\mathrm{UCB}}$ if $N_{n,B_n^{\mathrm{UCB}}}^{B_n^{\mathrm{UCB}}} \leq \beta L_{n+1,B_n^{\mathrm{UCB}}}$, else $I_n = C_n^{\mathrm{TC}}$;

**end for**

---

**Sampling rule** We initialize by sampling each arms once. At time $n > K$, a Top Two sampling rule defines a leader $B_n \in [K]$ and a challenger $C_n \neq B_n$, and chooses $I_n = B_n$ or $I_n = C_n$ based on a fixed allocation $\beta$. In prior work this choice was done at random, which means that the leader was sampled with probability $\beta$. We replace randomization by tracking, and show similar theoretical and numerical results (see Figure 4 in Appendix G.2). For fixed $\beta$, we recommend to use $\beta = 1/2$ without prior knowledge on the unknown mean parameters (see Section 3.4 for adaptive proportions). This recommendation is supported theoretically by the fact that $w^\star(\mu)_{i^\star} \leq 1/2$ (Lemma C.4) and that $T_{1/2}^\star(\mu)/T^\star(\mu)$ is significantly smaller than 2 for most instances (Lemma C.6 and Figure 2).

Let $L_{n,i} := \sum_{t \in [n-1]} \mathbb{1}(B_t = i)$ be the number of time arm $i$ was the leader, and $N_{n,j}^i := \sum_{t \in [n-1]} \mathbb{1}((B_t, I_t) = (i,j))$ be the number of pulls of arm $j$ at rounds in which $i$ was the leader. We use $K$ independent tracking procedures. A tracking procedure is a deterministic method to convert a sequence of allocations over arms into a sequence of arms, which ensures that the empirical proportions are close to the averaged allocation over arms. For each leader, we track the allocation $(\beta, 1 - \beta)$ between the leader and the challenger. Formally, we set $I_n = B_n$ if $N_{n,B_n}^{B_n} \leq \beta L_{n+1,B_n}$, else $I_n = C_n$. Using Theorem 6 in [13] for each tracking procedure yields Lemma 2.2.

**Lemma 2.2.** *For all $n > K$ and all $i \in [K]$, we have $-1/2 \leq N_{n,i}^i - \beta L_{n,i} \leq 1$.*

Using tracking over randomization is motivated by practical and theoretical reasons. First, in some specific applications, the practitioner might be only willing to use a deterministic algorithm. Second, in the analysis, it is easier to control deterministic counts since it removes the need for martingales arguments to bound the deviations of the samples. Therefore, tracking simplifies the non-asymptotic analysis. Third, Lemma 2.2 shows that the speed of convergence is at least $\mathcal{O}(1/n)$ for tracking, while we would obtain a speed of $O(1/\sqrt{n})$ for randomization.

At time $n$, the UCB leader is defined as

$$B_n^{\mathrm{UCB}} = \arg\max_{i \in [K]} \{\mu_{n,i} + \sqrt{g(n)/N_{n,i}}\} , \qquad (3)$$

where $\sqrt{g(n)/N_{n,i}}$ is a bonus coping for uncertainty. Let $\alpha > 1$ and $s > 1$ be two concentration parameters. The choice of $g(n)$ should ensure that we have an upper confidence bound on $\mu_i$ holding with high probability: with probability $1 - Kn^{-s}$, for all $t \in [n^{1/\alpha}, n]$ and all arms $i \in [K]$, $\mu_i \in [\mu_{t,i} \pm \sqrt{g(t)/N_{t,i}}]$. For Gaussian observations, a function $g$ which is sufficient for the purpose of our proof can be obtained by a union bound over time, giving $g_u(n) = 2\alpha(1 + s) \log n$. We can improve on $g_u$ with mixtures of martingales, yielding $g_m(n) = \overline{W}_{-1}(2s\alpha \log(n) + 2\log(2 + \alpha \log n) + 2)$ with $\overline{W}_{-1}(x) = -W_{-1}(-e^{-x})$ for all $x \geq 1$, where $W_{-1}$ is the negative branch of the Lambert $W$ function, and $\overline{W}_{-1}(x) \approx x + \log(x)$. A UCB leader with $g_0(n) = 0$ recovers the Empirical Best (EB) leader [21]. Choosing $g$ is central for empirical performance and non-asymptotic guarantees, but not for asymptotic ones. The lowest $g$ will yield better empirical performance since larger $g$ means more conservative confidence bounds. In our experiments where $\alpha = s = 1.2$, we will consider $g_m$ since $g_m(n) \leq g_u(n)$ for $n \geq 50$.

Given a leader $B_n$, the TC challenger is defined as

$$C_n^{\mathrm{TC}} = \arg\min_{i \neq B_n} \frac{(\mu_{n,B_n} - \mu_{n,i})_+}{\sqrt{1/N_{n,B_n} + 1/N_{n,i}}} , \qquad (4)$$

where $x_+ = \max\{x, 0\}$. [39] introduced the TC challenger as a computationally efficient approximation of the challenger in TTTS [38], which uses re-sampling till an unlikely event occurs. Both T3C and TTTS use the TS leader which takes the best arm of a vector of realization drawn from a sampler, e.g. $\theta_i \sim \mathcal{N}(\mu_{n,i}, 1/N_{n,i})$ for Gaussian distributions with unit variance.

**Computational cost**  Computing the stopping rule (1) and the UCB leader (3) can be done in $\mathcal{O}(K)$. At time $n$ where $B_n$ coincides with $\hat{i}_n$, computing the TC challenger (4) is done as a by-product of the computation of the stopping rule, without additional cost. When $B_n \neq \hat{i}_n$, we draw at random an arm with larger empirical mean. The per-round computational and memory cost of TTUCB is $\mathcal{O}(K)$.

## 2.1 Sample complexity upper bound

Leveraging the unified analysis of Top Two algorithms proposed by [21], we obtain the asymptotic $\beta$-optimality of TTUCB (Theorem 2.3). After showing the required properties for the UCB leader, we proved that tracking Top Two algorithms have similar properties as their sampling-based counterparts.

**Theorem 2.3.** *Let $(\delta, \beta) \in (0,1)^2$, $s > 1$ and $\alpha > 1$. Using the threshold (2) in (1) and $g_u$ (or $g_m$) in (3), the TTUCB algorithm is $\delta$-correct and asymptotically $\beta$-optimal for all $\mu \in \mathbb{R}^K$ such that $\min_{i \neq j} |\mu_i - \mu_j| > 0$, i.e. it satisfies $\limsup_{\delta \to 0} \mathbb{E}_\mu[\tau_\delta]/\log(1/\delta) \leq T_\beta^\star(\mu)$.*

Theorem 2.3 and guarantees for other Top Two algorithms hold only for arms having distinct means. Moreover, an asymptotic result provides no guarantees on the performance in moderate regime of $\delta$. We address those two limitations.

**Non-asymptotic upper bound**  Theorem 2.4 gives an upper bound on the expected sample complexity holding for any $\delta$ and any instance having a unique best arm. It is a direct corollary of a more general result holding for any $\beta \in (0,1)$, $s > 1$ and $\alpha > 1$ (Theorem D.4).

**Theorem 2.4.** *Let $\delta \in (0,1)$. Using the threshold (2) in (1) and $g_u$ in (3) with $s = \alpha = 1.2$, the TTUCB algorithm with $\beta = 1/2$ satisfies that, for all $\mu \in \mathbb{R}^K$ such that $|i^\star(\mu)| = 1$,*

$$\mathbb{E}_\mu[\tau_\delta] \leq \inf_{w_0 \in [0,(K-1)^{-1}]} \max\left\{ T_0(\delta, w_0), C_\mu^{1.2}, C_0(w_0)^6, (2/\varepsilon)^{1.2} \right\} + 12K \,,$$

*where $C_\mu = h_1(26H(\mu))$, $C_0(w_0) = 2/(\varepsilon a_\mu(w_0)) + 1$ with $\varepsilon \in (0,1]$,*

$$T_0(\delta, w_0) = \sup\{n \mid n - 1 \leq 2T_{1/2}^\star(\mu)(1+\varepsilon)^2(1-w_0)^{-d_\mu(w_0)}(\sqrt{c(n-1, \delta)} + \sqrt{4\log n})^2\} \,,$$

*with $a_\mu(w_0) = (1 - w_0)^{d_\mu(w_0)} \max\{\min_{i \neq i^\star(\mu)} w_{1/2}^\star(\mu)_i, w_0/2\}$ and $d_\mu(w_0) = |\{i \neq i^\star(\mu) \mid w_{1/2}^\star(\mu)_i < w_0/2\}|$. The function $h_1(x) := x\overline{W}_{-1}\left(\log(x) + \frac{2+2K}{x}\right)$ is positive, increasing for $x \geq 2 + 2K$, and satisfies $h_1(x) \approx x(\log x + \log\log x)$.*

The TTUCB sampling rule using $g_m$ in (3) satisfies a similar upper bound (Corollary D.5). Since Theorem 2.4 holds for any instance having a unique best arm, we corroborate the intuition that assuming $\min_{i \neq j} |\mu_i - \mu_j| > 0$ is an artifact of the existing proof to obtain asymptotic $\beta$-optimality.

The upper bound on $\mathbb{E}_\mu[\tau_\delta]$ involves several terms. The $\delta$-dependent term is $T_0(\delta)$. In the asymptotic regime, we can show that $\limsup_{\delta \to 0} T_0(\delta)/\log(1/\delta) \leq 2T_{1/2}^\star(\mu)$ by taking $w_0 = 0$ and letting $\varepsilon$ go to zero. While there is (sub-optimal) factor 2 in $T_0(\delta)$, Theorem 2.3 shows that TTUCB is asymptotically $1/2$-optimal. This factor is a price we paid to obtain more explicit non-asymptotic terms, and removing it would require more sophisticated arguments in order to control the convergence of the empirical proportions $N_n/(n-1)$ towards $w_{1/2}^\star(\mu)$.

In the regime where $H(\mu) \to +\infty$, the upper bound is dominated by the $\delta$-independent term $C_\mu^{1.2}$ (when $\alpha = 1.2$) with satisfies $C_\mu = \mathcal{O}(H(\mu)\log H(\mu))$. Compared to the best known upper and lower bounds in this regime (see discussion below), our non-asymptotic term has a sub-optimal scaling in $\mathcal{O}((H(\mu)\log H(\mu))^\alpha)$ with $\alpha > 1$. While taking $\alpha \approx 1$ would mitigate this sub-optimality, it would yield a larger dependency in $C_0(w_0)^{\alpha/(\alpha-1)}$. Empirically, Figures 1(b) and 5 (Appendix G.2) hints that the empirical performance of TTUCB has a better scaling with $H(\mu)$ than $H(\mu)^\alpha$.

For instances such that $\min_{i \neq i^\star} w_{1/2}^\star(\mu)_i$ is arbitrarily small, taking $w_0 = 0$ yields an arbitrarily large $C_0(0)$. By clipping with $w_0/2$, we circumvent this pitfall and ensure that $C_0(w_0) = \mathcal{O}(K/\varepsilon)$.

Table 1: Upper bound on the sample complexity $\tau_\delta$ in probability (§) or in expectation (†). The notation $\mathcal{O}$ displays the dominating term when $\delta \to 0$ for the asymptotic regime, and when $H(\mu) \to +\infty$ (or $\Delta_i \to 0$) for the finite-confidence one. The notation $\tilde{\mathcal{O}}$ hides polylogarithmic factors. (*) Upper bound on $\mathbb{E}_\mu[\tau_\delta \mathbb{1}\,(\mathcal{E})]$ where $\mathbb{P}[\mathcal{E}^\complement] \le \gamma$. (**) The asymptotic upper bound holds for instances having all distinct means, while the non-asymptotic one doesn't require this assumption.

| Algorithm | Asymptotic behavior | Finite-confidence behavior |
|---|---|---|
| LUCB1† [24] | $\mathcal{O}\left(H(\mu)\log(1/\delta)\right)$ | $\mathcal{O}\left(H(\mu)\log H(\mu)\right)$ |
| Exp-Gap§ [25] | $\mathcal{O}\left(H(\mu)\log(1/\delta)\right)$ | $\mathcal{O}(\sum_{i \neq i^\star} \Delta_i^{-2} \log\log \Delta_i^{-1})$ |
| lil' UCB§ [19] | $\mathcal{O}\left(H(\mu)\log(1/\delta)\right)$ | $\mathcal{O}(\sum_{i \neq i^\star} \Delta_i^{-2} \log\log \Delta_i^{-1})$ |
| DKM† [12] | $T^\star(\mu)\log(1/\delta) + \tilde{\mathcal{O}}(\sqrt{\log(1/\delta)})$ | $\tilde{\mathcal{O}}\left(KT^\star(\mu)^2\right)$ |
| Peace§ [26] | $\mathcal{O}\left(T^\star(\mu)\log(1/\delta)\right)$ | $\mathcal{O}\left(H(\mu)\log(K/\Delta_{\min})\right)$ |
| FWS† [42] | $T^\star(\mu)\log(1/\delta) + \mathcal{O}(\log\log(1/\delta))$ | $\mathcal{O}\left(e^K H(\mu)^{19/2}\right)$ |
| EBS† [6]* | $T^\star(\mu)\log(1/\delta) + o(1)$ | $\mathcal{O}\left(KH(\mu)^4/w_{\min}^2\right)$ |
| **TTUCB**†** | $T_\beta^\star(\mu)\log(1/\delta) + \mathcal{O}(\log\log(1/\delta))$ | $\mathcal{O}\left((H(\mu)\log H(\mu))^\alpha\right)$ with $\alpha > 1$ |

Since it yields a larger $T_0(\delta)$, we are trading-off asymptotic terms for improved non-asymptotic ones. We illustrate this with two archetypal instances. For the "1-sparse" instance, in which $\mu_1 > 0$ and $\mu_i = 0$ for all $i \neq 1$, we have by symmetry that $2w^\star_{1/2}(\mu)_i = 1/(K-1)$ for all $i \neq 1$. Therefore, we have $C_0(w_0) = \mathcal{O}(K/\varepsilon)$ since $d_\mu(w_0) = 0$ for all $w_0 \in [0, 1/(K-1)]$. The "almost dense" instance is such that $\mu_1 = 1$, $\mu_K = 0$ and $\mu_i = 1 - \gamma$ for all $i \notin \{1, K\}$. By symmetry, there exists a function $h : [0, 1) \to [0, (K-1)^{-1})$ with $\lim_{\gamma \to 0} h(\gamma) = 0$, such that $2w^\star_{1/2}(\mu)_K = h(\gamma)$ and $2w^\star_{1/2}(\mu)_i = (1 - h(\gamma))/(K-2)$ for all $i \notin \{1, K\}$. While $\lim_{\gamma \to 0} C_0(0) = +\infty$, we obtain $\lim_{\gamma \to 0} C_0(w_0) = \mathcal{O}(K/\varepsilon)$ by taking $w_0 = (1 - h(\gamma))/(K-2)$ since $d_\mu(w_0) = 1$.

**Comparison with existing upper bounds**   Table 1 summarizes the asymptotic and non-asymptotic scalings of the upper bound on the sample complexity of existing BAI algorithms. Among the class of asymptotically ($\beta$-)optimal algorithms, very few of them also enjoy non-asymptotic guarantees, e.g. the analyses of Track-and-Stop and Top Two algorithms are asymptotic. The gamification approach of [12] is the first attempt to provide both. Their non-asymptotic upper bound on $\mathbb{E}_\mu[\tau_\delta]$ involves an implicit time $T_1(\delta)$ which scales with $KT^\star(\mu)^2$ and is only valid for $\log(1/\delta) \gtrsim KT^\star(\mu)$ (see Lemma 2, with constants in Appendix D.7). Let $T_\delta^\star := T^\star(\mu)\log(1/\delta)$. As a first order approximation, they obtain $T_1(\delta) \approx T_\delta^\star + \Theta\left(\sqrt{T_\delta^\star \log T_\delta^\star}\right)$, and we obtain $T_0(\delta) \approx \Theta\left(T_\delta^\star + \log T_\delta^\star\right)$ (Lemma D.13). [42] were the first to obtain an upper bound on $\mathbb{E}_\mu[\tau_\delta]$ of the form $\Theta(T_\delta^\star + \log\log(1/\delta))$. While they improved the second-order $\delta$-dependent term, the $\delta$-independent term scales with $e^K H(\mu)^{19/2}$ (see their Theorem 2 for $\varepsilon^{-1} \gtrsim T^\star(\mu)$, with constants given by Appendix N). The algorithm proposed by [6] has a non-asymptotic upper bound on $\mathbb{E}_\mu[\tau_\delta \mathbb{1}\,(\mathcal{E})]$ of the form $(1+\varepsilon)T_\delta^\star + f(\mu, \delta)$ which is valid for $\log(1/\delta) \gtrsim w_{\min}^{-2}K/\Delta_{\min}$, where $\mathcal{E}$ is such that $\mathbb{P}_\mu(\mathcal{E}^\complement) \le \gamma$. Since $f(\mu, \delta) =_{\delta \to 0} o(1)$, they obtain a better $\delta$-dependency. However, $f(\mu, \delta)$ is arbitrarily large when $w_{\min} := \min_{i \in [K]} w^\star(\mu)_i$ is arbitrarily small since it scales with $KH(\mu)^4/w_{\min}^2$. Therefore, they suffer from the pitfall which we avoided by clipping. In light of Table 1, TTUCB enjoys the best scaling when $H(\mu) \to +\infty$ in the class of asymptotically ($\beta$-)optimal BAI algorithms.

The LUCB1 algorithm [24] has a structure similar to a Top Two algorithm, with the difference that LUCB samples both the leader and the challenger instead of choosing one. As LUCB1 satisfies $\mathbb{E}_\mu[\tau_\delta] \le 292H(\mu)\log(H(\mu)/\delta) + 16$, it enjoys better scaling when $H(\mu) \to +\infty$ than TTUCB. Since the empirical allocation of LUCB1 is not converging towards $w^\star_{1/2}(\mu)$, it is not asymptotically $1/2$-optimal. The Peace algorithm [26] has a non-asymptotic upper bound on $\tau_\delta$ of the form $\mathcal{O}((T_\delta^\star + \gamma^\star(\mu))\log(K/\Delta_{\min}))$ holding with probability $1 - \delta$. The term $\gamma^\star(\mu)$ is a Gaussian-width which originates from concentration on the suprema of Gaussian processes and satisfies $\gamma^\star(\mu) = \Omega(H(\mu))$.

Another class of BAI algorithms focus on the dependency in the gaps $\Delta_i := \mu_{i^\star} - \mu_i$, and derive non-asymptotic upper bound on $\tau_\delta$ holding with high probability. [25, 19, 8, 9] gives $\delta$-PAC algorithms with an upper bound of the form $\mathcal{O}(H(\mu)\log(1/\delta) + \sum_{i \neq i^\star} \Delta_i^{-2} \log\log \Delta_i^{-1})$, and [19] shows that

for two arms the dependency $\Delta^{-2} \log \log \Delta^{-1}$ is optimal when $\Delta \to 0$. While those algorithms obtain the best scaling when $H(\mu) \to +\infty$, they are not asymptotically ($\beta$-)optimal.

# 3 Non-asymptotic analysis

## 3.1 Proof sketch of Theorem 2.4

Existing analyses of Top Two algorithms are asymptotic in nature and requires too much control on the empirical means and proportions to yield any meaningful information in the finite-confidence regime. Therefore, we adopt a different approach which ressembles the non-asymptotic analysis of [12]. We first define concentration events to control the deviations of the random variables used in the UCB leader and the TC challenger. For all $n > K$, let $\mathcal{E}_n := \bigcap_{i \in [K]} \bigcap_{t \in [n^{5/6}, n]} (\mathcal{E}^1_{t,i} \cap \mathcal{E}^2_{t,i})$ where

$$\mathcal{E}^1_{t,i} := \{\sqrt{N_{t,i}}|\mu_{t,i} - \mu_i| < \sqrt{6\log t}\} \quad \text{and} \quad \mathcal{E}^2_{t,i} := \{\frac{(\mu_{t,i^\star} - \mu_{t,i}) - (\mu_{i^\star} - \mu_i)}{\sqrt{1/N_{t,i^\star} + 1/N_{t,i}}} > -\sqrt{8\log t}\} \,.$$

Using Lemmas D.8 and E.6, the proof boils down to constructing a time $T(\delta)$ after which $\mathcal{E}_n \subset \{\tau_\delta \le n\}$ for $n > T(\delta)$ since it would yield that $\mathbb{E}_\mu[\tau_\delta] \le T(\delta) + 12K$.

Let $n > K$ such that $\mathcal{E}_n \cap \{n < \tau_\delta\}$ holds true, and $t \in [n^{5/6}, n]$ such that $B_t^{\text{UCB}} = i^\star$. Using that $t \le n < \tau_\delta$, under $\bigcap_{i \ne i^\star} \mathcal{E}^2_{t,i}$, the stopping condition yields that

$$\sqrt{2c(n-1, \delta)} \ge ((\mu_{i^\star} - \mu_{C_t^{\text{TC}}})(1/N_{t,i^\star}^{i^\star} + 1/N_{t,C_t^{\text{TC}}}^{i^\star})^{-1/2} - \sqrt{8\log n})_+ \,.$$

Let $w^\star_{1/2}$ be the unique element of $w^\star_{1/2}(\mu)$. Lemma 3.1 links the empirical proportions $N_{t,i}^{i^\star}/(t-1)$ to $w^\star_{1/2,i}$ for $i \in \{i^\star, C_t^{\text{TC}}\}$. It is the key technical challenge of our non-asymptotic proof strategy.

**Lemma 3.1.** *Let $\varepsilon \in (0, 1]$. There exist $T_\mu > 0$ such that for all $n > T_\mu$ such that $\mathcal{E}_n \cap \{n < \tau_\delta\}$ holds true, there exists $t \in [n^{5/6}, n]$ with $B_t^{UCB} = i^\star$, which satisfies*

$$(n-1)(1/N_{t,i^\star}^{i^\star} + 1/N_{t,C_t^{\text{TC}}}^{i^\star}) \le (1+\varepsilon)^2(2 + 1/w^\star_{1/2,C_t^{\text{TC}}})/\beta \,.$$

Before proving Lemma 3.1, we conclude the proof of Theorem 2.4. Let $\varepsilon$, $T_\mu$ and $t$ as in Lemma 3.1 and $T(\delta) := \sup\{n \mid n - 1 \le T^\star_{1/2}(\mu)(1+\varepsilon)^2(\sqrt{c(n-1, \delta)} + \sqrt{4\log n})^2/\beta\}$. For all $n > \max\{T_\mu, T(1)\}$, we have $\sqrt{c(n-1, \delta)} + \sqrt{4\log n} \ge \sqrt{\beta(n-1)T^\star_{1/2}(\mu)^{-1}(1+\varepsilon)^{-2}}$. Therefore, we have proved that $\mathcal{E}_n \cap \{n < \tau_\delta\} = \emptyset$ for all $n > \max\{T_\mu, T(\delta)\}$. This concludes the proof.

Provided that $B_t = i^\star$, the above only used the stopping condition and the TC challenger, and no other properties of the leader. Lemma 3.2 shows that $B_t^{\text{UCB}} = i^\star$, except for a sublinear number of times. Section 3.3 exhibits sufficient conditions on a regret minimization algorithm to obtain a non-asymptotic upper bound.

**Lemma 3.2.** *Under the event $\bigcap_{k \in [K]} \bigcap_{t \in [n^{5/6}, n]} \mathcal{E}^1_{t,k}$, we have $L_{n,i^\star} \ge n - 1 - 24H(\mu)\log n - 2K$.*

**Proof sketch of Lemma 3.1** The key technical challenge is to link $N_{t,C_t^{\text{TC}}}^{i^\star}/(n-1)$ with $w^\star_{1/2,C_t^{\text{TC}}}$. We adopt the approach used to analyze of APT [30]: consider an arm being over-sampled and study the last time this arm was pulled. By the pigeonhole principle, at time $n$,

$$\exists k_1 \ne i^\star, \text{ s.t. } N_{n,k_1}^{i^\star} \ge 2(L_{n,i^\star} - N_{n,i^\star}^{i^\star})w^\star_{1/2,k_1} \,. \tag{5}$$

Let $t_1$ be the last time at which $B_t^{\text{UCB}} = i^\star$ and $C_t^{\text{TC}} = k_1$, hence $N_{t_1,k_1}^{i^\star} \ge N_{n,k_1}^{i^\star} - 1$. Using Lemmas 2.2 and 3.2, we show that $N_{t_1,k_1}^{i^\star} \gtrsim w^\star_{1/2,k_1}(n-1)$, hence $t_1 \ge n^{5/6}$ for $n$ large enough (see Appendix D.2). Then, we need to link $N_{t_1,i^\star}^{i^\star}$ to $(n-1)/2$. When $w^\star_{1/2,k_1}$ is small, (5) can be true at $t_1 = n^{5/6}$, hence there is no hope to show that $t_1 = n - o(n)$. To circumvent this problem, we link $N_{t_1,i^\star}^{i^\star}$ to $N_{t_1,k_1}^{i^\star}$ thanks to Lemma 2.2, and use that

$$\frac{n-1}{N_{t_1,i^\star}^{i^\star}} + \frac{n-1}{N_{t_1,k_1}^{i^\star}} \le \left(2 + \frac{n-1}{N_{t_1,k_1}^{i^\star}}\right)\left(\frac{N_{t_1,k_1}^{i^\star}}{N_{t_1,i^\star}^{i^\star}} + 1\right) \le 2(1+\varepsilon)^2(2 + 1/w^\star_{1/2,k_1}) \,,$$

for $n > T_\mu(w_-)$ with $T_\mu(w_-) \leq \max\{C_\mu^{1.2}, (2/(\varepsilon w_-) + 1)^6, (2/\varepsilon)^{1.2}\}$ (Lemmas D.10 and D.11), where $w_- = \min_{i \neq i^\star} w_{1/2,i}^\star > 0$ lower bounds $w_{1/2,k_1}^\star$. This concludes the proof for $w_0 = 0$. The (sub-optimal) multiplicative factor 2 in $T_0(\delta)$ comes from the inequality (6). To remove it, we need to control the deviation between the empirical proportion of arm $i$ and $w_{1/2,i}^\star$ for all $i \in [K]$. Nevertheless, TTUCB is asymptotically $1/2$-optimal (Theorem 2.3).

**Refined analysis** For $w_0 \in (0, (K-1)^{-1}]$, we clip $\min_{i \neq i^\star} w_{1/2,i}^\star$ by $w_0/2$ (see Appendix D). Our method can be used to analyze other algorithms, and it improves existing results on APT.

## 3.2 Beyond Gaussian distributions

Theorems 2.3 and 2.4 hold for sub-Gaussian r.v. thanks to direct adaptations of concentration results (Lemmas 2.1, E.2 and E.5). The situation is akin to the regret bound of UCB: it holds for any sub-Gaussian, but it is close to optimality in a distribution-dependent sense only for Gaussians. However, if the focus is on asymptotically $\beta$-optimal algorithms, then it is challenging to express the characteristic time $T^\star(\mu)$ for the non-parametric class of sub-Gaussian distributions.

The TTUCB algorithm can also be defined for more general distributions such as single-parameter exponential families or bounded distributions. It is only a matter of adapting the definition of the UCB leader and the TC challenger. For bounded distributions, the UCB leader was studied in [1] and the TC challenger was analyzed in [21]. Leveraging their unified analysis of Top Two algorithms with our tracking-based results, we can show asymptotic $\beta$-optimality of TTUCB for bounded distributions and single-parameter exponential families with sub-exponential tails. We believe that non-asymptotic guaranties could be obtained for more general distributions, but it will come at the price of more technical arguments and less explicit non-asymptotic terms.

## 3.3 Generic regret minimizing leader

Our non-asymptotic analysis highlights that any regret minimization algorithm that selects the arm $i^\star$ except for a sublinear number of times (Property 1) can be used as leader with the TC challenger.

*Property* 1. There exists $(\tilde{\mathcal{E}}_n)_n$ with $\sum_n \mathbb{P}_\mu(\tilde{\mathcal{E}}_n^\complement) < +\infty$ and a function $h$ with $h(n) = \mathcal{O}(n^\gamma)$ for some $\gamma \in (0,1)$ such that under event $\tilde{\mathcal{E}}_n$, $L_{n,i^\star} \geq n - 1 - h(n)$.

For asymptotic guaranties, the sufficient properties on the leader from [21] are weaker since they are even satisfied by the greedy choice $B_n = \hat{i}_n$. While Top Two algorithms were introduced by [38] to adapt Thompson Sampling to BAI, we have shown that other regret minimization algorithms can be used: *the Top Two method is a generic wrapper to convert any regret minimization algorithm into a best arm identification strategy*.

The regret of an algorithm at time $n$, $\bar{R}_n = \sum_{i \neq i^\star} \Delta_i N_{n,i}$, is almost always studied through its expectation $\mathbb{E}[\bar{R}_n]$. This is however not sufficient for our application. We need to prove that with high probability, $N_{n,i}$ is small for all arm $i \neq i^\star$. Such guarantees are known for UCB [3] and ETC [29], but are yet unknown for Thompson Sampling. We cannot in general obtain a good enough bound on $N_{n,i}$ from a bound on $\mathbb{E}[\bar{R}_n]$. However, we can if we have high probability bounds on $\bar{R}_n$. Suppose that a regret minimization algorithm $\mathrm{Alg}_1$ satisfies Property 2 and is independent of the horizon $n$.

*Property* 2. There exists $s > 1$, $\gamma \in (0,1)$, $(\mathcal{E}_{n,\delta})_{(n,\delta)}$ with $\sum_n \mathbb{P}_\mu[\mathcal{E}_{n,n^{-s}}^\complement] < +\infty$ and a function $h$ with $h(n, n^{-s}) = \mathcal{O}(n^\gamma)$ such that under event $\mathcal{E}_{n,\delta}$, $\bar{R}_n \leq h(n, \delta)$.

Let $\mathrm{Alg}_2$ be the algorithm $\mathrm{Alg}_1$ used in a Top Two procedure, but which uses only the observations obtained at times $n$ such that $I_n = B_n$ and discards the rest. Let $\tilde{\mathcal{E}}_n = \mathcal{E}_{n,n^{-s}}$ and $\Delta_{\min} = \min_{i \neq i^\star} \Delta_i$. Then, under $\tilde{\mathcal{E}}_n$, $\mathrm{Alg}_2$ satisfies $\sum_{i \neq i^\star} N_{n,i} \leq h(n, n^{-s})/\Delta_{\min}$ and Lemma 2.2 yields $N_{n,i^\star}^{i^\star} \geq \beta(n-1) - h(n, n^{-s})/\Delta_{\min} - K/2$. Therefore, Property 1 holds for $\tilde{\mathcal{E}}_n$ and $h(n) = (h(n, n^{-s})/\Delta_{\min} + K/2 + 1)/\beta$. Given a specific algorithm, a finer analysis could avoid discarding information by using $\mathrm{Alg}_1$ with every observations.

## 3.4 Adaptive proportions

Given a fixed allocation $\beta$, any Top Two algorithm can at best be asymptotically $\beta$-optimal. Since the optimal allocation $\beta^\star \in \arg\min_{\beta \in (0,1)} T_\beta^\star(\mu)$ is unknown, it should be learned from the observations by a Top Two algorithm using an adaptive proportion $\beta_n$ at time $n$. Recently, [44] proposes IDS to choose $\beta_n$ in an adaptive manner. For BAI with Gaussian observations, IDS yields $\beta_n = N_{n,C_n}/(N_{n,B_n} + N_{n,C_n})$. Let $\bar{\beta}_n^i := \frac{1}{L_{n,i}} \sum_{t \in [n-1]} \beta_t \mathbb{1}(B_t = i)$ be the average proportion when arm $i$ was the leader before time $n$. Tracking with IDS requires to use $\bar{\beta}_{n+1}^{B_n}$ instead of $\beta$. Using the analysis of [44], it is reasonable to believe that one could obtain asymptotic optimality of TTUCB with IDS. However it is not clear how to adapt the non-asymptotic analysis since it heavily relies on $\beta$ being fixed and bounded away from $\{0, 1\}$. Experiments with IDS are available in Appendix G.2.1.

## 4 Experiments

In the moderate regime ($\delta = 0.1$), we assess the empirical performance of TTUCB with bonus $g_m$ and concentration parameters $s = \alpha = 1.2$. As benchmarks, we compare our algorithm with three sampling-based Top Two algorithms: TTTS, T3C and $\beta$-EB-TCI. In addition, we consider Track-and-Stop (TaS) [17], FWS [42], DKM [12], LUCB [24] and uniform sampling. At time $n$, the LUCB algorithm computes a leader and a challenger, then sample them both (see Appendix G.1). To provide a clear comparison with Top Two algorithms, we define a new $\beta$-LUCB algorithm which sample the leader with probability $\beta$, else sample the challenger. At the exception of LUCB and $\beta$-LUCB which have their own stopping rule, all algorithms uses the stopping rule (1) with the heuristic threshold $c(n, \delta) = \log((1 + \log n)/\delta)$. Even though this choice is not sufficient to prove $\delta$-correctness, it yields an empirical error which is several orders of magnitude lower than $\delta$. Top Two algorithms and $\beta$-LUCB use $\beta = 1/2$. To allow for a fair numerical comparison, LUCB and $\beta$-LUCB use $\sqrt{2c(n-1, \delta)/N_{n,i}}$ as bonus, which is too tight to yield valid confidence intervals. Supplementary experiments are available in Appendix G.

**Random instances** We assess the performance on 5000 random Gaussian instances with $K = 10$ such that $\mu_1 = 0.6$ and $\mu_i \sim \mathcal{U}([0.2, 0.5])$ for all $i \neq 1$. Numerically, we observe $w^\star(\mu)_{i^\star} \approx 1/3 \pm 0.02$ (mean $\pm$ std). In Figure 1(a), we see that TTUCB performs on par with existing Top Two algorithms, and slightly outperforms TaS and FWS. Our algorithm achieves significantly better result than DKM, LUCB, 1/2-LUCB and uniform sampling. The CPU running time is reported in Table 4, and the observed empirical errors before stopping is displayed in Figure 3 (Appendix G.2).

**Larger sets of arms** We evaluate the impact of larger number of arms. The "1-sparse" scenario of [20] sets $\mu_1 = 1/4$ and $\mu_i = 0$ for all $i \neq 1$, i.e. $H(\mu) = 32(K - 1)$ (see Appendix G.2 for other instances). We consider algorithms with low computational cost. In Figure 1(b), all algorithms have the same linear scaling in $K$ (i.e. in $H(\mu)$). Faced with an increase in the number of arms, the TS leader used in T3C appears to be more robust than the UCB leader in TTUCB. This is a common feature of UCB algorithms which have to overcome the bonus of sub-optimal arms.

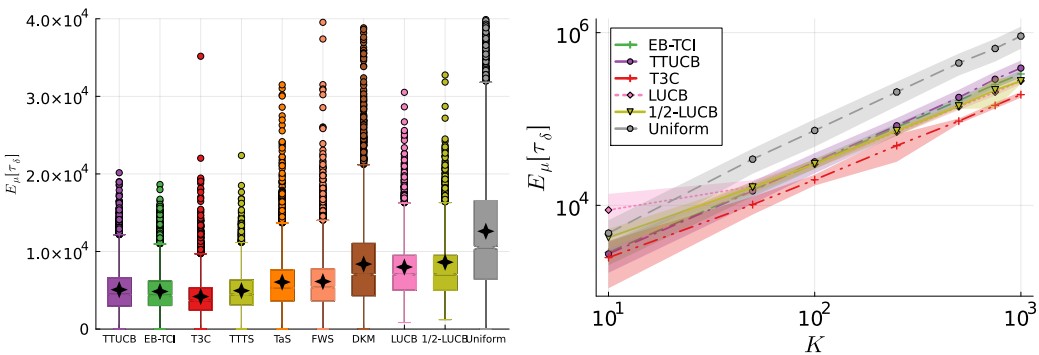

Figure 1: Empirical stopping time on (a) random instances ($K = 10$) and (b) "1-Sparse" instances.

## 5  Conclusion

In this paper, we have shown the first non-asymptotic upper bound on the expected sample complexity of a Top Two algorithm, which holds for any error level and for any instance having a unique best arm. Furthermore, we have demonstrated that the TTUCB algorithm achieves competitive empirical performance compared to other algorithms, including Top Two methods.

While our guarantees hold for a fixed proportion $\beta$ allocated to the leader, [44] recently introduced IDS to define an adaptive proportion $\beta_n$ at time $n$ and show asymptotic optimality for Gaussian distributions. Deriving guarantees for IDS for single-parameter exponential families is a challenging open problem. Finally, Top Two algorithms are a promising method to tackle complex settings. While heuristics exist for some structured bandits such as Top-$k$, it would be interesting to efficiently adapt Top Two methods to deal with sophisticated structure, e.g. linear bandits.

## Acknowledgments and Disclosure of Funding

Experiments presented in this paper were carried out using the Grid'5000 testbed, supported by a scientific interest group hosted by Inria and including CNRS, RENATER and several Universities as well as other organizations (see https://www.grid5000.fr). This work has been partially supported by the THIA ANR program "AI_PhD@Lille". The authors acknowledge the funding of the French National Research Agency under the project FATE (ANR22-CE23-0016-01).

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

# A Outline

The appendices are organized as follows:

- Appendix B gathers notation used in this work.
- In Appendix C, we study the link between $T^\star(\mu)$ and $T_\beta^\star(\mu)$ for Gaussian distributions.
- The detailed analysis of our non-asymptotic upper bound (Theorem 2.4), sketched in Section 3, is detailed in Appendix D. We also give a non-asymptotic upper bound on the TTUCB using $g_m$ in (3) (Corollary D.5), and on uniform sampling (Theorem D.6).
- We show the asymptotic optimality of our algorithm (Theorem 2.3) in Appendix F.
- Appendix E gathers concentration results used by the stopping rule (Lemma 2.1) and the sampling rule.
- Implementation details and supplementary experiments are detailed in Appendix G.

Table 2: Notation for the setting.

| Notation | Type | Description |
|---|---|---|
| $K$ | $\mathbb{N}$ | Number of arms |
| $\mu_i$ | $\mathbb{R}$ | Mean of arm $i \in [K]$ |
| $\mu$ | $\mathbb{R}^K$ | Vector of means, $\mu := (\mu_i)_{i \in [K]}$ |
| $i^\star$ | $\mathbb{R}^K \to [K]$ | Best arm operator, $i^\star(\mu) = \arg\max_{i \in [K]} \mu_i$ |
| $T^\star(\mu), T_\beta^\star(\mu)$ | $\mathbb{R}_+^\star$ | Asymptotic ($\beta$-)characteristic time |
| $w^\star(\mu), w_\beta^\star(\mu) = \{(w_{\beta,i}^\star)_{i \in [K]}\}$ | $\triangle_K$ | Asymptotic ($\beta$-)optimal allocation |

# B Notation

We recall some commonly used notation: the set of integers $[n] := \{1, \cdots, n\}$, the complement $X^{\complement}$ and interior $\mathring{X}$ of a set $X$, Landau's notation $o, \mathcal{O}, \Omega$ and $\Theta$, the $(K-1)$-dimensional probability simplex $\triangle_K := \left\{ w \in \mathbb{R}_+^K \mid w \geq 0, \sum_{i \in [K]} w_i = 1 \right\}$. While Table 2 gathers problem-specific notation, Table 3 groups notation for the algorithms. We emphasize that $N_{n,i}^i$ is the number of times where we pulled arm $i$ as a leader before time $n$.

Table 3: Notation for algorithms.

| Notation | Type | Description |
|---|---|---|
| $B_n$ | $[K]$ | Leader at time $n$ |
| $C_n$ | $[K]$ | Challenger at time $n$ |
| $I_n$ | $[K]$ | Arm sampled at time $n$ |
| $\beta$ | $(0,1)$ | Proportion parameter |
| $X_{n,I_n}$ | $\mathbb{R}$ | Sample observed at the end of time $n$, i.e. $X_{n,I_n} \sim \mathcal{N}(\mu_{I_n}, 1)$ |
| $\mathcal{F}_n$ | | History before time $n$, i.e. $\mathcal{F}_n := \sigma(I_1, X_{1,I_1}, \cdots, I_n, X_{n,I_n})$ |
| $\hat{\imath}_n$ | $[K]$ | Arm recommended before time $n$, i.e. $\hat{\imath}_n \in \arg\max_{i \in [K]} \mu_{n,i}$ |
| $\tau_\delta$ | $\mathbb{N}$ | Sample complexity (stopping time of the algorithm) |
| $\hat{\imath}$ | $[K]$ | Arm recommended by the algorithm |
| $c(n,\delta)$ | $\mathbb{R}_+^\star$ | Stopping threshold function |
| $N_{n,i}$ | $\mathbb{N}$ | Number of pulls of arm $i$ before time $n$, $N_{n,i} := \sum_{t \in [n-1]} \mathbb{1}(I_t = i)$ |
| $\mu_{n,i}$ | $\mathcal{I}$ | Empirical mean of arm $i$ before time $n$, $\mu_{n,i} := \frac{1}{N_{n,i}} \sum_{t \in [n-1]} X_{t,I_t} \mathbb{1}(I_t = i)$ |
| $L_{n,i}$ | $\mathbb{N}$ | Counts of $B_t = i$ before time $n$, $L_{n,i} := \sum_{t \in [n-1]} \mathbb{1}(B_t = i)$ |
| $N_{n,j}^i$ | $\mathbb{N}$ | Counts of $(B_t, I_t) = (i,j)$ before time $n$, $N_{n,j}^i := \sum_{t \in [n-1]} \mathbb{1}((B_t, I_t) = (i,j))$ |

## C  Characteristic times

Let $\mu \in \mathcal{D}^K$ such that $i^\star(\mu) = \{i^\star\}$. Let $\beta \in (0,1)$ and $w_\beta^\star$ be the unique allocation $\beta$-optimal allocation satisfying $w_{\beta,i}^\star > 0$ for all $i \in [K]$ (Lemma C.2), i.e. $w_\beta^\star(\mu) = \{w_\beta^\star\}$ where

$$w_\beta^\star(\mu) := \underset{w \in \triangle_K : w_{i^\star} = \beta}{\arg\max} \, \min_{i \neq i^\star} \frac{(\mu_{i^\star} - \mu_i)^2}{2(1/\beta + 1/w_i)} = \underset{w \in \triangle_K : w_{i^\star} = \beta}{\arg\max} \, \min_{i \neq i^\star} \frac{\mu_{i^\star} - \mu_i}{\sqrt{1/\beta + 1/w_i}} \, .$$

We restate without proof two fundamental results on the characteristic time and the associated allocation, which were first shown in [38]. Lemma C.1 gives an upper bound on $T_\beta^\star(\mu)/T^\star(\mu)$ and Lemma C.2 shows that the ($\beta$-)optimal allocation is unique with strictly positive values. [38] shows that these two results hold for any single-parameter exponential families. [21] extended their proof for the non-parametric family of bounded distributions. Moreover, they argue that these results should hold for more general distributions provided some regularity assumptions are satisfied.

**Lemma C.1** ([38]). $T_{1/2}^\star(\mu) \leq 2T^\star(\mu)$ and with $\beta^\star = w_{i^\star}^\star(\mu)$,

$$\frac{T_\beta^\star(\mu)}{T^\star(\mu)} \leq \max\left\{ \frac{\beta^\star}{\beta}, \frac{1-\beta^\star}{1-\beta} \right\} \, .$$

**Lemma C.2** ([38]). If $i^\star(\mu)$ is a singleton and $\beta \in (0,1)$, then $w^\star(\mu)$ and $w_\beta^\star(\mu)$ are singletons, i.e. the optimal allocations are unique, and $w^\star(\mu)_i > 0$ and $w_\beta^\star(\mu)_i > 0$ for all $i \in [K]$.

**Gaussian distributions**   Since Lemma C.1 is a worst-case inequality holding for general distributions, we expect that tighter inequality can be achieved for Gaussian distributions by leveraging their symmetry. This intuition is fueled by recent results of [6]. Using a rewriting of the optimization problem underlying $T^\star(\mu)$ (Lemma C.3), they provide a better understanding of characteristic times and their optimal allocations (Lemma C.4). In particular, for Gaussian distributions, Lemma C.4 shows that the optimal allocation of arm $i^\star$ is never above $1/2$ and is larger than $1/(\sqrt{K-1} + 1) \geq 1/K$.

**Lemma C.3** (Proposition 8 in [6]). Let $\mu \in \mathbb{R}^K$ be a $K$-arms Gaussian bandits and $r(\mu)$ be the solution of $\psi_\mu(r) = 0$, where

$$\forall r \in (1/\min_{i \neq i^\star}(\mu_{i^\star} - \mu_i)^2, +\infty), \quad \psi_\mu(r) = \sum_{i \neq i^\star} \frac{1}{(r(\mu_{i^\star} - \mu_i)^2 - 1)^2} - 1 \, ,$$

and $\psi_\mu$ is convex and decreasing. Then,

$$T^\star(\mu) = \frac{2r(\mu)}{1 + \sum_{i \neq i^\star} \frac{1}{r(\mu)(\mu_{i^\star} - \mu_i)^2 - 1}} \, .$$

**Lemma C.4** (Proposition 10 in [6]). Let $\mu \in \mathbb{R}^K$ be a $K$-arms Gaussian bandits. For $K = 2$,

$$w^\star(\mu) = (0.5, 0.5) \quad and \quad T^\star(\mu) = 8(\mu_1 - \mu_2)^2 \, .$$

For $K \geq 3$, we have

$$1/(\sqrt{K-1} + 1) \leq w^\star(\mu)_{i^\star} \leq 1/2 \, ,$$

and

$$\max\left\{ \frac{8}{\min_{i \neq i^\star}(\mu_{i^\star} - \mu_i)^2}, 4\frac{1 + \sqrt{K-1}}{\overline{\Delta}^2} \right\} \leq T^\star(\mu) \leq 2\frac{(1 + \sqrt{K-1})^2}{\min_{i \neq i^\star}(\mu_{i^\star} - \mu_i)^2} \, ,$$

where $\overline{\Delta}^2 = \frac{1}{K-1} \sum_{i \neq i^\star}(\mu_{i^\star} - \mu_i)^2$. In particular, the equalities $w^\star(\mu)_{i^\star} = 1/(\sqrt{K-1} + 1)$ and $T^\star(\mu) = \frac{2(1 + \sqrt{K-1})^2}{\min_{i \neq i^\star}(\mu_{i^\star} - \mu_i)^2}$ are reached if and only if $\mu_i = \max_{i \neq i^\star} \mu_i$ for all $i \neq i^\star$.

By inspecting the proof of Proposition 8 in [6], we obtain directly the following rewriting of $T_\beta^\star(\mu)$.

**Lemma C.5.** Let $\mu \in \mathbb{R}^K$ be a $K$-arms Gaussian bandits and $r_\beta(\mu)$ be the solution of $\varphi_{\mu,\beta}(r) = 0$, where

$$\forall r \in (1/\min_{i \neq i^\star}(\mu_{i^\star} - \mu_i)^2, +\infty), \quad \varphi_{\mu,\beta}(r) = \sum_{i \neq i^\star} \frac{1}{r(\mu_{i^\star} - \mu_i)^2 - 1} - \frac{1 - \beta}{\beta} \, ,$$

and $\varphi_{\mu,\beta}$ is convex and decreasing. Then,

$$T_\beta^\star(\mu) = \frac{2r_\beta(\mu)}{\beta} \, .$$

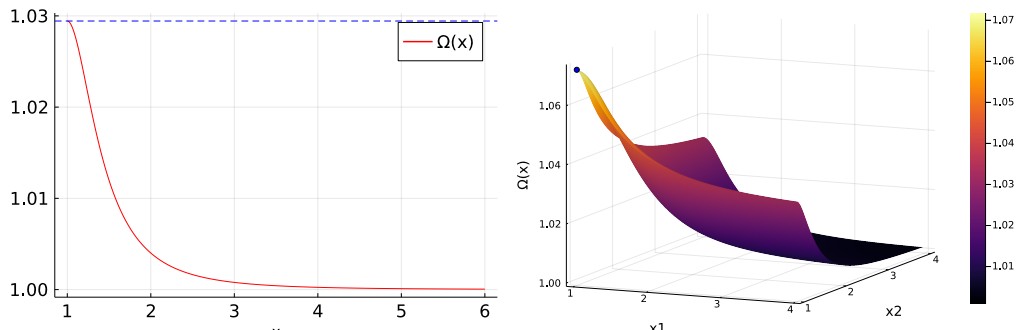

Figure 2: Ratio of characteristic times $\Omega(x) = T^\star_{1/2}(\mu)/T^\star(\mu)$ for $K = 3$ (left) and $K = 4$ (right). The dashed blue line is $r_2 = 6/(1 + \sqrt{2})^2$ (left), and the blue point is $r_3 = 8/(1 + \sqrt{3})^2$ (right).

*Proof.* Using the proof of Proposition 8 in [6], we obtain directly that

$$2T^\star_\beta(\mu)^{-1} = C_\beta(\mu)$$

where $C_\beta(\mu)$ is the solution of $\Phi_\mu(C) = 0$ where

$$\Phi_\mu(C) = \sum_{i \neq i^\star} \frac{\beta}{\beta(\mu_{i^\star} - \mu_i)^2/C - 1} - (1 - \beta) \,.$$

The idea behind the above result is that at the equilibrium, i.e. at $w^\star$, all the transportation costs are equal to $C$. Then, the implicit equation defining $C$ is obtained by using the constraints that $\sum_{i \neq i^\star} w^\star_i = 1 - \beta$. To conclude, we simply use $r = \beta/C$.

Since it is the sum of $K - 1$ convex and decreasing functions, $\varphi_{\mu,\beta}$ is also convex and decreasing. $\square$

Lemma C.6 aims at improving the worst-case inequality between $T^\star_{1/2}(\mu)$ and $T^\star(\mu)$ in the Gaussian setting. For $K = 2$, those two quantities are equal. For $K \geq 3$, we showed that $\max_{\mu:|i^\star(\mu)|=1} T^\star_{1/2}(\mu)/T^\star(\mu)$ is at least $r_K$, which is achieved when all sub-optimal arms have the same mean. As the gradient of the ratio is the null vector for those instances, we conjecture this is the maximum, i.e. $T^\star_{1/2}(\mu) \leq r_K T^\star(\mu)$. Our conjecture is supported by numerical simulations for $K \geq 3$. In Figure 2, we plot the ratio of characteristic times $\Omega(x) = T^\star_{1/2}(\mu)/T^\star(\mu)$ for $K \in \{3, 4\}$. We observed that our conjecture is validated empirically, and that $T^\star_{1/2}(\mu)/T^\star(\mu)$ is often close to 1.

**Lemma C.6.** *For $K = 2$, we have $T^\star_{1/2}(\mu) = T^\star(\mu)$. For $K \geq 3$, let $r_K = 2K/(1 + \sqrt{K - 1})^2$. Then, for all $\mu$ such that $i^\star(\mu)$ is unique, we have $\frac{T^\star_{1/2}(\mu)}{T^\star(\mu)} = \Omega(x)$ where $x_j = \frac{\mu_{i^\star} - \mu_j}{\mu_{i^\star} - \mu_{j^\star}} \geq 1$ for all $j \notin \{i^\star, j^\star\}$ with $j^\star \in \arg\min_{j \neq i^\star} \mu_{i^\star} - \mu_j$. In other words, $\frac{T^\star_{1/2}(\mu)}{T^\star(\mu)}$ is independent from $\mu_{i^\star}$ and $\min_{j \neq i^\star} \mu_{i^\star} - \mu_j$. Moreover, we have*

$$\Omega(1_{K-2}) = r_K \quad and \quad \nabla_x \Omega(1_{K-2}) = 0_{K-2} \,.$$

*Proof.* Using Lemma C.4, we have $T^\star_{1/2}(\mu) = T^\star(\mu)$ directly for $K = 2$.

For $K \geq 3$, we want to upper bound $T^\star_{1/2}(\mu)/T^\star(\mu)$. If we denote by $\Delta_{\min}(\mu) = \min_{i \neq i^\star}(\mu_{i^\star} - \mu_i)^2$, it is easy to see that $T^\star_{1/2}(\mu)/T^\star(\mu) = T^\star_{1/2}(\tilde{\mu})/T^\star(\tilde{\mu})$ when $\Delta_{\min}(\mu) = \Delta_{\min}(\tilde{\mu})$. Likewise, this ratio is invariant by translation of all means by a same quantity. Therefore, we consider without restriction an instance $\mu$ such that $\Delta_{\min}(\mu) = 1$ and with

$$\mu_1 = 0 > \mu_2 = -x_2 \geq \cdots \geq \mu_K = -x_K \,,$$

where $x_2 = 1$ and $x_i \geq 1$ for all $i \geq 3$.

First, we can rewrite

$$T^\star_{1/2}(x)^{-1} = \frac{1}{4} \max_{q \in \triangle_{K-1}} \min_{i \geq 2} \frac{x_i^2}{1 + 1/q_i} \geq \frac{1}{4} \max_{q \in \triangle_{K-1}} \min_{i \geq 2} \frac{1}{1 + 1/q_i} = \frac{1}{4K} \,,$$

where the inequality is an equality if and only $x_i = 1$ for all $i \geq 2$. Using Lemma C.4, we know that

$$T^\star(x) = 2(1 + \sqrt{K-1})^2$$

if and only $x_i = 1$ for all $i \geq 2$. Therefore, we have exhibited an instance such that

$$\frac{T^\star_{1/2}(\mu)}{T^\star(\mu)} = \frac{2K}{(1 + \sqrt{K-1})^2} \, .$$

Applying Lemma C.5 for $\beta = 1/2$, we obtain $T^\star_{1/2}(x) = 4C(x)$ where $C(x)$ is the implicit solution of the equation

$$\varphi(x, C(x)) = 0 \quad \text{where} \quad \varphi(x, C) = \frac{1}{C-1} + \sum_{i \geq 3} \frac{1}{Cx_i^2 - 1} - 1 \, .$$

Using Lemma C.3, we know that

$$T^\star(x) = \frac{2r(x)}{1 + \frac{1}{r(x)-1} + \sum_{i \geq 3} \frac{1}{r(x)x_i^2 - 1}}$$

where $r(x)$ is the implicit solution of the equation

$$\psi(x, r(x)) = 0 \quad \text{where} \quad \psi(x, r) = \frac{1}{(r-1)^2} + \sum_{i \geq 3} \frac{1}{(rx_i^2 - 1)^2} - 1 \, .$$

Therefore, we obtain that

$$\Omega(x) = \frac{1}{2} \frac{T^\star_{1/2}(x)}{T^\star(x)} = \frac{C(x)}{r(x) - 1} + \sum_{i \geq 3} \frac{C(x)}{r(x)(r(x)x_i^2 - 1)} \, ,$$

and our goal is to show that the above quantity is maximum if and only $x_i = 1$ for all $i \geq 2$. One way of doing this is by computing the gradient and showing it is negative, which would imply that the function is decreasing. Let $j \geq 3$. Using the implicit differentiation theorem, we obtain

$$\frac{\partial C}{\partial x_j}(x) = -\frac{\frac{\partial \varphi}{\partial x_j}(x, C(x))}{\frac{\partial \varphi}{\partial C}(x, C(x))} = -\frac{\frac{2C(x)x_j}{(C(x)x_j^2 - 1)^2}}{\frac{1}{(C(x)-1)^2} + \sum_{i \geq 3} \frac{x_i^2}{(C(x)x_i^2 - 1)^2}} \quad \text{and} \quad \frac{\partial C}{\partial x_j}(1_{K-2}) = -\frac{2C(1_{K-2})}{K-1} \, ,$$

and

$$\frac{\partial r}{\partial x_j}(x) = -\frac{\frac{\partial \psi}{\partial x_j}(x, r(x))}{\frac{\partial \psi}{\partial r}(x, r(x))} = -\frac{\frac{2r(x)x_j}{(r(x)x_j^2 - 1)^3}}{\frac{1}{(r(x)-1)^3} + \sum_{i \geq 3} \frac{x_i^2}{(r(x)x_i^2 - 1)^3}} \quad \text{and} \quad \frac{\partial r}{\partial x_j}(1_{K-2}) = -\frac{2r(1_{K-2})}{K-1} \, .$$

Direct computations yield that

$$\frac{\partial}{\partial x_j}\left(\frac{C(x)}{r(x)-1}\right) = \frac{\frac{\partial C}{\partial x_j}(x)(r(x)-1) - C(x)\frac{\partial r}{\partial x_j}(x)}{(r(x)-1)^2} \quad \text{and} \quad \frac{\partial}{\partial x_j}\left(\frac{C(x)}{r(x)-1}\right)_{x=1_{K-2}} = \frac{1}{K-1}\frac{2C(1_{K-2})}{(r(1_{K-2})-1)^2} \, ,$$

and

$$\frac{\partial}{\partial x_j}\left(\frac{C(x)}{r(x)(r(x)x_j^2 - 1)}\right) = \frac{\frac{\partial C}{\partial x_j}(x)r(x)(r(x)x_j^2 - 1) - C(x)\frac{\partial r}{\partial x_j}(x)\left(2r(x)x_j^2 - 1\right) - 2C(x)r(x)^2 x_j}{\left(r(x)(r(x)x_j^2 - 1)\right)^2}$$

$$\frac{\partial}{\partial x_j}\left(\frac{C(x)}{r(x)(r(x)x_j^2 - 1)}\right)_{x=1_{K-2}} = \left(\frac{1}{K-1} - 1\right)\frac{2C(1_{K-2})}{(r(1_{K-2})-1)^2} \, ,$$

and, for $i \geq 3$ s.t. $i \neq j$,

$$\frac{\partial}{\partial x_j}\left(\frac{C(x)}{r(x)(r(x)x_i^2 - 1)}\right) = \frac{\frac{\partial C}{\partial x_j}(x)r(x)(r(x)x_i^2 - 1) - C(x)\frac{\partial r}{\partial x_j}(x)\left(2r(x)x_i^2 - 1\right)}{\left(r(x)(r(x)x_i^2 - 1)\right)^2}$$

$$\frac{\partial}{\partial x_j}\left(\frac{C(x)}{r(x)(r(x)x_i^2 - 1)}\right)_{x=1_{K-2}} = \frac{1}{K-1}\frac{2C(1_{K-2})}{(r(1_{K-2})-1)^2} \, .$$

Then, plugging everything together, we obtained that

$$\frac{\partial \Omega}{\partial x_j}(1_{K-2}) = \frac{2C(1_{K-2})}{(r(1_{K-2})-1)^2}\left(\frac{1}{K-1} + \frac{1}{K-1} - 1 + \sum_{i\geq 3, i\neq j}\frac{1}{K-1}\right) = 0\,.$$

Therefore, we have shown that $\nabla\Omega(1_{K-2}) = 0_{K-2}$. $\qquad\square$

## D  Non-asymptotic analysis

In Appendix D.1, we state and prove one key result for each one of the three main components of the TTUCB sampling rule: the UCB leader (Lemma D.1), the TC challenger (Lemma D.2) and the tracking (Lemma D.3). The proof of Theorem 2.4 is detailed in Appendix D.2, which uses the stopping rule (1) and a proof method from [30]. It is a direct consequence of a more general result (Theorem D.4). In Appendix D.3, we prove a non-asymptotic upper bound for the TTUCB when using $g_m$ instead of $g_u$ (Corollary D.5). We compare our results with uniform sampling in Appendix D.4 (Theorem D.6). Other technicalities are gathered in Appendix D.5.

### D.1  Key properties

Before delving in the proof of Theorem D.4 itself, we present the key properties of each component of the TTUCB sampling rule under a some concentration event.

Let $\alpha > 1$ and $s > 1$. Let $(\mathcal{E}_n)_{n>K}$ be the sequence of concentration events defined as $\mathcal{E}_n := \mathcal{E}_{1,n} \cap \mathcal{E}_{2,n}$ for all $n > K$ where $\mathcal{E}_{1,n}$ and $\mathcal{E}_{2,n}$ are defined in (17) and (19) as

$$\mathcal{E}_{1,n} := \left\{\forall k \in [K], \forall t \in [n^{1/\alpha}, n], \ |\mu_{t,k} - \mu_k| < \sqrt{\frac{g_u(t)}{N_{t,k}}}\right\}\,,$$

$$\mathcal{E}_{2,n} := \left\{\forall k \neq i^\star, \forall t \in [n^{1/\alpha}, n], \ (\mu_{t,i^\star} - \mu_{t,k}) - (\mu_{i^\star} - \mu_k) > -\sqrt{2\alpha(2+s)\log(t)\left(\frac{1}{N_{t,i^\star}} + \frac{1}{N_{t,k}}\right)}\right\}\,,$$

with $g_u(n) = 2\alpha(1+s)\log n$. In Lemma E.6, it is shown that $\sum_{n>K}\mathbb{P}(\mathcal{E}_n^\complement) \leq (2K-1)\zeta(s)$.

**UCB leader**  Lemma D.1 shows that the UCB leader is different from $i^\star$ for only a sublinear number of times under a certain concentration event. It is slightly more general than Lemma 3.2 presented in Section 2, which follows from $H_1(\mu) \leq H_1(\mu) + 2\Delta_{\min}^{-2} = H(\mu)$.

**Lemma D.1.** *Let $(\mathcal{E}_{1,n})_n$ and $g_u$ as in Lemma E.2. Let $H_1(\mu) = \sum_{i\neq i^\star(\mu)}\frac{2}{(\mu_{i^\star(\mu)}-\mu_i)^2}$. For all $n > K$, under the event $\mathcal{E}_{1,n}$,*

$$\forall t \in [n^{1/\alpha}, n], \quad L_{t,i^\star} \geq t - 1 - 2H(\mu)g_u(t)/\beta - K/\beta\,, \tag{6}$$

*Let $(\mathcal{E}_{3,n})_n$ and $g_m$ as in Lemma E.3. Under the event $\mathcal{E}_{3,n}$, (6) holds by using $g_m$ instead of $g_u$.*

*Proof.* Suppose that at time $t \in [n^{1/\alpha}, n]$, the UCB leader is different from $i^\star$, i.e. $B_t = k \neq i^\star$. Using the event $\mathcal{E}_{1,n}$ and the definition of $B_t$ yields

$$\mu_{i^\star} \leq \mu_{t,i^\star} + \sqrt{\frac{g_u(t)}{N_{t,i^\star}}} \leq \mu_{t,k} + \sqrt{\frac{g_u(t)}{N_{t,k}}} \leq \mu_k + \sqrt{\frac{4g_u(t)}{N_{t,k}}}\,.$$

We get that if $t \geq n^{1/\alpha}$, then $N_{t,k} \leq \frac{4g_u(t)}{(\mu_{i^\star}-\mu_k)^2}$. Therefore, we obtain the following upper bound on the number of times the leader is different from $i^\star$ up to time $t$

$$t - 1 - L_{t,i^\star} = \sum_{k\neq i^\star}L_{t,k} \leq \frac{1}{\beta}\sum_{k\neq i^\star}N_{t,k}^k + \frac{K-1}{2\beta} \leq \frac{1}{\beta}\sum_{k\neq i^\star}N_{t,k} + \frac{K-1}{2\beta} \leq \sum_{k\neq i^\star}\frac{4g_u(t)}{(\mu_{i^\star}-\mu_k)^2\beta} + \frac{K-1}{2\beta}\,,$$

where we used Lemma D.3 for the second inequality and $N_{t,k}^k \leq N_{t,k}$ for the third. This concludes the proof for $g_u$. The same reasoning can be applied for $g_m$. $\qquad\square$

**TC challenger** Lemma D.2 shows a lower bound on the "transportation" costs used by the TC challenger provided a certain concentration holds. This lower bound depends only on the empirical counts when the best arm is the leader.

**Lemma D.2.** *For all $n > K$, under the event $\mathcal{E}_{2,n}$, for all $t \in [n^{1/\alpha}, n]$ such that $B_t = i^\star$,*

$$\forall k \neq i^\star, \quad \frac{\mu_{t,B_t} - \mu_{t,k}}{\sqrt{1/N_{t,B_t} + 1/N_{t,k}}} \geq \frac{\mu_{i^\star} - \mu_k}{\sqrt{1/N_{t,i^\star}^{i^\star} + 1/N_{t,k}^{i^\star}}} - \sqrt{2\alpha(2+s)\log n} \,.$$

*Proof.* Under $\mathcal{E}_{2,n}$, using $B_t = i^\star$ yields

$$\frac{\mu_{t,B_t} - \mu_{t,k}}{\sqrt{1/N_{t,B_t} + 1/N_{t,k}}} \geq \frac{\mu_{i^\star} - \mu_k}{\sqrt{1/N_{t,i^\star} + 1/N_{t,k}}} - \sqrt{2\alpha(2+s)\log t} \geq \frac{\mu_{i^\star} - \mu_k}{\sqrt{1/N_{t,i^\star}^{i^\star} + 1/N_{t,k}^{i^\star}}} - \sqrt{2\alpha(2+s)\log n} \,,$$

where the second inequality uses that $N_{t,k} \geq N_{t,k}^{i^\star}$ for all $k \neq i^\star$. $\qquad\square$

**Tracking** Lemma D.3 shows the key property satisfied by the $K$ independent tracking procedures used by the TTUCB sampling rule. It is slightly more general than Lemma 2.2 presented in Section 2. It is a simple corollary of Theorem 6 in [13].

**Lemma D.3.** *For all $n > K$ and $i \in [K]$, we have $-1/2 \leq N_{n,i}^i - \beta L_{n,i} \leq 1$ and for all $k \neq i$,*

$$N_{n,i}^i \geq \frac{\beta}{1-\beta} N_{n,k}^i - \frac{1}{2}\frac{1}{1-\beta} \,.$$

*Proof.* We can rewrite the tracking condition $N_{n,B_n}^{B_n} \leq \beta L_{n+1,B_n}$ as

$$N_{n,B_n}^{B_n} - \beta L_{n+1,B_n} \leq (L_{n+1,B_n} - N_{n,B_n}^{B_n}) - (1-\beta)L_{n+1,B_n} \,.$$

For all $k \in [K]$, this corresponds to a two-arms C-Tracking between the leader $k$ and the challengers with $w_n = (\beta, 1-\beta)$ for all $n$ such that $B_n = k$. The leader's pulling count is $N_{n,B_n}^{B_n}$ and the challengers' pulling count is $L_{n+1,B_n} - N_{n,B_n}^{B_n}$. We recall that C-Tracking was defined as $I_n = \arg\min_{k \in [K]} N_{n,k} - \sum_{t \in [n]} w_{t,k}$.

Theorem 6 in [13] yields for all $n > K$, $-\frac{1}{2} \leq N_{n,B_n}^{B_n} - \beta L_{n,B_n} \leq 1$. The $K$ parallel tracking procedures are independent since they are considering counts partitioned on the considered leader. Therefore, the above results holds for all $i \in [K]$. This concludes the first part of the proof.

Direct manipulations yield the second part of the result, namely

$$N_{n,k}^i \leq L_{n,i} - N_{n,i}^i \leq (1-\beta)L_{n,i} + \frac{1}{2} \quad \text{and} \quad N_{n,i}^i \geq -\frac{1}{2} + L_{n,i}\beta \geq \frac{\beta}{1-\beta}\left(N_{n,k}^i - \frac{1}{2}\right) - \frac{1}{2} \,.$$

$\qquad\square$

The choice of $K$ independent tracking procedures was made for two reasons. First, independent procedures are simpler to analyze for theoretical purpose. Second, independent procedures yields better empirical performance since it avoids over-sampling a sub-optimal arm when it is mistakenly chosen as leader. To understand the second argument, let's look at another design with one tracking procedure. Namely we set $I_n = B_n$ if $N_{n,B_n} \leq \beta n$, else $I_n = C_n$. When $B_n \neq i^\star$, then it will (almost always) take $I_n = B_n$ since $N_{n,B_n}$ is lower than $\beta n$. On the other hand, when $B_n \neq i^\star$ with $K$ independent tracking procedures, both $N_{n,B_n}^{B_n}$ and $L_{n,B_n}$ are small, hence it is less systematic.

### D.2 Proof of Theorem 2.4

Theorem 2.4 is a direct corollary of Theorem D.4, which holds for any $\beta \in (0,1)$, $s > 1$ and $\alpha > 1$.

**Theorem D.4.** *Let $(\delta, \beta) \in (0,1)^2$, $s > 1$ and $\alpha > 1$. Using the threshold (2) in (1) and $g_u$ in (3), the TTUCB algorithm satisfies that, for all $\mu \in \mathbb{R}^K$ such that $|i^\star(\mu)| = 1$,*

$$\mathbb{E}_\mu[\tau_\delta] \leq \max\left\{T_0(\delta), C_\mu^\alpha, C_0^{\frac{\alpha}{\alpha-1}}, C_1^\alpha\right\} + C_2 \,,$$

*where*

$$T_0(\delta) = \sup\left\{n > K \mid n - 1 \le T_\beta^\star(\mu)\frac{(1+\varepsilon)^2}{\beta(1-w_0)^{d_\mu(w_0)}}(\sqrt{c(n-1,\delta)} + \sqrt{\alpha(2+s)\log n})^2\right\},$$

$$C_\mu = h_1\left(4\alpha^2(1+s)H(\mu)/\beta\right), \quad C_0 = \frac{2}{\varepsilon a_\mu(w_0)} + 1, \quad C_1 = 1/(\beta\varepsilon), \quad C_2 = (2K-1)\zeta(s) + 1,$$

$$a_\mu(w_0) = (1-w_0)^{d_\mu(w_0)}\max\{\min_{i\neq i^\star(\mu)} w_\beta^\star(\mu)_i, (1-\beta)w_0\}, \quad d_\mu(w_0) = |\{i \in [K] \setminus \{i^\star(\mu)\} \mid w_\beta^\star(\mu)_i < (1-\beta)w_0\}|,$$

*with* $\varepsilon \in (0,1]$, $w_0 \in [0, 1/(K-1)]$ *and* $\zeta$ *is the Riemann* $\zeta$ *function. For all* $x > 0$, *the function* $h_1(x) := x\overline{W}_{-1}\left(\log(x) + \frac{2+K/\beta}{x}\right)$ *is positive, increasing for* $x \ge 2 + K/\beta$, *and satisfies* $h_1(x) \approx x(\log x + \log\log x)$.

Let $n > K$ such that $\mathcal{E}_n$ holds true and the algorithm has not stop yet, i.e. $\mathcal{E}_n \cap \{n < \tau_\delta\}$. Let $t \in [n^{1/\alpha}, n]$ such that $B_t = i^\star$. Let $c(n,\delta)$ as in (2), which satisfies that $n \mapsto c(n,\delta)$ is increasing. Using the stopping rule (1) and $t \le n < \tau_\delta$, we obtain

$$\sqrt{2c(n-1,\delta)} \ge \sqrt{2c(t-1,\delta)} \ge \min_{i\neq\hat{\imath}_t} \frac{\mu_{t,\hat{\imath}_t} - \mu_{t,i}}{\sqrt{1/N_{t,\hat{\imath}_t} + 1/N_{t,i}}} \ge \min_{i\neq B_t} \frac{(\mu_{t,B_t} - \mu_{t,i})_+}{\sqrt{1/N_{t,B_t} + 1/N_{t,i}}} = \frac{(\mu_{t,B_t} - \mu_{t,C_t})_+}{\sqrt{1/N_{t,B_t} + 1/N_{t,C_t}}},$$

The last inequality is an equality when $B_t = \hat{\imath}_t$, and trivially true when $B_t \neq \hat{\imath}_t$ since a positive term is higher than zero (already null when taking $i = \imath_t$). Using Lemma D.2, we obtain

$$\frac{\mu_{t,B_t} - \mu_{t,C_t}}{\sqrt{1/N_{t,B_t} + 1/N_{t,C_t}}} \ge \frac{\mu_{i^\star} - \mu_{C_t}}{\sqrt{1/N_{t,i^\star}^{i^\star} + 1/N_{t,C_t}^{i^\star}}} - \sqrt{2\alpha(2+s)\log n}$$

$$\ge \sqrt{\frac{1/\beta + 1/w_{\beta,C_t}^\star}{1/N_{t,i^\star}^{i^\star} + 1/N_{t,C_t}^{i^\star}}} \min_{i\neq i^\star} \frac{\mu_{i^\star} - \mu_i}{\sqrt{1/\beta + 1/w_{\beta,i}^\star}} - \sqrt{2\alpha(2+s)\log n}$$

$$\ge \sqrt{\frac{1/\beta + 1/w_{\beta,C_t}^\star}{1/N_{t,i^\star}^{i^\star} + 1/N_{t,C_t}^{i^\star}}} \sqrt{2T_\beta^\star(\mu)^{-1}} - \sqrt{2\alpha(2+s)\log n},$$

where the second inequality is obtained by artificially making appear $1/\beta + 1/w_{\beta,C_t}^\star$ and taking the minimum of $i \neq i^\star$. The last inequality simply uses the definition of $w_\beta^{i^\star}$.

While combining Lemma D.1 and Lemma D.3 links $N_{t,i^\star}^{i^\star}/(t-1)$ with $\beta$, we need another argument to compare the empirical allocation $N_{t,C_t}^{i^\star}/(t-1)$ of the sub-optimal arm $C_t$ with its $\beta$-optimal allocation $w_{\beta,C_t}^\star$. Before delving into this key argument, we conclude the proof under an assumption that will be shown to hold later: there exists $D_0 > 0$ and $T_\mu > 0$ such that for all $n > T_\mu$, there exists a well chosen $t \in [n^{1/\alpha}, n]$ with $B_t = i^\star$, which satisfies

$$1/N_{t,i^\star}^{i^\star} + 1/N_{t,C_t}^{i^\star} \le \frac{D_0}{n-1}\left(1/\beta + 1/w_{\beta,C_t}^\star\right). \tag{7}$$

Let's define

$$T(\delta, D_0) := \sup\left\{n > K \mid n - 1 \le T_\beta^\star(\mu)D_0\left(\sqrt{c(n-1,\delta)} + \sqrt{\alpha(2+s)\log n}\right)^2\right\}.$$

For all $n > \max\{T_\mu, T(1, D_0)\}$, the lower bound on $\frac{\mu_{t,B_t} - \mu_{t,C_t}}{\sqrt{1/N_{t,B_t} + 1/N_{t,C_t}}}$ is positive, hence we can use that it is upper bounded by $\sqrt{2c(n-1,\delta)}$. Putting everything together, we have shown that

$$\forall n > \max\{T_\mu, T(1, D_0)\}, \quad n - 1 \le T_\beta^\star(\mu)D_0\left(\sqrt{c(n-1,\delta)} + \sqrt{\alpha(2+s)\log n}\right)^2.$$

Therefore, we have $\mathcal{E}_n \cap \{n < \tau_\delta\} = \emptyset$ (i.e. $\mathcal{E}_n \subset \{\tau_\delta \le n\}$) for all $n \ge \max\{T(\delta, D_0), T(1, D_0), T_\mu\} + 1$. Using that $\delta \to T(\delta, D_0)$ is an decreasing function (since $\mathcal{C}_G$ is increasing), we obtain that $T(\delta, D_0) = \max\{T(\delta, D_0), T(1, D_0)\}$.

Combining Lemmas D.8 and E.6 yields

$$\mathbb{E}_\mu[\tau_\delta] \le \max\{T(\delta, D_0), T_\mu\} + 1 + \sum_{n \ge 1} \mathbb{P}_\mu(\mathcal{E}_n^{\complement}) \le \max\{T(\delta, D_0), T_\mu\} + 1 + (2K-1)\zeta(s) \,.$$

At this stage, the proof of Theorem D.4 boils down to exhibiting $D_0 > 0$ and $T_\mu > 0$ such that: for all $n > T_\mu$, there exists a well chosen $t \in [n^{1/\alpha}, n]$ with $B_t = i^\star$ and such that (7) holds. As mentioned above, the crux of the problem is to relate $N_{t,C_t}^{i^\star}/(t-1)$ and $w_{\beta,C_t}^\star$. To do so, we will build on the idea behind the proof for APT from [30]: consider an arm being over-sampled and study the last time this arm was pulled.

By the pigeonhole principle, at time $n$, there is an index $k_1 \ne i^\star$ such that (5) holds, i.e.

$$N_{n,k_1}^{i^\star} \ge \frac{w_{\beta,k_1}^\star}{1 - \beta}(L_{n,i^\star} - N_{n,i^\star}^{i^\star}) \,,$$

and we take such $k_1$. Let $t_1 := \sup\{t < n \mid (B_t, C_t) = (i^\star, k_1)\}$ be the last time at which $i^\star$ was the leader and $k_1$ was the challenger. If $I_{t_1} = B_{t_1}$ then $N_{t_1,k_1}^{i^\star} = N_{n,k_1}^{i^\star}$, else $N_{t_1,k_1}^{i^\star} = N_{n,k_1}^{i^\star} - 1$. In both cases, we have $N_{t_1,k_1}^{i^\star} \ge N_{n,k_1}^{i^\star} - 1$. Let $f_1$ as in (14). Combined the above with Lemma D.9, we obtain

$$N_{t_1,k_1}^{i^\star} \ge N_{n,k_1}^{i^\star} - 1 \ge \frac{w_{\beta,k_1}^\star}{1 - \beta}(L_{n,i^\star} - N_{n,i^\star}^{i^\star}) - 1 \ge w_{\beta,k_1}^\star(n-1) - f_1(n) \,.$$

Let $w_- > 0$ be a lower bound on $w_{\beta,k_1}^\star$, for example consider $w_- = \min_{i \ne i^\star} w_{\beta,i}^\star$. Let

$$C_0(w_-) = \sup\left\{n \ge 1 \mid n - 1 < \frac{1}{w_-}\left(n^{1/\alpha} + f_1(n)\right)\right\} \,. \tag{8}$$

Since $t_1 - 1 \ge N_{t_1,k_1}^{i^\star} \ge w_{\beta,k_1}^\star(n-1) - f_1(n)$, we obtain that $t_1 \ge n^{1/\alpha}$ for all $n > N_0(w_-)$. For instances $\mu$ such that $w_{\beta,k_1}^\star$ is small, the equation (5) can be satisfied at the very beginning, hence $t_1$ might be sublinear in $n$. Therefore, while combining Lemma D.1 and Lemma D.3 yields $N_{t_1,i^\star}^{i^\star} \gtrsim \beta(t_1 - 1)$, it is not possible to obtain $N_{t_1,i^\star}^{i^\star} \gtrsim \beta(n-1)$. Due to the missing link between $t_1$ and $n$, we use the following inequality

$$1/N_{t_1,i^\star}^{i^\star} + 1/N_{t_1,k_1}^{i^\star} \le \frac{1}{n-1}\left(1/\beta + \frac{n-1}{N_{t_1,k_1}^{i^\star}}\right)\left(\frac{N_{t_1,k_1}^{i^\star}}{N_{t_1,i^\star}^{i^\star}} + 1\right) \,,$$

which is a suboptimal step which artificially introduces $1/\beta$, and is responsible for the multiplicative factor $1/\beta$ in Theorem D.4. Improving on this suboptimal step is an interesting question, whose answer still eludes us. One idea would be to leverage the information of the sampled arm at time $t$ since we have $N_{t,i^\star}^{i^\star} \le \beta L_{t+1,i^\star}$ when $i^\star = B_t = I_t$, else $N_{t,i^\star}^{i^\star} \ge \beta L_{t+1,i^\star}$.

Let $\varepsilon \in (0, 1]$. It remains to control both terms. First, we obtain

$$1/\beta + \frac{n-1}{N_{t_1,k_1}^{i^\star}} \le 1/\beta + \frac{1}{w_{\beta,k_1}^\star - f_1(n)/(n-1)} \le (1 + \varepsilon)\left(1/\beta + 1/w_{\beta,k_1}^\star\right) \,,$$

for all $n > C_1(w_-)$. The last inequality is obtained by definition of

$$C_1(w_-) = \sup\left\{n \ge 1 \mid n - 1 < \frac{f_1(n)}{w_-}\left(1 + \frac{1}{\varepsilon}\right)\right\} \,, \tag{9}$$

which ensures that, for all $n > C_1(w_-)$, the last condition of the equivalence

$$w_{\beta,k_1}^\star - f_1(n)/(n-1) \ge (1+\varepsilon)^{-1} w_{\beta,k_1}^\star \quad\Longleftrightarrow\quad n - 1 \ge \frac{f_1(n)}{w_{\beta,k_1}^\star}\left(1 + \frac{1}{\varepsilon}\right)$$

is satisfied since $w_{\beta,k_1}^\star \ge w_-$ and $n > C_1(w_-)$. Second, using Lemma D.3 with $N_{t_1,k_1}^{i^\star} \ge w_{\beta,k_1}^\star(n-1) - f_1(n)$, we obtain

$$\frac{N_{t_1,k_1}^{i^\star}}{N_{t_1,i^\star}^{i^\star}} + 1 \le \left(\frac{\beta}{1-\beta} - \frac{1}{2(1-\beta)\left(w_{\beta,k_1}^\star(n-1) - f_1(n)\right)}\right)^{-1} + 1 \le (1+\varepsilon)/\beta \,,$$

for all $n > C_2(w_-)$. The last inequality is obtained by definition of

$$C_2(w_-) := \sup \left\{ n \in \mathbb{N}^\star \mid n - 1 < \frac{1}{w_-} \left( f_1(n) + \frac{1 - \beta + \varepsilon}{2\beta\varepsilon} \right) \right\}, \tag{10}$$

which ensures that, for all $n > C_2(w_-)$, the last condition of the equivalence

$$\left( \frac{\beta}{1 - \beta} - \frac{1}{2(1 - \beta)\left( w_{\beta,k_1}^\star(n - 1) - f_1(n) \right)} \right)^{-1} + 1 \leq (1 + \varepsilon)/\beta$$

$$\iff \frac{1}{w_{\beta,k_1}^\star(n - 1) - f_1(n)} \leq \frac{2\beta\varepsilon}{1 - \beta + \varepsilon} \iff n - 1 \geq \frac{1}{w_{\beta,k_1}^\star} \left( f_1(n) + \frac{1 - \beta + \varepsilon}{2\beta\varepsilon} \right)$$

is satisfied since $w_{\beta,k_1}^\star \geq w_-$ and $n > C_2(w_-)$. By comparison between (8) and (10), we notice that

$$\max \left\{ C_0(w_-), C_2(w_-) \right\} \leq \max \left\{ C_0(w_-), \left( \frac{1 - \beta + \varepsilon}{2\beta\varepsilon} \right)^\alpha \right\}$$

Putting everything together, we have shown that taking $D_0 = (1 + \varepsilon)^2/\beta$, we have for all $n > \max\{K, C_0(w_-), C_1(w_-), C_2(w_-)\}$, there exists $t_1 \in [n^{1/\alpha}, n]$ with $B_{t_1} = i^\star$ and such that (7) holds. Let $h_1$ defined in (15). Since $\varepsilon \leq 1$, using Lemmas D.10 and D.11 with the above yields

$$\max\{C_0(w_-), C_1(w_-), C_2(w_-)\} \leq \max \left\{ h_1 \left( 4\alpha^2(1 + s)H(\mu) \right), \left( \frac{2}{\varepsilon w_-} + 1 \right)^{1/(\alpha - 1)}, \frac{1}{\beta\varepsilon} \right\}^\alpha.$$

Using $w_- = \min_{i \neq i^\star} w_{\beta,i}^\star$ yields the first part of Theorem D.4, i.e. the special case of $w_0 = 0$.

**Refined non-asymptotic upper bound**   When considering large $K$ or instances with unbalanced $\beta$-optimal allocation, $\min_{i \neq i^\star} w_{\beta,i}^\star$ can become arbitrarily small. Therefore, the dependency in the inverse of $\min_{i \neq i^\star} w_{\beta,i}^\star$ is undesired, and we would like to clip it with a value of our choosing which is away from zero.

Let $w_0 \in (0, 1/(K - 1)]$ be an allocation threshold and $d_\mu(w_0) := |\{i \in [K] \setminus \{i^\star\} \mid w_{\beta,i}^\star < (1 - \beta)w_0\}|$ be the number of arms having a $\beta$-optimal allocation strictly smaller than $(1 - \beta)w_0$. As discussed above, for instances $\mu$ such that $w_{\beta,k_1}^\star$ (defined above) is small, the equation (5) can be satisfied at the very beginning for a small empirical allocation $N_{n,k_1}^{i^\star}$. To provide a more meaningful result, one needs to have a sub-optimal arm $k_1$ such that either:

- **Case 1:** $w_{\beta,k_1}^\star$ is not too small, i.e. $w_{\beta,k_1}^\star \geq (1 - \beta)w_0$.

- **Case 2:** $w_{\beta,k_1}^\star$ is too small but $N_{n,k_1}^{i^\star}$ is large enough, i.e. $w_{\beta,k_1}^\star < (1 - \beta)w_0$ and $N_{n,k_1}^{i^\star} \geq w_0(L_{n,i^\star} - N_{n,i^\star}^{i^\star})$.

In case 1, we can conduct the same manipulations as above simply by using $w_- = \max\{(1 - \beta)w_0, \min_{i \neq i^\star} w_{\beta,i}^\star\}$ instead of $w_- = \min_{i \neq i^\star} w_{\beta,i}^\star$, since it is a lower bound for $w_{\beta,k_1}^\star$.

In case 2, the above proof can also be applied by conducting the same algebraic manipulations with $(1 - \beta)w_0$ instead of $w_{\beta,k_1}^\star$, and using $w_- = (1 - \beta)w_0 = \max\{(1 - \beta)w_0, \min_{i \neq i^\star} w_{\beta,i}^\star\}$. To slightly detail the argument, we can show similarly that $N_{t_1,k_1}^{i^\star} \geq (1 - \beta)w_0(n - 1) - f_1(n)$, where $f_1$ as in (14) since we have $(1 - \beta)w_0 < 1 - \beta$. Then, for all $n > C_1((1 - \beta)w_0)$,

$$1/\beta + \frac{n - 1}{N_{t_1,k_1}^{i^\star}} \leq (1 + \varepsilon) \left( 1/\beta + \frac{1}{(1 - \beta)w_0} \right) \leq (1 + \varepsilon) \left( 1/\beta + 1/w_{\beta,k_1}^\star \right).$$

The problematic situation happens when we are neither in case 1 nor in case 2:

- **Case 3:** both $w_{\beta,k_1}^\star$ and $N_{n,k_1}^{i^\star}$ are too small, i.e. $w_{\beta,k_1}^\star < (1 - \beta)w_0$ and $N_{n,k_1}^{i^\star} < w_0(L_{n,i^\star} - N_{n,i^\star}^{i^\star})$.

In case 3 it is not possible to conclude the proof with the arm $k_1$ without paying the price of an inverse of $\min_{i \neq i^\star} w^\star_{\beta,i}$. To overcome this price, we need to find another arm $k$ such that either case 1 or case 2 happens. Since $N^{i^\star}_{n,k_1}$ and $w^\star_{\beta,k_1}$ are small, we will ignore arm $k_1$ and use the pigeonhole principle on all arm $i \in [K] \setminus \{i^\star, k_1\}$. As in (5), we obtain that there exists $k_2$ such that

$$N^{i^\star}_{n,k_2} \geq \frac{w^\star_{\beta,k_2}}{1 - \beta - w^\star_{\beta,k_1}}(L_{n,i^\star} - N^{i^\star}_{n,i^\star} - N^{i^\star}_{n,k_1}) \geq \frac{(1-w_0)w^\star_{\beta,k_2}}{1-\beta}(L_{n,i^\star} - N^{i^\star}_{n,i^\star}) \,,$$

where the last inequality is obtained by using $w^\star_{\beta,k_1} > 0$ and $N^{i^\star}_{n,k_1} < w_0(L_{n,i^\star} - N^{i^\star}_{n,i^\star})$. Based on $k_2$ the same dichotomy happens: either we can conclude the proof when we are in case 1 or 2 or we cannot since we are in case 3. If case 3 occurs also for $k_2$, i.e. $w^\star_{\beta,k_2} < (1-\beta)w_0$ and $N^{i^\star}_{n,k_2} < w_0(L_{n,i^\star} - N^{i^\star}_{n,i^\star} - N^{i^\star}_{n,k_1})$, we should also ignore it since it is non informative.

The main idea is then to peel off arms that are not informative, till we find an informative one. By induction, we construct a sequence $(k_a)_{a \in [d]}$ of such arms, where $k_d$ is the first arm for which either case 1 or case 2 holds. This means that for all $a \in [d-1]$, we have $w^\star_{\beta,k_a} < (1-\beta)w_0$ and

$$N^{i^\star}_{n,k_a} < w_0(L_{n,i^\star} - N^{i^\star}_{n,i^\star} - \sum_{b \in [a-1]} N^{i^\star}_{n,k_b}) \,.$$

The construction, which rely on the pigeonhole principle for $i \in [K] \setminus (\{i^\star\} \cup \{k_a\}_{a \in [d-1]})$, yields that $k_d$ satisfies

$$N^{i^\star}_{n,k_d} \geq \frac{w^\star_{\beta,k_d}}{1 - \beta - \sum_{a \in [d-1]} w^\star_{\beta,k_a}}(L_{n,i^\star} - N^{i^\star}_{n,i^\star} - \sum_{a \in [d-1]} N^{i^\star}_{n,k_a}) \geq \frac{(1-w_0)^{d-1}w^\star_{\beta,k_d}}{1-\beta}(L_{n,i^\star} - N^{i^\star}_{n,i^\star}) \,,$$

where the last inequality is obtained since $\sum_{a \in [d-1]} w^\star_{\beta,k_a} > 0$ and by a simple recurrence on the arms $\{k_a\}_{a \in [d-1]}$. Since the arm $k_d$ satisfies case 1 or case 2, we can conclude similarly as above. Let $t_d := \sup\{t < n \mid (B_t, C_t) = (i^\star, k_d)\}$.

When $w^\star_{\beta,k_d} \geq (1-\beta)w_0$, the above proof can also be applied by conducting the same algebraic manipulations with $(1-w_0)^{d-1}w^\star_{\beta,k_d}$, and using $w_- = (1-w_0)^{d-1}\max\{(1-\beta)w_0, \min_{i \neq i^\star} w^\star_{\beta,i}\}$. In more details, we can show that $N^{i^\star}_{t_d,k_d} \geq (1-w_0)^{d-1}w^\star_{\beta,k_d}(n-1) - f_1(n)$, where $f_1$ as in (14) since we have $(1-w_0)^{d-1}w^\star_{\beta,k_d} < 1 - \beta$. Then, for all $n > C_1(w_-)$,

$$1/\beta + \frac{n-1}{N^{i^\star}_{t_d,k_d}} \leq (1+\varepsilon)\left(1/\beta + \frac{1}{(1-w_0)^{d-1}w^\star_{\beta,k_d}}\right) \leq \frac{1+\varepsilon}{(1-w_0)^{d-1}}\left(1/\beta + 1/w^\star_{\beta,k_d}\right) \,.$$

This allow to conclude the result with $D_0 = \frac{(1+\varepsilon)^2}{\beta(1-w_0)^{d-1}}$, hence paying a multiplicative factor of $1/(1-w_0)^{d-1}$.

When $w^\star_{\beta,k_d} < (1-\beta)w_0$ and $N^{i^\star}_{n,k_a} \geq w_0(L_{n,i^\star} - N^{i^\star}_{n,i^\star} - \sum_{a \in [d-1]} N^{i^\star}_{n,k_a})$, we conclude similarly by manipulating $(1-w_0)^{d-1}(1-\beta)w_0$, and using $w_- = (1-w_0)^{d-1}(1-\beta)w_0 = (1-w_0)^{d-1}\max\{(1-\beta)w_0, \min_{i \neq i^\star} w^\star_{\beta,i}\}$. First, we have $N^{i^\star}_{t_d,k_d} \geq (1-w_0)^{d-1}w_0(n-1) - f_1(n)$, where $f_1$ as in (14) since we have $(1-\beta)(1-w_0)^{d-1}w_0 < 1 - \beta$. Then, for all $n > C_1(w_-)$,

$$1/\beta + \frac{n-1}{N^{i^\star}_{t_d,k_d}} \leq (1+\varepsilon)\left(1/\beta + \frac{1}{(1-w_0)^{d-1}(1-\beta)w_0}\right) \leq \frac{1+\varepsilon}{(1-w_0)^{d-1}}\left(1/\beta + 1/w^\star_{\beta,k_d}\right) \,.$$

This allow to conclude the result with $D_0 = \frac{(1+\varepsilon)^2}{\beta(1-w_0)^{d-1}}$.

To remove the dependency in the random variable $d$, we consider the worst case scenario where $\{k_a\}_{a \in [d-1]} = \{i \in [K] \setminus \{i^\star\} \mid w^\star_{\beta,i} < (1-\beta)w_0\}$, i.e. $d - 1 \leq d_\mu(w_0)$. In words, it means that we had to enumerate over all arms with small allocation, such that case 2 didn't hold, before finding an arm with large allocation, i.e. satisfying case 1.

In all the cases considered above, the parameters always satisfied $w_- \geq (1-w_0)^{d_\mu(w_0)}\max\{(1-\beta)w_0, \min_{i \neq i^\star} w^\star_{\beta,i}\}$ and $D_0 \leq \frac{(1+\varepsilon)^2}{\beta(1-w_0)^{d_\mu(w_0)}}$. This yields the second part of Theorem D.4, i.e. for $w_0 \in (0, 1/(K-1)]$.

## D.3 Tighter UCB leader

While Theorem 2.3 holds for the TTUCB sampling rule using $g_m$ and $g_u$, Theorem D.4 is formulated solely for $g_u$. Experiments highlight that using $g_u$ leads to worse performances than when using $g_m$. This is not surprising since $g_m$ is smaller than $g_u$.

It is direct to see that the proof of Theorem D.4 also holds for $g_m$ up to additional technicalities which we detail below. As the obtained non-asymptotic upper bound is less explicit, we chose not to include it in Theorem 2.3.

Let $\alpha > 1$ and $s > 1$. Let $(\tilde{\mathcal{E}}_n)_n$ be the sequence of concentration events defined as $\tilde{\mathcal{E}}_n := \mathcal{E}_{3,n} \cap \mathcal{E}_{2,n}$ for all $n > K$ where $\mathcal{E}_{3,n}$ is defined in (18) as

$$\mathcal{E}_{3,n} := \left\{ \forall k \in [K], \forall t \in [n^{1/\alpha}, n], \ |\mu_{t,k} - \mu_k| < \sqrt{\frac{g_m(t)}{N_{t,k}}} \right\},$$

$$g_m(n) := \overline{W}_{-1} \left( 2s\alpha \log(n) + 2\log(2 + \alpha \log n) + 2 \right),$$

where $\overline{W}_{-1}(x) = -W_{-1}(-e^{-x})$ for all $x \geq 1$, with $W_{-1}$ is the negative branch of the Lambert $W$ function.

Let $n > K$ such that $\tilde{\mathcal{E}}_n$ holds true and the algorithm has not stop yet, i.e. $\tilde{\mathcal{E}}_n \cap \{n < \tau_\delta\}$. Using the second part of Lemma D.1, the vast majority of the proof is unchanged. Modifying the definition of $f_1$ in (14) of Lemma D.9 to account for $g_m$, we define $f_2(n) := 2H(\mu)g_m(n)/\beta + K/\beta + 2$ for all $n > K$.

In light of the proofs of the technical Lemmas D.10 and D.11, we can define

$$\tilde{C}_\mu = \sup \left\{ x \in \mathbb{N}^\star \mid x < f_2(x^\alpha) \right\},$$

and obtain directly Corollary D.5 with the same proof as in Appendix D.2.

**Corollary D.5.** *Let $(\delta, \beta) \in (0,1)^2$, $\varepsilon \in (0,1]$, $s > 1$, $\alpha > 1$, $w_0 \in [0, 1/(K-1)]$. Combining the stopping rule (1) with threshold (2) and the TTUCB sampling rule using $g_u$ in (3) yields a $\delta$-correct algorithm such that, for all $\mu \in \mathbb{R}^K$ with $|i^\star(\mu)| = 1$,*

$$\mathbb{E}_\mu[\tau_\delta] \leq \max \left\{ T_0(\delta), \tilde{C}_\mu^\alpha, C_0^{\frac{\alpha}{\alpha-1}}, C_1^\alpha \right\} + C_2,$$

*where $T_0(\delta)$, $C_0$, $C_1$ and $C_2$ are defined in Theorem 2.3, and*

$$\tilde{C}_\mu := \sup \left\{ x \in \mathbb{N}^\star \mid x < 2H(\mu)g_m(x^\alpha)/\beta + K/\beta + 2 \right\}. \tag{11}$$

**Explicit upper bound**  Since $g_m$ is itself a non-standard function (like logarithm), it is not straightforward to obtain an explicit upper bound on (11). For $g_u$, it was done in Lemma D.10 by using Lemma D.7.

Using Lemma D.7, we obtain that

$$x \geq 2H(\mu)g_m(x^\alpha)/\beta + K/\beta + 2$$

$$\iff \frac{x - c_0}{c_\mu} - \log\left(\frac{x - c_0}{c_\mu}\right) \geq 2s\alpha^2 \log(x) + 2\log(2 + \alpha^2 \log x) + 2$$

$$\impliedby y - \log(y) \geq \frac{1}{s\alpha^2 + 1/2}\left(\log(2 + \alpha^2 \log x) + \frac{c_0}{2c_\mu}\right) + \frac{s\alpha^2}{s\alpha^2 + 1/2}\log c_\mu + c_1$$

$$\iff x \geq h_2(c_\mu, x),$$

where we used $y = \frac{x}{c_\mu(2s\alpha^2+1)}$, $c_1 = \log(2s\alpha^2 + 1) + \frac{1}{s\alpha^2+1/2}$, $c_\mu = 2H(\mu)$, $c_0 = K/\beta + 2$ and define

$$h_2(z, x) := z(2s\alpha^2 + 1)\overline{W}_{-1}\left(\frac{1}{s\alpha^2 + 1/2}\left(s\alpha^2 \log z + \frac{c_0}{2z} + \log(2 + \alpha^2 \log x)\right) + c_1\right). \tag{12}$$

For both equivalences above, we used that we are interested in larger values of $x$, hence we used that $\frac{x-c_0}{c_\mu} \geq 1$, $\frac{x}{c_\mu(2s\alpha^2+1)} \geq 1$, $2s\alpha^2 \log(x) + 2\log(2 + \alpha^2 \log x) + 2 \geq 1$ and

$\frac{1}{s\alpha^2+1/2}\left(s\alpha^2\log c_\mu + \frac{c_0}{2c_\mu} + \log(2+\alpha^2\log x)\right) + c_1 \geq 1$. Since those conditions are implied by $x \geq h_2(c_\mu, x)$, those conditions are neither restrictive nor informative.

Therefore, we have shown that $\tilde{C}_\mu$ defined in (11) satisfies

$$\tilde{C}_\mu \leq \sup\{x \in \mathbb{N}^\star \mid x < h_2(2H(\mu), x)\},$$

where $h_2(z, x)$ is defined in (12) is such that $h_2(z, x) \approx_x 2z\log(2+\alpha^2\log x)$ and $h_2(z, x) \approx_z 2s\alpha^2 z\log z$.

### D.4 Uniform sampling

While the shortcomings of uniform sampling are well known for general bandit instances, it is also clear that uniform sampling perform well for highly symmetric instances $\mu$ such that $w^\star(\mu) \approx 1_K/K$. Therefore, we derive a non-asymptotic upper bound for uniform sampling (Theorem D.6), which allow a clear comparison with Theorem D.4 (and Corollary D.5).

**Theorem D.6.** *Let $\delta \in (0,1)$. Combining the stopping rule* (1) *with threshold* (2) *and the uniform sampling rule yields a $\delta$-correct algorithm such that, for all $\mu \in \mathbb{R}^K$ with $|i^\star(\mu)| = 1$,*

$$\mathbb{E}_\mu[\tau_\delta] \leq \max\left\{T_1(\delta), h_3\left(\frac{8\alpha K(1+s)}{\min_{i\neq i^\star}(\mu_{i^\star}-\mu_i)^2}\right)\right\} + 1 + (2K-1)\zeta(s), \tag{13}$$

*where $h_3(x) = x\overline{W}_{-1}(\log(x))$ for all $x \geq e$ and $h_3(x) = x$ for all $x \in (0, e)$ and*

$$T_1(\delta) = \sup\left\{n > K \mid n - 1 \leq \frac{4K}{\min_{i\neq i^\star}(\mu_{i^\star}-\mu_i)^2}\left(\sqrt{c(n-1,\delta)} + \sqrt{\alpha(2+s)\log n}\right)^2\right\}.$$

*Proof.* Let $n > K$ such that $\mathcal{E}_n$ holds true and the algorithm has not stop yet, i.e. $\mathcal{E}_n \cap \{n < \tau_\delta\}$. Let $t := \sup\{t \in [n^{1/\alpha}, n] \mid (t-1)/K \in \mathbb{N}, \hat{\imath}_t = i^\star\}$. Using the stopping rule (1) and $t \leq n < \tau_\delta$, we obtain

$$\sqrt{2c(n-1,\delta)} \geq \sqrt{2c(t-1,\delta)} \geq \min_{i\neq\hat{\imath}_t}\frac{\mu_{t,\hat{\imath}_t}-\mu_{t,i}}{\sqrt{1/N_{t,\hat{\imath}_t}+1/N_{t,i}}} = \frac{\mu_{t,\hat{\imath}_t}-\mu_{t,k_t}}{\sqrt{1/N_{t,\hat{\imath}_t}+1/N_{t,k_t}}},$$

where $k_t = \arg\min_{i\neq\hat{\imath}_t}\frac{\mu_{t,\hat{\imath}_t}-\mu_{t,i}}{\sqrt{1/N_{t,\hat{\imath}_t}+1/N_{t,i}}}$. Since we sample uniformly, for all time $t$ such that $(t-1)/K \in \mathbb{N}$, we have $N_{t,i} = (t-1)/K$ for all $i \in [K]$. Combining the above with Lemma D.2, we obtain

$$\sqrt{2c(n-1,\delta)} + \sqrt{2\alpha(2+s)\log n} \geq \frac{\mu_{i^\star}-\mu_{k_t}}{\sqrt{1/N_{t,i^\star}^{i^\star}+1/N_{t,k_t}^{i^\star}}} \geq \sqrt{\frac{t-1}{2K}}\min_{i\neq i^\star}(\mu_{i^\star}-\mu_i),$$

where the last inequality is obtained by taking the minimum over $i \in [K]$. To conclude the proof, we simply have to link $t$ with $n$. More precisely, we will show that $n - t \leq K - 1$ and $\hat{\imath}_t = i^\star$ for $n$ large enough. By concentration, we have

$$\mu_i - \sqrt{\frac{Kg_u(t)}{t-1}} = \mu_i - \sqrt{\frac{g_u(t)}{N_{t,i}}} \leq \mu_{t,i} \leq \mu_i + \sqrt{\frac{g_u(t)}{N_{t,i}}} = \mu_i + \sqrt{\frac{Kg_u(t)}{t-1}}.$$

Therefore, we have $\mu_{t,i^\star} > \max_{i\neq i^\star}\mu_{t,i}$, i.e. $\hat{\imath}_t = i^\star$, for all $t > N_3$ where

$$N_3 := \sup\left\{n \in \mathbb{N} \mid n-1 \leq \frac{4Kg_u(n)}{\min_{i\neq i^\star}(\mu_{i^\star}-\mu_i)^2}\right\},$$

which means that we have at worse $n-t \leq K-1$. For $n \geq N_3+K$, we obtain $t \geq n-(K-1) > N_3$.

Let $c_\mu = \frac{8\alpha K(1+s)}{\min_{i\neq i^\star}(\mu_{i^\star}-\mu_i)^2}$ and $x = n/c_\mu$. Assume that $c_\mu \geq e$. Using the definition of $g_u$ and Lemma D.7, direct manipulations yields

$$n > \frac{4Kg_u(n)}{\min_{i\neq i^\star}(\mu_{i^\star}-\mu_i)^2} \iff x - \log(x) > \log(c_\mu) \iff x > \overline{W}_{-1}(\log(c_\mu)),$$

where the last inequality uses that $\log(c_\mu) \geq 1$ and $x \geq 1$, which is not restrictive (or informative) since it is implied by $x > \overline{W}_{-1}(\log(c_\mu))$. When $c_\mu < e$, which means that the problem is easy, we have directly that $x - \log(x) > 1 > \log(c_\mu)$ for $x > 1$. Therefore, the condition becomes $x > 1$, i.e. $n \geq c_\mu$. Defining $h_3(x) = x\overline{W}_{-1}(\log(x))$ for all $x \geq e$ and $h_3(x) = x$ for all $x \in (0, e)$, we have

$$N_3 \leq h_3\left(\frac{8\alpha K(1+s)}{\min_{i \neq i^\star}(\mu_{i^\star} - \mu_i)^2}\right) .$$

Using a similar argument as the one in Appendix D.2 which rely on D.8 and the definition of $T_1(\delta)$, we can conclude the proof. □

The structure of the non-asymptotic upper bound of uniform sampling in (13) is similar to the one for the TTUCB sampling rule in Theorem D.4. Therefore, we can compare the dominating terms for both the asymptotic and the non-asymptotic regime.

First, we look at the asymptotic dominant term, namely we compare $T_0(\delta)$ and $T_1(\delta)$. Taking $w = 1_K/K$ instead of $w^\star(\mu)$ in the definition of $T^\star(\mu)$, it is direct to see that $T^\star(\mu) \leq \frac{4K}{\min_{i \neq i^\star}(\mu_{i^\star} - \mu_i)^2}$. Using Lemma C.1, we have $T_\beta^\star(\mu) \leq T^\star(\mu) \max\left\{\frac{\beta^\star}{\beta}, \frac{1-\beta^\star}{1-\beta}\right\}$ where $\beta^\star = w_{i^\star}^\star(\mu)$. Therefore, while we can't say that $T_\beta^\star(\mu)\frac{(1+\varepsilon)^2}{\beta(1-w_0)^{d_\mu(w_0)}} \leq \frac{4K}{\min_{i \neq i^\star}(\mu_{i^\star} - \mu_i)^2}$ for all instances $\mu$, empirical evidences suggest that the gap is significant for reasonable instances. It is even possible to design instances where the gap between both notion of complexity explodes.

Second, we examine the dominating $\delta$-independent term. Using Lemma D.7, we obtain by concavity that for all $x > e$

$$0 < h_1(x) - h_3(x) \leq 2\overline{W}'_{-1}(\log(x)) = 2\left(1 - \frac{1}{\overline{W}_{-1}(\log(x))}\right)^{-1} ,$$

which yields that $\lim_{x \to +\infty} h_1(x) - h_3(x) \leq 2$. While those two functions diverges when $x \to +\infty$, their difference remains bounded by a finite quantity. Therefore, $h_1$ and $h_3$ are qualitatively similar. It is direct to see that $\frac{2}{\min_{i \neq i^\star}(\mu_{i^\star} - \mu_i)^2} < H(\mu) \leq \frac{2K}{\min_{i \neq i^\star}(\mu_{i^\star} - \mu_i)^2}$ Therefore, while we can't say that $\alpha H(\mu) \leq \frac{2K}{\min_{i \neq i^\star}(\mu_{i^\star} - \mu_i)^2}$ for all instances $\mu$, the gap can become significant for some instances.

### D.5 Technicalities

Lemma D.7 gathers properties on the function $\overline{W}_{-1}$ which we used in this work.

**Lemma D.7** ([22]). *Let $\overline{W}_{-1}(x) = -W_{-1}(-e^{-x})$ for all $x \geq 1$, where $W_{-1}$ is the negative branch of the Lambert W function. The function $\overline{W}_{-1}$ is increasing on $(1, +\infty)$ and strictly concave on $(1, +\infty)$. In particular, $\overline{W}'_{-1}(x) = \left(1 - \frac{1}{\overline{W}_{-1}(x)}\right)^{-1}$ for all $x > 1$. Then, for all $y \geq 1$ and $x \geq 1$,*

$$\overline{W}_{-1}(y) \leq x \quad \iff \quad y \leq x - \log(x) .$$

*Moreover, for all $x > 1$,*

$$x + \log(x) \leq \overline{W}_{-1}(x) \leq x + \log(x) + \min\left\{\frac{1}{2}, \frac{1}{\sqrt{x}}\right\} .$$

Lemma D.8 is a standard result to upper bound the expected sample complexity of an algorithm.

**Lemma D.8.** *Let $(\mathcal{E}_n)_{n>K}$ be a sequence of events and $T(\delta) > K$ be such that for $n \geq T(\delta)$, $\mathcal{E}_n \subset \{\tau_\delta \leq n\}$. Then, $\mathbb{E}_\mu[\tau_\delta] \leq T(\delta) + \sum_{n>K} \mathbb{P}_\mu(\mathcal{E}_n^\complement)$.*

*Proof.*

$$\mathbb{E}_\mu[\tau_\delta] = \sum_n \mathbb{P}_\mu(\tau_\delta > n) \leq \sum_{n < T(\delta)} \mathbb{P}_\mu(\tau_\delta > n) + \sum_{n \geq T(\delta)} \mathbb{P}_\mu(\mathcal{E}_n^\complement) \leq T(\delta) + \sum_{n>K} \mathbb{P}_\mu(\mathcal{E}_n^\complement) .$$

□

**Technical results**    Lemma D.9 shows that a linear lower bound on the number of samples allocated to the challenger, i.e. $L_{n,i^\star} - N_{n,i^\star}^{i^\star} \gtrsim (1-\beta)(n-1)$.

**Lemma D.9.** *Let $w \in (0, 1-\beta)$ and $(\mathcal{E}_{1,n})_n$ as in (17). For all $n \in \mathbb{N}^\star$, under $\mathcal{E}_{1,n}$,*

$$\frac{w}{1-\beta}\left(L_{n,i^\star} - N_{n,i^\star}^{i^\star}\right) - 1 \geq (n-1)w - f_1(n),$$

*where*

$$f_1(n) = 2H(\mu)g_u(n)/\beta + K/\beta + 2. \tag{14}$$

*Proof.* Using Lemma D.3, we obtain that, under $\mathcal{E}_{1,n}$,

$$\frac{w}{1-\beta}\left(L_{n,i^\star} - N_{n,i^\star}^{i^\star}\right) - 1 = wL_{n,i^\star} - \frac{w}{1-\beta} - 1 \geq w(n-1) - 2H(\mu)g_u(n)/\beta - K/\beta - 2.$$

where the last inequality uses Lemma D.1 for $t = n$ and $w \in (0, 1-\beta)$. $\qquad\square$

Lemma D.10 gives an explicit upper bound on the constant $C_0(w_-)$ defined implicitly in Appendix D.2.

**Lemma D.10.** *Let $w_- > 0$ and $C_0(w_-)$ as in (8). Then, we have*

$$C_0(w_-) \leq \max\left\{h_1\left(4\alpha^2(1+s)H(\mu)/\beta\right), \left(\frac{2}{w_-} + 1\right)^{1/(\alpha-1)}\right\}^\alpha.$$

*where $h_1 : \mathbb{R}_+^\star \to \mathbb{R}_+^\star$ is an increasing function for $x \geq 2 + K/\beta$ defined as*

$$h_1(x) := x\overline{W}_{-1}\left(\log(x) + \frac{2 + K/\beta}{x}\right). \tag{15}$$

*Proof.* Using the definition of $g_u$ and Lemma D.7, direct manipulations yields

$$n^{1/\alpha} \geq f_1(n) \iff n^{1/\alpha} \geq c_\mu \log(n^{1/\alpha}) + 2 + K/\beta \iff x - \log(x) \geq d_\mu \iff x \geq \overline{W}_{-1}(d_\mu),$$

where $x = n^{1/\alpha}/c_\mu$, $d_\mu = \log(c_\mu) + (2 + K/\beta)/c_\mu$ and $c_\mu = 4\alpha^2(1+s)H(\mu)/\beta$. For the last equivalence we used that we are only interested in $x \geq 1$ (small values are not relevant for upper bounds). Since $\overline{W}_{-1}(d_\mu) \geq 1$, this condition is neither restrictive nor informative as we obtain the final condition $x \geq \overline{W}_{-1}(d_\mu)$. Moreover, we can show

$$d_\mu \geq 1 \iff c_\mu(\log(c_\mu) - 1) \geq -(2 + K/\beta) \impliedby c_\mu(\log(c_\mu) - 1) \geq -1,$$

where the last part is true since $2 + K/\beta \geq 1$ and $\min_{x\in\mathbb{R}} x(\log(x) - 1) = -1$ by direct computations. Therefore, for all $n \geq h_1(c_\mu)^\alpha$, we have

$$n - 1 \geq \frac{1}{w_-}\left(n^{1/\alpha} + f_1(n)\right) \impliedby n - 1 \geq \frac{2}{w_-}n^{1/\alpha} \impliedby n \geq \left(\frac{2}{w_-} + 1\right)^{\alpha/(\alpha-1)}.$$

Given the definition of $C_0(w_-)$ as in (8), this concludes the proof.

Since $x \to \overline{W}_{-1}(x)$ is a positive and increasing function, i.e. $\overline{W}_{-1}(x) > 0$ and $\overline{W}'_{-1}(x) > 0$, a sufficient condition for $h_1$ to be increasing is to have $f : x \to \log(x) + \frac{2+K/\beta}{x}$ increasing since

$$h'_1(x) = \overline{W}_{-1}(f(x)) + xf'(x)\overline{W}'_{-1}(f(x)) > 0 \impliedby \overline{W}_{-1}(f(x)) > 0 \text{ and } f'(x)\overline{W}'_{-1}(f(x)) \geq 0.$$

Since $f'(x) = \frac{1}{x} - \frac{2+K/\beta}{x^2}$, we have that $h_1$ is increasing for $x \geq 2 + K/\beta$. $\qquad\square$

Lemma D.11 gives an explicit upper bound on the constant $C_1(w_-)$ defined implicitly in Appendix D.2.

**Lemma D.11.** *Let $w_- > 0$ and $C_1(w_-)$ as in (9). Then, we have*

$$C_1(w_-) \leq \max\left\{h_1\left(4\alpha^2(1+s)H(\mu)/\beta\right), \left(\frac{1 + 1/\varepsilon}{w_-} + 1\right)^{1/(\alpha-1)}\right\}^\alpha.$$

*Proof.* Using manipulations conducted in Lemma D.10, we obtain that, for all $n \geq h_1 \left(4\alpha^2(1+s)H(\mu)/\beta\right)^\alpha$,

$$n - 1 \geq \frac{f_1(n)}{w_-}\left(1 + \frac{1}{\varepsilon}\right) \impliedby n - 1 \geq \frac{1 + 1/\varepsilon}{w_-}n^{1/\alpha} \impliedby n \geq \left(\frac{1 + 1/\varepsilon}{w_-} + 1\right)^{\alpha/(\alpha-1)} .$$

Given the definition of $C_1(w_-)$ as in (9), this concludes the proof. $\square$

Lemma D.12 gives an asymptotic upper bound for times that are defined implicitly. For example, it can be used with $T_0(\delta)$ and $T_1(\delta)$ defined in Theorem D.4 and Theorem D.6.

**Lemma D.12.** *Let $C > 0$, $D > 0$, $c(n,\delta)$ as in (2) and*

$$T(\delta) := \sup\left\{n \in \mathbb{N}^\star \mid n - 1 \leq C\left(\sqrt{c(n-1,\delta)} + \sqrt{D\log n}\right)^2\right\} .$$

*Then, we have* $\limsup_{\delta \to 0} \frac{T(\delta,C)}{\log(1/\delta)} \leq C$.

*Proof.* Let $\gamma > 0$. Direct manipulations yield that

$$n - 1 \leq C\left(\sqrt{c(n-1,\delta)} + \sqrt{D\log n}\right)^2$$

$$\iff \left(\sqrt{n-1} - \sqrt{CD\log n}\right)^2 - 4\log\left(4 + \log\frac{n}{2}\right) \leq 2CC_G\left(\frac{1}{2}\log\left(\frac{K-1}{\delta}\right)\right)$$

$$\impliedby n \leq \frac{2C}{1+\gamma}C_G\left(\frac{1}{2}\log\left(\frac{K-1}{\delta}\right)\right) \qquad\qquad \text{for } n > N_\gamma$$

where $N_\gamma = \sup\left\{n \in \mathbb{N}^\star \mid \left(\sqrt{n-1} - \sqrt{CD\log n}\right)^2 - 4\log\left(4 + \log\frac{n}{2}\right) > (1+\gamma)n\right\}$. Therefore, we have

$$T(\delta,C) \leq N_\gamma + 1 + \frac{2C}{1+\gamma}C_G\left(\frac{1}{2}\log\left(\frac{K-1}{\delta}\right)\right) .$$

Using that $C_G(x) \approx x + \log(x)$ (Lemma E.1), we obtain directly that, for all $\gamma > 0$,

$$\limsup_{\delta \to 0} \frac{T(\delta,C)}{\log(1/\delta)} \leq \frac{C}{1+\gamma} ,$$

which yields the result by letting $\gamma$ go to zero. $\square$

Lemma D.13 gives a non-asymptotic upper bound for times that are defined implicitly. For example, when considering the idealized choice $c(n,\delta) = \log(1/\delta)$, it can be used as a first order approximation of $T_0(\delta)$ and $T_1(\delta)$ defined in Theorem D.4 and Theorem D.6.

**Lemma D.13.** *Let $C > 0$, $D > 0$ and $T(\delta) := \sup\{n \in \mathbb{N}^\star \mid n - 1 \leq C\log(1/\delta) + D\log n\}$. Then, we have*

$$T(\delta) < D\overline{W}_{-1}\left(\frac{C}{D}\log(1/\delta) + 1/D + \log D\right)$$

*for $\log(1/\delta) \geq \frac{D - D\log(D) - 1}{C}$, else $T(\delta) < D$.*

*Proof.* Direct manipulations yield that

$$n - 1 \geq C\log(1/\delta) + D\log n \iff y - \log y \geq \frac{C}{D}\log(1/\delta) + c_D \iff n \geq D\overline{W}_{-1}\left(\frac{C}{D}\log(1/\delta) + c_D\right)$$

where $y = n/D$ and $c_D = 1/D + \log D$. For the last equivalence, we used that $\frac{C}{D}\log(1/\delta) + c_D \geq 1$ if and only if $\log(1/\delta) \geq \frac{D}{C}(1 - c_D) = \frac{D - D\log(D) - 1}{C}$. When $\frac{C}{D}\log(1/\delta) + c_D < 1$, which means that the $\delta$ parameter is large, we have directly that $y - \log y \geq 1 > \frac{C}{D}\log(1/\delta) + c_D$ for $y \geq 1$. Therefore, the condition becomes $n \geq D$. $\square$

# E Concentration

The proof of Lemma 2.1 is given in Appendix E.1. In Appendix E.2, we show concentration results for the UCB leader (both with $g_u$ and $g_m$) and the TC challenger.

## E.1 Proof of Lemma 2.1

Proving $\delta$-correctness of a GLR stopping rule is done by leveraging concentration results. In particular, we build upon Theorem 9 of [27], which is restated below.

**Lemma E.1.** *Consider Gaussian bandits with means $\mu \in \mathbb{R}^K$ and unit variance. Let $S \subseteq [K]$ and $x > 0$.*

$$\mathbb{P}_\mu \left[ \exists n \in \mathbb{N}, \sum_{k \in S} \frac{N_{n,k}}{2}(\mu_{n,k} - \mu_k)^2 > \sum_{k \in S} 2\log\left(4 + \log\left(N_{n,k}\right)\right) + |S|\mathcal{C}_G\left(\frac{x}{|S|}\right) \right] \leq e^{-x}$$

*where $\mathcal{C}_G$ is defined in [27] by $\mathcal{C}_G(x) = \min_{\lambda \in ]1/2,1]} \frac{g_G(\lambda)+x}{\lambda}$ and*

$$g_G(\lambda) = 2\lambda - 2\lambda \log(4\lambda) + \log\zeta(2\lambda) - \frac{1}{2}\log(1-\lambda), \tag{16}$$

*where $\zeta$ is the Riemann $\zeta$ function and $\mathcal{C}_G(x) \approx x + \log(x)$.*

Since $\hat{\imath}_n = i^\star(\mu_n)$, standard manipulations yield that for all $k \neq \hat{\imath}_n$

$$\frac{(\mu_{n,\hat{\imath}_n} - \mu_{n,k})^2}{\frac{1}{N_{n,\hat{\imath}_n}} + \frac{1}{N_{n,k}}} = \inf_{u \in \mathbb{R}}\left(N_{n,\hat{\imath}_n}(\mu_{n,\hat{\imath}_n} - u)^2 + N_{n,k}(\mu_{n,k} - u)^2\right) = \inf_{y \geq x}\left(N_{n,\hat{\imath}_n}(\mu_{n,\hat{\imath}_n} - x)^2 + N_{n,k}(\mu_{n,k} - y)^2\right).$$

Let $i^\star = i^\star(\mu)$. Using the stopping rule (1) and the above manipulations, we obtain

$$\mathbb{P}_\mu\left(\tau_\delta < +\infty, \hat{\imath}_{\tau_\delta} \neq i^\star\right)$$

$$\leq \mathbb{P}_\mu\left(\exists n \in \mathbb{N}, \exists i \neq i^\star, i = i^\star(\mu_n), \min_{k \neq i}\inf_{y \geq x}\left(N_{n,i}(\mu_{n,i} - x)^2 + N_{n,k}(\mu_{n,k} - y)^2\right) \geq 2c(n-1,\delta)\right)$$

$$\leq \mathbb{P}_\mu\left(\exists n \in \mathbb{N}, \exists i \neq i^\star, i = i^\star(\mu_n), \frac{N_{n,i}}{2}(\mu_{n,i} - \mu_i)^2 + \frac{N_{n,i^\star}}{2}(\mu_{n,i^\star} - \mu_{i^\star})^2 \geq c(n-1,\delta)\right)$$

$$\leq \sum_{i \neq i^\star} \mathbb{P}_\mu\left(\exists n \in \mathbb{N}, \frac{N_{n,i}}{2}(\mu_{n,i} - \mu_i)^2 + \frac{N_{n,i^\star}}{2}(\mu_{n,i^\star} - \mu_{i^\star})^2 \geq c(n-1,\delta)\right)$$

where the second inequality is obtained with $(k, x, y) = (i^\star, \mu_i, \mu_{i^\star})$, and the third by union bound. By concavity of $x \mapsto \log(4 + \log(x))$ and $N_{n,i^\star} + N_{n,i} \leq \sum_{k \in [K]} N_{n,k} = n - 1$, we obtain

$$\forall i \neq i^\star, \quad \log(4 + \log N_{n,i^\star}) + \log(4 + \log N_{n,i}) \leq 2\log(4 + \log((n-1)/2))$$

Combining the above with Lemma E.1 for all $i \neq i^\star$, we obtain $\mathbb{P}_\mu\left(\tau_\delta < +\infty, \hat{\imath} = i^\star\right) \leq \sum_{i \neq i^\star} \frac{\delta}{K-1} = \delta$.

## E.2 Sampling rule

In Lemmas E.2 and E.3, we prove that the UCB leader using $g_u$ and $g_m$ is truly an upper confidence bounds for the unknown mean parameters, when a certain concentration event occurs. Then, when another concentration event occurs, we show a lower bound on the "transportation" costs used by the TC challenger in Lemma E.5. Lemma E.6 upper bounds the probability of not being in the intersection of the two above sequence of concentration events.

**UCB Leader** Lemma E.2 proves that the bonus $g_u$ is sufficient to have upper confidence bounds on the unknown mean $\mu$ for Gaussian observations. The proof uses a simple union bound argument over the time.

**Lemma E.2.** *Let $\alpha > 1$ and $s > 1$. For all $n > K$, let $g_u(n) = 2\alpha(1+s)\log(n)$ and*

$$\mathcal{E}_{1,n} := \left\{\forall k \in [K], \forall t \in [n^{1/\alpha}, n], |\mu_{t,k} - \mu_k| < \sqrt{\frac{g_u(t)}{N_{t,k}}}\right\}. \tag{17}$$

*Then, for all $n > K$, $\mathbb{P}(\mathcal{E}_{1,n}^\complement) \leq Kn^{-s}$.*

*Proof.* Let $(X_s)_{s\in[n]}$ be Gaussian observations from one distribution with unit variance. By union bound over $[K]$ and using that $n \le t^\alpha$, we obtain

$$\mathbb{P}\left(\exists k \in [K], \exists t \in [n^{1/\alpha}, n], |\mu_{t,k} - \mu_k| \ge \sqrt{\frac{2\alpha(1+s)\log(t)}{N_{t,k}}}\right)$$

$$\le \sum_{k\in[K]} \mathbb{P}\left(\exists t \in [n^{1/\alpha}, n], |\mu_{t,k} - \mu_k| \ge \sqrt{\frac{2(1+s)\log(n)}{N_{t,k}}}\right)$$

$$\le \sum_{k\in[K]} \mathbb{P}\left(\exists m \in [n], \left|\frac{1}{m}\sum_{s\in[m]} X_s\right| \ge \sqrt{\frac{2(1+s)\log(n)}{m}}\right)$$

$$\le \sum_{k\in[K]}\sum_{m\in[n]} \mathbb{P}\left(\left|\frac{1}{m}\sum_{s\in[m]} X_s\right| \ge \sqrt{\frac{2(1+s)\log(n)}{m}}\right)$$

$$\le \sum_{k\in[K]}\sum_{m\in[n]} n^{-(1+s)} = Kn^{-s},$$

where we used that $\mu_{t,k} - \mu_k = \frac{1}{N_{t,k}}\sum_{s=1}^t \mathbb{1}(I_s = k)X_{s,k}$ and concentration results for Gaussian observations. $\qquad\square$

Lemma E.3 proves that the bonus $g_m$ is sufficient to have upper confidence bounds on the unknown mean $\mu$ for Gaussian observations. The proof uses a more sophisticated argument based on mixture of martingales.

**Lemma E.3.** *Let $\alpha > 1$ and $s > 1$. For all $x \ge 1$, let $\overline{W}_{-1}(x) = -W_{-1}(-e^{-x})$ where $W_{-1}$ is the negative branch of the Lambert $W$ function. For all $n > K$, let $g_m(n) = \overline{W}_{-1}(2s\alpha\log(n) + 2\log(2 + \alpha\log n) + 2)$ and*

$$\mathcal{E}_{3,n} := \left\{\forall k \in [K], \forall t \in [n^{1/\alpha}, n], |\mu_{t,k} - \mu_k| < \sqrt{\frac{g_m(t)}{N_{t,k}}}\right\}. \tag{18}$$

*Then, for all $n > K$, $\mathbb{P}(\mathcal{E}_{3,n}^{\complement}) \le Kn^{-s}$.*

*Proof.* Let $(X_s)_{s\in[t]}$ the observations from a standard normal distributions and denote $S_t = \sum_{s\in[t]} X_s$. To derived concentration result, we use peeling.

Let $\eta > 0$ and $D = \lceil\frac{\log(n)}{\log(1+\eta)}\rceil$. For all $i \in [D]$, let $\gamma_i > 0$ and $N_i = (1+\eta)^{i-1}$. For all $i \in [D]$, we define the family of priors $f_{N_i,\gamma_i}(x) = \sqrt{\frac{\gamma_i N_i}{2\pi}}\exp\left(-\frac{x^2\gamma_i N_i}{2}\right)$ with weights $w_i = \frac{1}{D}$ and process

$$\overline{M}(t) = \sum_{i\in[D]} w_i \int f_{N_i,\gamma_i}(x)\exp\left(xS_t - \frac{1}{2}x^2 t\right) dx,$$

which satisfies $\overline{M}(0) = 1$. It is direct to see that $M(t) = \exp\left(xS_t - \frac{1}{2}x^2 t\right)$ is a non-negative martingale. By Tonelli's theorem, then $\overline{M}(t)$ is also a non-negative martingale of unit initial value. Let $i \in [D]$ and consider $t \in [N_i, N_{i+1})$. For all $x$,

$$f_{N_i,\gamma}(x) \ge \sqrt{\frac{N_i}{t}}f_{t,\gamma_i}(x) \ge \frac{1}{\sqrt{1+\eta}}f_{t,\gamma_i}(x)$$

Direct computations shows that

$$\int f_{t,\gamma_i}(x)\exp\left(xS_t - \frac{1}{2}x^2 t\right) dx = \frac{1}{\sqrt{1+\gamma_i^{-1}}}\exp\left(\frac{S_t^2}{2(1+\gamma_i)t}\right).$$

Minoring $\overline{M}(t)$ by one of the positive term of its sum, we obtain

$$\overline{M}(t) \geq \frac{1}{D} \frac{1}{\sqrt{(1 + \gamma_i^{-1})(1 + \eta)}} \exp\left(\frac{S_t^2}{2(1 + \gamma_i)t}\right) ,$$

Using Ville's maximal inequality, we have that with probability greater than $1 - \delta$, $\log \overline{M}(t) \leq \log(1/\delta)$. Therefore, with probability greater than $1 - \delta$, for all $i \in [D]$ and $t \in [N_i, N_{i+1})$,

$$\frac{S_t^2}{t} \leq (1 + \gamma_i)\left(2\log(1/\delta) + 2\log D + \log(1 + \gamma_i^{-1}) + \log(1 + \eta)\right) .$$

Since this upper bound is independent of $t$, we can optimize it and choose $\gamma_i$ as in Lemma E.4 for all $i \in [D]$.

**Lemma E.4** (Lemma A.3 in [11]). *For $a, b \geq 1$, the minimal value of $f(\eta) = (1+\eta)(a + \log(b + \frac{1}{\eta}))$ is attained at $\eta^\star$ such that $f(\eta^\star) \leq 1 - b + \overline{W}_{-1}(a + b)$. If $b = 1$, then there is equality.*

Therefore, with probability greater than $1 - \delta$, for all $i \in [D]$ and $t \in [N_i, N_{i+1})$,

$$\frac{S_t^2}{t} \leq \overline{W}_{-1}\left(1 + 2\log(1/\delta) + 2\log D + \log(1 + \eta)\right)$$
$$\leq \overline{W}_{-1}\left(1 + 2\log(1/\delta) + 2\log(\log(1 + \eta) + \log n) - 2\log\log(1 + \eta) + \log(1 + \eta)\right)$$
$$= \overline{W}_{-1}\left(2\log(1/\delta) + 2\log(2 + \log n) + 3 - 2\log 2\right)$$

The second inequality is obtained since $D \leq 1 + \frac{\log n}{\log(1+\eta)}$. The last equality is obtained for the choice $\eta^\star = e^2 - 1$, which minimizes $\eta \mapsto \log(1 + \eta) - 2\log(\log(1 + \eta))$. Since $[n] \subseteq \bigcup_{i \in [D]}[N_i, N_{i+1})$ and $N_{t,k}(\mu_{t,k} - \mu_k) = \sum_{s \in [N_{t,k}]} X_{s,k}$ (unit-variance), this yields

$$\mathbb{P}\left(\exists t \leq n, \left|\frac{1}{t}\sum_{s=1}^{t} X_s\right| \geq \sqrt{\frac{1}{t}\overline{W}_{-1}\left(2\log(1/\delta) + 2\log(2 + \log(n)) + 3 - 2\log 2\right)}\right) \leq \delta .$$

Since $3 - 2\log 2 \leq 2$ and $\overline{W}_{-1}$ is increasing, taking $\delta = n^{-s}$ and restricting to $m \in [n^{1/\alpha}, n]$ yields

$$\mathbb{P}\left(\exists m \in [n^{1/\alpha}, n], \sqrt{N_{m,k}}|\mu_{m,k} - \mu_k| \geq \sqrt{\overline{W}_{-1}\left(2s\log(n) + 2\log(2 + \log(n)) + 2\right)}\right) \leq n^{-s} .$$

Using $n \leq m^\alpha$ and doing a union bound over arms yield the result. □

**TC challenger**  Lemma E.5 lower bounds the difference between the empirical gap and the unknown gap.

**Lemma E.5.** *Let $\alpha > 1$ and $s > 1$. For all $n > K$, let*

$$\mathcal{E}_{2,n} := \left\{\forall k \neq i^\star, \forall t \in [n^{1/\alpha}, n], (\mu_{t,i^\star} - \mu_{t,k}) - (\mu_{i^\star} - \mu_k) > -\sqrt{2\alpha(2 + s)\log(t)\left(\frac{1}{N_{t,i^\star}} + \frac{1}{N_{t,k}}\right)}\right\} .$$

(19)

*Then, for all $n > K$, $\mathbb{P}(\mathcal{E}_{2,n}^{\complement}) \leq (K - 1)n^{-s}$.*

*Proof.* Let $(X_s)_{s \in [n]}$ and $(Y_s)_{s \in [n]}$ be Gaussian observations from two distributions with unit variance. Then $\frac{1}{m_1}\sum_{i=1}^{m_1} X_i - \frac{1}{m_2}\sum_{i=1}^{m_2} Y_i$ is Gaussian with variance $\frac{1}{m_1} + \frac{1}{m_2}$. By union bound and

using that $n \leq t^\alpha$, we obtain

$$
\mathbb{P}\left(\exists k \neq i^\star, \exists t \in [n^{1/\alpha}, n], \frac{(\mu_{t,i^\star} - \mu_{t,k}) - (\mu_{i^\star} - \mu_k)}{\sqrt{1/N_{t,i^\star} + 1/N_{t,k}}} \leq -\sqrt{2\alpha(2+s)\log(t)}\right)
$$

$$
\leq \sum_{k \neq i^\star} \mathbb{P}\left(\exists t \in [n^{1/\alpha}, n], \frac{(\mu_{t,i^\star} - \mu_{t,k}) - (\mu_{i^\star} - \mu_k)}{\sqrt{1/N_{t,i^\star} + 1/N_{t,k}}} \leq -\sqrt{2(2+s)\log(n)}\right)
$$

$$
\leq \sum_{k \neq i^\star} \mathbb{P}\left(\exists (m_1, m_2) \in [n]^2, \frac{\frac{1}{m_1}\sum_{i=1}^{m_1} X_i - \frac{1}{m_2}\sum_{i=1}^{m_2} Y_i}{\sqrt{1/m_1 + 1/m_2}} \leq -\sqrt{2(2+s)\log(n)}\right)
$$

$$
\leq \sum_{k \neq i^\star} \sum_{(m_1, m_2) \in [n]^2} \mathbb{P}\left(\frac{1}{m_1}\sum_{i=1}^{m_1} X_i - \frac{1}{m_2}\sum_{i=1}^{m_2} Y_i \leq -\sqrt{2(2+s)\log(n)\,(1/m_1 + 1/m_2)}\right)
$$

$$
\leq \sum_{k \neq i^\star} \sum_{(m_1, m_2) \in [n]^2} n^{-(2+s)} = (K-1)n^{-s},
$$

where we used that $(\mu_{t,i^\star} - \mu_{i^\star}) - (\mu_{t,k} - \mu_k) = \frac{1}{N_{t,i^\star}}\sum_{s=1}^t \mathbb{1}(I_s = i^\star) X_{s,i^\star} - \frac{1}{N_{t,k}}\sum_{s=1}^t \mathbb{1}(I_s = k) X_{s,k}$ and concentration results for Gaussian observations. $\qquad\square$

Using a mixture of martingale arguments, we could improve on Lemma E.5 similarly as $g_u$ improved on $g_m$. This will impact second order terms of our non-asymptotic theoretical guarantees, at the price of less explicit non-asymptotic terms.

**Concentration event**   Lemma E.6 upper bounds the summed probabilities of the complementary events.

**Lemma E.6.** *Let $\zeta$ be the Riemann $\zeta$ function. Let $(\mathcal{E}_{1,n})_{n>K}$, $(\mathcal{E}_{2,n})_{n>K}$ and $(\mathcal{E}_{3,n})_{n>K}$ as in (17), (19) and (18). For all $n > K$, let $\mathcal{E}_n = \mathcal{E}_{1,n} \cap \mathcal{E}_{2,n}$ and $\tilde{\mathcal{E}}_n = \mathcal{E}_{3,n} \cap \mathcal{E}_{2,n}$. Then,*

$$
\max\left\{\sum_{n>K} \mathbb{P}(\mathcal{E}_n^\complement), \sum_{n>K} \mathbb{P}(\tilde{\mathcal{E}}_n^\complement)\right\} \leq (2K-1)\zeta(s).
$$

*Proof.*  Using Lemmas E.2 and E.5, direct union bound yields

$$
\sum_{n \in \mathbb{N}^\star} \mathbb{P}(\mathcal{E}_n^\complement) \leq \sum_{n \in \mathbb{N}^\star} \mathbb{P}(\mathcal{E}_{1,n}^\complement) + \mathbb{P}(\mathcal{E}_{2,n}^\complement) \leq \sum_{n \in \mathbb{N}^\star} (2K-1)n^{-s} = (2K-1)\zeta(s).
$$

The same proof trivially holds for $\tilde{\mathcal{E}}_n$ by using Lemmas E.3 and E.5. $\qquad\square$

## F   Asymptotic analysis

Based solely on Theorem D.4, it is not possible to obtain asymptotic $\beta$-optimality due to the multiplicative factor $1/\beta$. Building on the unified analysis proposed in [21], we prove Theorem 2.3.

Our main technical contribution for this proof lies in the use of tracking instead of sampling. Given that the cumulative probability of being sampled is the expectation of the random empirical counts, it is not surprising that tracking-based Top Two algorithms enjoy the same theoretical guarantees as their sampling counterpart. As we will see, the analysis to obtain similar result is even simpler (Lemmas F.6, F.8 and F.10). Apart from this technical subtlety, the proof of Theorem 2.3 boils down to showing that the UCB leader satisfies the two sufficient properties highlighted by previous work (Lemmas F.4 and F.7).

Using the fact that $x \to \sqrt{2x}$ is increasing, it is direct to see that the TC challenger (4) coincides with the definition used by [39, 21], i.e.

$$
C_n^{\mathrm{TC}} = \arg\min_{i \neq B_n} \frac{(\mu_{n,B_n} - \mu_{n,i})_+}{\sqrt{1/N_{n,B_n} + 1/N_{n,i}}} = \arg\min_{i \neq B_n} \mathbb{1}(\mu_{n,B_n} > \mu_{n,i}) \frac{(\mu_{n,B_n} - \mu_{n,i})^2}{2(1/N_{n,B_n} + 1/N_{n,i})}.
$$

## F.1 Proof of Theorem 2.3

Let $\mu \in \mathcal{D}^K$ such that $\min_{i \neq j} |\mu_i - \mu_j| > 0$, and let $i^\star = i^\star(\mu)$ be the unique best arm. Let $\beta \in (0, 1)$ and $w_\beta^\star$ be the unique allocation $\beta$-optimal allocation satisfying $w_{\beta,i}^\star > 0$ for all $i \in [K]$ (Lemma C.2), i.e. $w_\beta^\star(\mu) = \{w_\beta^\star\}$ where

$$w_\beta^\star(\mu) := \underset{w \in \triangle_K : w_{i^\star} = \beta}{\arg\max} \ \min_{i \neq i^\star} \frac{(\mu_{i^\star} - \mu_i)^2}{2(1/\beta + 1/w_i)} = \underset{w \in \triangle_K : w_{i^\star} = \beta}{\arg\max} \ \min_{i \neq i^\star} \frac{\mu_{i^\star} - \mu_i}{\sqrt{1/\beta + 1/w_i}} \ .$$

Let $\varepsilon > 0$. Following [36, 39, 21], we aim at upper bounding the expectation of the *convergence time* $T_\beta^\varepsilon$, which is a random variable quantifies the number of samples required for the empirical allocations $N_n/(n-1)$ to be $\varepsilon$-close to $w_\beta^\star$:

$$T_\beta^\varepsilon := \inf \left\{ T \geq 1 \mid \forall n \geq T, \ \left\| \frac{N_n}{n-1} - w_\beta^\star \right\|_\infty \leq \varepsilon \right\} \ . \tag{20}$$

Lemma F.1 shows that a sufficient condition for asymptotic $\beta$-optimality is to show $\mathbb{E}_\mu[T_\beta^\varepsilon] < +\infty$ for all $\varepsilon$ small enough.

**Lemma F.1.** *Let $(\delta, \beta) \in (0,1)^2$. Assume that there exists $\varepsilon_1(\mu) > 0$ such that for all $\varepsilon \in (0, \varepsilon_1(\mu)]$, $\mathbb{E}_\mu[T_\beta^\varepsilon] < +\infty$. Combining the stopping rule (1) with threshold as in (2) yields an algorithm such that, for all $\mu \in \mathbb{R}^K$ with $|i^\star(\mu)| = 1$,*

$$\limsup_{\delta \to 0} \frac{\mathbb{E}_\mu[\tau_\delta]}{\log(1/\delta)} \leq T_\beta^\star(\mu) \ .$$

*Proof.* While the first result in the spirit of Lemma F.1 was derived by [36] for Gaussian distributions, a proof holding for more general distributions is given by Theorem 2 in [21]. The sole criterion on the stopping threshold is to be asymptotically tight (Definition F.2).

**Definition F.2.** A threshold $c : \mathbb{N} \times (0,1] \to \mathbb{R}_+$ is said to be asymptotically tight if there exists $\alpha \in [0, 1)$, $\delta_0 \in (0, 1]$, functions $f, \bar{T} : (0, 1] \to \mathbb{R}_+$ and $C$ independent of $\delta$ satisfying: (1) for all $\delta \in (0, \delta_0]$ and $n \geq \bar{T}(\delta)$, then $c(n, \delta) \leq f(\delta) + Cn^\alpha$, (2) $\limsup_{\delta \to 0} f(\delta)/\log(1/\delta) \leq 1$ and (3) $\limsup_{\delta \to 0} \bar{T}(\delta)/\log(1/\delta) = 0$.

Since $\mathcal{C}_G$ defined in (16) satisfies $\mathcal{C}_G \approx x + \log(x)$, it is direct to see that

$$c(n, \delta) = 2\mathcal{C}_G\left(\frac{1}{2}\log\left(\frac{K-1}{\delta}\right)\right) + 4\log\left(4 + \log\frac{n}{2}\right)$$

is asymptotically tight, e.g. by taking $(\alpha, \delta_0, C) = (1/2, 1, 4)$, $f(\delta) = 2\mathcal{C}_G\left(\frac{1}{2}\log\left(\frac{K-1}{\delta}\right)\right)$ and $\bar{T}(\delta) = 1$. This concludes the proof. $\qquad\square$

Throughout the proof, we will use a concentration result of the empirical mean (Lemma F.3) Since this is a standard result for Gaussian observations (see Lemma 5 of [36]), we omit the proof.

**Lemma F.3.** *There exists a sub-Gaussian random variable $W_\mu$ such that almost surely, for all $i \in [K]$ and all $n$ such that $N_{n,i} \geq 1$,*

$$|\mu_{n,i} - \mu_i| \leq W_\mu \sqrt{\frac{\log(e + N_{n,i})}{N_{n,i}}} \ .$$

*In particular, any random variable which is polynomial in $W_\mu$ has a finite expectation.*

**Sufficient exploration** To upper bound the expected convergence time, as prior work we first establish sufficient exploration. Given an arbitrary threshold $L \in \mathbb{R}_+^*$, we define the sampled enough set and its arms with highest mean (when not empty) as

$$S_n^L := \{i \in [K] \mid N_{n,i} \geq L\} \quad \text{and} \quad \mathcal{I}_n^\star := \underset{i \in S_n^L}{\arg\max} \ \mu_i \ . \tag{21}$$

Since $\min_{i\neq j}|\mu_i - \mu_j| > 0$, $\mathcal{I}_n^\star$ is a singleton. We define the highly and the mildly under-sampled sets

$$U_n^L := \{i \in [K] \mid N_{n,i} < \sqrt{L}\} \quad \text{and} \quad V_n^L := \{i \in [K] \mid N_{n,i} < L^{3/4}\} . \tag{22}$$

[21] identifies the properties that the leader and the challenger should satisfy to ensure sufficient exploration. Lemma F.4 show that the desired property for the UCB leader defined in (3) with bonus $g_u(n) = 2\alpha(1+s)\log n$ or

$$g_m(n) = \overline{W}_{-1}\left(2s\alpha\log(n) + 2\log(2 + \alpha\log n) + 2\right) .$$

**Lemma F.4.** *There exists $L_0$ with $\mathbb{E}_\mu[(L_0)^\alpha] < +\infty$ for all $\alpha > 0$ such that if $L \geq L_0$, for all $n$ (at most polynomial in $L$) such that $S_n^L \neq \emptyset$, $B_n^{UCB} \in S_n^L$ implies $B_n^{UCB} \in \mathcal{I}_n^\star$ and $B_n^{UCB} \in \arg\max_{i\in S_n^L}\mu_{n,i}$.*

*Proof.* Let $\varepsilon > 0$ and $\Delta_{\min} = \min_{i\neq j}|\mu_i - \mu_j|$. Let $g$ denote either $g_u$ or $g_m$. Let $L \geq L_0$, where $L_0$ will be specified later, and $n$ (at most polynomial in $L$) such that $S_n^L \neq \emptyset$. Then, there exists a polynomial function $P$ such that $n \leq P(L)$. By considering arms that are sampled more than $L$, we can show that for all $k \in S_n^L \setminus \mathcal{I}_n^\star$,

$$\mu_{n,k} + \sqrt{\frac{g(n)}{N_{n,k}}} \leq \mu_k + W_\mu\sqrt{\frac{\log(e + N_{n,k})}{1 + N_{n,k}}} + \sqrt{\frac{g(P(L))}{L}} \leq \mu_k + W_\mu\sqrt{\frac{\log(e + L)}{1 + L}} + \sqrt{\frac{g(P(L))}{L}} \leq \mu_k + 2\varepsilon ,$$

where the last inequality is obtained for $L \geq L_0$ with

$$L_0 = 1 + \sup\left\{L \in \mathbb{N}^\star \mid W_\mu\sqrt{\frac{\log(e + L)}{1 + L}} > \varepsilon, \ \sqrt{\frac{g(P(L))}{L}} > \varepsilon\right\} .$$

Since $\overline{W}_{-1}(x) \approx x + \log(x)$, both $g_u$ and $g_m$ have a logarithmic behavior, hence $g(P(L)) =_{L\to+\infty} o(L)$. Since $L_0$ is polynomial in $W_\mu$, Lemma F.3 yields $\mathbb{E}_\mu[(L_0)^\alpha] < +\infty$ for all $\alpha > 0$.

Moreover, for all $k \in \mathcal{I}_n^\star$,

$$\mu_{n,k} + \sqrt{\frac{g(n)}{N_{n,k}}} \geq \mu_k - W_\mu\sqrt{\frac{\log(e + N_{n,k})}{1 + N_{n,k}}} \geq \mu_k - W_\mu\sqrt{\frac{\log(e + L)}{1 + L}} \geq \mu_k - \varepsilon .$$

Assume that $B_n^{\text{UCB}} \in S_n^L$ and that $B_n^{\text{UCB}} \notin \mathcal{I}_n^\star$. Since $B_n^{\text{UCB}} = \arg\max_{k\in[K]}\left\{\mu_{n,k} + \sqrt{\frac{g(n)}{N_{n,k}}}\right\}$, taking $\varepsilon < \Delta_{\min}/3$ in the above yields a direct contradiction. $\square$

We can use the proof of [21] to obtain Lemma F.5. While their result accounts for the randomization of the sampling procedure, the argument is direct for tracking since it removes the need for a concentration argument.

**Lemma F.5** (Lemma 19 in [21]). *Let $\mathcal{J}_n^\star = \arg\max_{i\in\overline{V_n^L}}\mu_i$. There exists $L_1$ with $\mathbb{E}_\mu[L_1] < +\infty$ such that if $L \geq L_1$, for all $n$ (at most polynomial in $L$) such that $U_n^L \neq \emptyset$, $B_n^{UCB} \notin V_n^L$ implies $C_n^{TC} \in V_n^L \cup \left(\mathcal{J}_n^\star \setminus \{B_n^{UCB}\}\right)$.*

Lemma F.6 proves sufficient exploration for the TTUCB sampling rule. It builds on the same reasoning than the one used in the proofs introduced by [39], and generalized by [21].

**Lemma F.6.** *Assume $\min_{j\neq i}|\mu_i - \mu_j| > 0$. Under the TTUCB sampling rule, there exist $N_0$ with $\mathbb{E}_\mu[N_0] < +\infty$ such that for all $n \geq N_0$ and all $i \in [K]$, $N_{n,i} \geq \sqrt{n/K}$.*

*Proof.* Let $L_0$ and $L_1$ as in Lemmas F.4 and F.5. Therefore, for $L \geq L_2 := \max\{L_1, L_0^{4/3}\}$, for all $n$ such that $U_n^L \neq \emptyset$, $B_n^{\text{UCB}} \in V_n^L$ or $C_n^{\text{TC}} \in V_n^L$ since $|\mathcal{J}_n^\star| = 1$ is implied by $\min_{j\neq i}|\mu_i - \mu_j| > 0$. We have $\mathbb{E}_\mu[L_2] < +\infty$. There exists a deterministic $L_4$ such that for all $L \geq L_4$, $\lfloor L \rfloor \geq KL^{3/4}$. Let $L \geq \max\{L_2, L_4\}$.

Suppose towards contradiction that $U_{\lfloor KL \rfloor}^L$ is not empty. Then, for any $1 \leq t \leq \lfloor KL \rfloor$, $U_t^L$ and $V_t^L$ are non empty as well. Using the pigeonhole principle, there exists some $i \in [K]$ such that

$N_{\lfloor L \rfloor, i} \geq L^{3/4}$. Thus, we have $\left| V_{\lfloor L \rfloor}^{L} \right| \leq K - 1$. Our goal is to show that $\left| V_{\lfloor 2L \rfloor}^{L} \right| \leq K - 2$. A sufficient condition is that one arm in $V_{\lfloor L \rfloor}^{L}$ is pulled at least $L^{3/4}$ times between $\lfloor L \rfloor$ and $\lfloor 2L \rfloor - 1$.

**Case 1.** Suppose there exists $i \in V_{\lfloor L \rfloor}^{L}$ such that $L_{\lfloor 2L \rfloor, i} - L_{\lfloor L \rfloor, i} \geq \frac{L^{3/4}}{\beta} + 3/(2\beta)$. Using Lemma D.3, we obtain

$$N_{\lfloor 2L \rfloor, i}^{i} - N_{\lfloor L \rfloor, i}^{i} \geq \beta(L_{\lfloor 2L \rfloor, i} - L_{\lfloor L \rfloor, i}) - 3/2 \geq L^{3/4} \ ,$$

hence $i$ is sampled $L^{3/4}$ times between $\lfloor L \rfloor$ and $\lfloor 2L \rfloor - 1$.

**Case 2.** Suppose that, for all $i \in V_{\lfloor L \rfloor}^{L}$, $L_{\lfloor 2L \rfloor, i} - L_{\lfloor L \rfloor, i} < \frac{L^{3/4}}{\beta} + 3/(2\beta)$. Then,

$$\sum_{i \notin V_{\lfloor L \rfloor}^{L}} (L_{\lfloor 2L \rfloor, i} - L_{\lfloor L \rfloor, i}) \geq (\lfloor 2L \rfloor - \lfloor L \rfloor) - K \left( \frac{L^{3/4}}{\beta} + 3/(2\beta) \right)$$

Using Lemma D.3, we obtain

$$\left| \sum_{i \notin V_{\lfloor L \rfloor}^{L}} (N_{\lfloor 2L \rfloor, i}^{i} - N_{\lfloor L \rfloor, i}^{i}) - \beta \sum_{i \notin V_{\lfloor L \rfloor}^{L}} (L_{\lfloor 2L \rfloor, i} - L_{\lfloor L \rfloor, i}) \right| \leq 3(K-1)/2 \ .$$

Combining all the above, we obtain

$$\sum_{i \notin V_{\lfloor L \rfloor}^{L}} (L_{\lfloor 2L \rfloor, i} - L_{\lfloor L \rfloor, i}) - \sum_{i \notin V_{\lfloor L \rfloor}^{L}} (N_{\lfloor 2L \rfloor, i}^{i} - N_{\lfloor L \rfloor, i}^{i})$$

$$\geq (1 - \beta) \sum_{i \notin V_{\lfloor L \rfloor}^{L}} (L_{\lfloor 2L \rfloor, i} - L_{\lfloor L \rfloor, i}) - 3(K-1)/2$$

$$\geq (1 - \beta) \left( (\lfloor 2L \rfloor - \lfloor L \rfloor) - K \left( \frac{L^{3/4}}{\beta} + 3/(2\beta) \right) \right) - 3(K-1)/2 \geq K L^{3/4} \ ,$$

where the last inequality is obtained for $L \geq L_5$ with

$$L_5 = \sup \left\{ L \in \mathbb{N} \mid (1 - \beta) \left( (\lfloor 2L \rfloor - \lfloor L \rfloor) - K \left( \frac{L^{3/4}}{\beta} + 3/(2\beta) \right) \right) - 3(K-1)/2 < K L^{3/4} \right\}.$$

The l.h.s. summation is exactly the number of times where an arm $i \in \overline{V_{\lfloor L \rfloor}^{L}}$ was leader but wasn't sampled, hence

$$\sum_{t=\lfloor L \rfloor}^{\lfloor 2L \rfloor - 1} \mathbb{1} \left( B_t^{\text{UCB}} \in \overline{V_{\lfloor L \rfloor}^{L}}, \ I_t = C_t^{\text{TC}} \right) \geq K L^{3/4}$$

For any $\lfloor L \rfloor \leq t \leq \lfloor 2L \rfloor - 1$, $U_t^L$ is non-empty, hence we have $B_t^{\text{UCB}} \in \overline{V_{\lfloor L \rfloor}^{L}} \subseteq \overline{V_t^{L}}$ implies $C_t^{\text{TC}} \in V_t^L \subseteq V_{\lfloor L \rfloor}^{L}$. Therefore, we have shown that

$$\sum_{t=\lfloor L \rfloor}^{\lfloor 2L \rfloor - 1} \mathbb{1} \left( I_t \in V_{\lfloor L \rfloor}^{L} \right) \geq \sum_{t=\lfloor L \rfloor}^{\lfloor 2L \rfloor - 1} \mathbb{1} \left( B_t^{\text{UCB}} \in \overline{V_{\lfloor L \rfloor}^{L}}, \ I_t = C_t^{\text{TC}} \right) \geq K L^{3/4} \ .$$

Therefore, there is at least one arm in $V_{\lfloor L \rfloor}^{L}$ that is sampled $L^{3/4}$ times between $\lfloor L \rfloor$ and $\lfloor 2L \rfloor - 1$.

In summary, we have shown $\left| V_{\lfloor 2L \rfloor}^{L} \right| \leq K - 2$. By induction, for any $1 \leq k \leq K$, we have $\left| V_{\lfloor kL \rfloor}^{L} \right| \leq K - k$, and finally $U_{\lfloor KL \rfloor}^{L} = \emptyset$ for all $L \geq L_3$. This concludes the proof. $\qquad \square$

**Convergence towards $\beta$-optimal allocation** Provided sufficient exploration holds (Lemma F.6), [21] identifies the properties that the leader and the challenger should satisfy to obtain convergence towards the $\beta$-optimal allocation $w_\beta^\star$. Lemma F.7 show that the desired property for the UCB leader defined in (3) with bonus $g_u(n) = 2\alpha(1 + s) \log n$ or

$$g_m(n) = \overline{W}_{-1} \left( 2s\alpha \log(n) + 2 \log(2 + \alpha \log n) + 2 \right) \ .$$

**Lemma F.7.** *There exists $N_1$ with $\mathbb{E}_\mu[N_1] < +\infty$ such that for all $n \geq N_1$, $B_n^{UCB} = i^\star(\mu)$.*

*Proof.* Let $\Delta = \min_{i \neq i^\star} |\mu_{i^\star} - \mu_i|$. Let $g$ denote either $g_u$ or $g_m$. Let $N_0$ as in Lemma F.6. For all $n \geq N_2$ (to be specified later), we obtain for all $k \neq i^\star$,

$$\mu_{n,k} + \sqrt{\frac{g(n)}{N_{n,k}}} \leq \mu_k + W_\mu\sqrt{\frac{\log(e + N_{n,k})}{1 + N_{n,k}}} + K^{1/4}\sqrt{\frac{g(n)}{\sqrt{n}}} \leq \mu_k + W_\mu\sqrt{\frac{\log(e + \sqrt{\frac{n}{K}})}{1 + \sqrt{\frac{n}{K}}}} + K^{1/4}\sqrt{\frac{g(n)}{\sqrt{n}}} \leq \mu_k + 2\varepsilon\,,$$

where the last inequality is obtained for $n \geq N_1$ where

$$N_1 = 1 + \sup\left\{n \in N^\star \mid W_\mu\sqrt{\frac{\log(e + \sqrt{\frac{n}{K}})}{1 + \sqrt{\frac{n}{K}}}} > \varepsilon, \ K^{1/4}\sqrt{\frac{g(n)}{\sqrt{n}}} > \varepsilon\right\}\,.$$

Since $\overline{W}_{-1}(x) \approx x + \log(x)$, both $g_u$ and $g_m$ have a logarithmic behavior, hence $g(n) =_{n \to +\infty} o(\sqrt{n})$. Since $N_1$ is polynomial in $W_\mu$, Lemma F.3 yields $\mathbb{E}_\mu[(N_1)^\alpha] < +\infty$ for all $\alpha > 0$.

Moreover

$$\mu_{n,i^\star} + \sqrt{\frac{g(n)}{N_{n,i^\star}}} \geq \mu_{i^\star} - W_\mu\sqrt{\frac{\log(e + N_{n,i^\star})}{1 + N_{n,i^\star}}} \geq \mu_{i^\star} - W_\mu\sqrt{\frac{\log(e + \sqrt{\frac{n}{K}})}{1 + \sqrt{\frac{n}{K}}}} \geq \mu_{i^\star} - \varepsilon\,.$$

Therefore, taking $\varepsilon < \Delta/3$, yields the result that $B_n^{\text{UCB}} = i^\star$. $\qquad\square$

Combining Lemma F.7 with tracking properties (Lemma D.3), we obtain convergence towards the $\beta$-optimal allocation for the best arm (Lemma F.8).

**Lemma F.8.** *Let $\varepsilon > 0$. Under the TTUCB sampling rule, there exists $N_2$ with $\mathbb{E}_\mu[N_2] < +\infty$ such that for all $n \geq N_2$,*

$$\left|\frac{N_{n,i^\star(\mu)}}{n-1} - \beta\right| \leq \varepsilon\,.$$

*Proof.* Let $N_0$ as in Lemma F.6 and $C_1$ as in Lemma F.7. For all $n \geq \max\{N_1, N_0\}$, we have $B_n = i^\star$. Let $M \geq \max\{N_1, N_0\}$. Then, we have $L_{n,i^\star} \geq n - M$ and $\sum_{k \neq i^\star} N_{n,i^\star}^k \leq M - 1$ for all $n \geq M + 1$. Using Lemma D.3, we have

$$\left|\frac{N_{n,i^\star}}{n-1} - \beta\right| \leq \frac{|N_{n,i^\star}^{i^\star} - \beta L_{n,i^\star}|}{n-1} + \beta\left|\frac{L_{n,i^\star}}{n-1} - 1\right| + \frac{1}{n-1}\sum_{k \neq i^\star} N_{n,i^\star}^k \leq \frac{1}{2(n-1)} + \beta\frac{2(M-1)}{n-1}\,.$$

Taking $N_2 = \max\{N_1, N_0, \frac{1/2 + 2\beta(M-1)}{\varepsilon} + 1\}$ yields the result. $\qquad\square$

We can use the proof of [21] to obtain Lemma F.9. While their result accounts for the randomization of the sampling procedure, the argument is direct for tracking since it removes the need for a concentration argument.

**Lemma F.9** (Lemma 20 in [21])**.** *Let $\varepsilon > 0$. Under the TTUCB sampling rule, there exists $N_3$ with $\mathbb{E}_\mu[N_3] < +\infty$ such that for all $n \geq N_3$ and all $i \neq i^\star(\mu)$,*

$$\frac{N_{n,i}}{n-1} \geq w_{\beta,i}^\star + \varepsilon \quad \implies \quad C_n^{TC} \neq i\,.$$

Combining Lemma F.9 with tracking properties (Lemma D.3), we obtain convergence towards the $\beta$-optimal allocation for all arms (Lemma F.10).

**Lemma F.10.** *Let $\varepsilon > 0$. Under the TTUCB sampling rule, there exists $N_4$ with $\mathbb{E}_\mu[N_4] < +\infty$ such that for all $n \geq N_4$,*

$$\forall i \in [K], \quad \left|\frac{N_{n,i}}{n-1} - w_{\beta,i}^\star\right| \leq \varepsilon\,.$$

*Proof.* Let $N_0$, $C_1$, $N_2$ and $N_3$ as in Lemmas F.6, F.7, F.8 and F.9. For all $n \geq \max\{N_0, N_1, N_2, N_3\}$, we have $B_n^{\mathrm{UCB}} = i^\star$, $\left|\frac{N_{n,i^\star}}{n-1} - \beta\right| \leq \varepsilon$ and for all $i \neq i^\star$,

$$\frac{N_{n-1,i}}{n-1} \geq w_{\beta,i}^\star + \varepsilon \implies C_n^{\mathrm{TC}} \neq i \,.$$

Let $M \geq \max\{N_0, N_1, N_2, N_3\}$ and $n \geq N_5 = \max\{\frac{M-1}{\varepsilon} + 1, M\}$. Let $t_{n-1,i}(\varepsilon) = \max\left\{t \leq n \mid \frac{N_{t,i}}{n-1} \leq w_{\beta,i}^\star + \varepsilon\right\}$. Since $\frac{N_{t,i}}{n-1} \leq \frac{N_{t,i}}{t-1}$ for $t \leq n$, we have

$$\frac{N_{n,i}}{n-1} \leq \frac{M-1}{n-1} + \frac{1}{n-1}\sum_{t=M}^n \mathbb{1}\left(B_t^{\mathrm{UCB}} = i^\star, C_t^{\mathrm{TC}} = i,\, I_t = i\right)$$

$$\leq \varepsilon + \frac{1}{n-1}\sum_{t=M}^n \mathbb{1}\left(\frac{N_{t,i}}{n-1} \leq w_{\beta,i}^\star + \varepsilon\right)\mathbb{1}\left(B_t^{\mathrm{UCB}} = i^\star, C_t^{\mathrm{TC}} = i,\, I_t = i\right)$$

$$\leq \varepsilon + \frac{N_{t_{n-1,i}(\varepsilon),i}}{n-1} \leq w_{\beta,i}^\star + 2\varepsilon \,.$$

As a similar upper bound is shown in the proof of Lemma F.8, we obtain $\frac{N_{n,i}}{n-1} \leq w_{\beta,i}^\star + 2\varepsilon$ for all $i \in [K]$ and all $n \geq N_4 := \max\{N_0, N_1, N_2, N_3, N_5\}$. Since $\frac{N_{n,i}}{n-1}$ and $w_{\beta,i}^\star$ sum to 1, we obtain for all $n \geq N_4$ and all $i \in [K]$,

$$\frac{N_{n,i}}{n-1} = 1 - \sum_{k \neq i}\frac{N_{n,k}}{n-1} \geq 1 - \sum_{k \neq i}\left(w_{\beta,k}^\star + 2\varepsilon\right) = w_{\beta,i}^\star - 2(K-1)\varepsilon \,.$$

Therefore, for all $n \geq N_4$ and all $i \in [K]$, $\left|\frac{N_{n,i}}{n-1} - w_{\beta,i}^\star\right| \leq 2(K-1)\varepsilon$. Since $\mathbb{E}_\mu[N_4] < +\infty$ and we showed the result for all $\varepsilon$, this concludes the proof. $\square$

Lemma F.11 shows that $\mathbb{E}_\mu[T_\beta^\varepsilon] < +\infty$, it is a direct consequence of Lemma F.10.

**Lemma F.11.** *Let $\varepsilon > 0$ and $T_\beta^\varepsilon$ as in (20). Under the TTUCB sampling rule, we have $\mathbb{E}_\mu[T_\beta^\varepsilon] < +\infty$.*

Theorem 2.3 is a direct consequence of Lemma F.1 and Lemma F.11.

## G Implementation details and additional experiments

The implementations details and supplementary experiments are detailed in Appendix G.1 and Appendix G.2.

### G.1 Implementation details

**Top Two sampling rules** Existing Top Two algorithms are based on a sampling procedure to choose between the leader $B_n$ and the challenger $C_n$, namely sample $B_n$ with probability $\beta$, else sample $C_n$. The difference lies in the choice of the leader and the challenger themselves.

TTTS [38] uses a TS (Thompson Sampling) leader and a RS (Re-Sampling) challenger based on a sampler $\Pi_n$. For Gaussian bandits, the sampler $\Pi_n$ is the posterior distribution $\bigtimes_{i \in [K]}\mathcal{N}(\mu_{n,i}, 1/N_{n,i})$ given the improper prior $\Pi_1 = (\mathcal{N}(0, +\infty))^K$. The TS leader is $B_n^{\mathrm{TS}} \in \arg\max_{i \in [K]}\theta_i$ where $\theta \sim \Pi_n$. The RS challenger samples vector of realizations $\theta \in \Pi_n$ until $B_n \notin \arg\max_{i \in [K]}\theta_i$, then it is defined as $C_n^{\mathrm{RS}} \arg\max_{i \in [K]}\theta_i$ for this specific vector of realization. When the posterior $\Pi_n$ and the leader $B_n$ have almost converged towards the Dirac distribution on $\mu$ and the best arm $i^\star(\mu)$ respectively, the event $B_n \notin \arg\max_{i \in [K]}\theta_i$ becomes very rare. The experiments in [21] reveals that computing the RS challenger can require more than millions of re-sampling steps. Therefore, the RS challenger can become computationally intractable even for Gaussian distribution where sampling from $\Pi_n$ can be done more efficiently.

T3C [39] combines the TS leader and the TC challenger. $\beta$-EB-TCI [21] combines the EB leader with the TCI challenger defined as

$$C_n^{\text{TCI}} = \underset{i \neq B_n}{\arg\min} \, \mathbb{1} \, (\mu_{n,B_n} > \mu_{n,i}) \, \frac{(\mu_{n,B_n} - \mu_{n,i})^2}{2(1/N_{n,B_n} + 1/N_{n,i})} + \log(N_{n,i}) \, .$$

Since the TC and TCI challenger can re-use computations from the stopping rule, those two challengers have no additional computational cost which makes it very attractive for larger sets of arms.

Each tracking procedure has a computational and memory cost in $\mathcal{O}(1)$, hence total cost of $\mathcal{O}(K)$ for the $K$ independent procedures. Each UCB index has a computational and memory cost in $\mathcal{O}(1)$, hence total cost of $\mathcal{O}(K)$ to compute $B_n^{\text{UCB}}$ as in (3). Each TC index (or stopping rule index) has a computational and memory cost in $\mathcal{O}(1)$, hence total cost of $\mathcal{O}(K)$ to compute $C_n^{\text{TC}}$ as in (4). Therefore, the per-round computational and memory cost of TTUCB is in $\mathcal{O}(K)$.

**Other sampling rules**  At each time $n$, Track-and-Stop (TaS) [17] computes the optimal allocation for the current empirical mean, $w_n = w^\star(\mu_n)$. Given $w_n \in \triangle_K$, it uses a tracking procedure to obtain an arm $I_n$ to sample. On top of this tracking a forced exploration is used to enforce convergence towards the optimal allocation for the true unknown parameters. The optimization problem defining $w^\star(\mu)$ can be rewritten as solving an equation $\psi_\mu(r) = 0$, where

$$\forall r \in (1/\min_{i \neq i^\star}(\mu_{i^\star} - \mu_i)^2, +\infty), \ \psi_\mu(r) = \sum_{i \neq i^\star} \frac{1}{\left(r(\mu_{i^\star} - \mu_i)^2 - 1\right)^2} - 1$$

The function $\psi_\mu$ is decreasing, and satisfies $\lim_{r \to +\infty} \psi_\mu(r) = -1$ and $\lim_{y \to 1/\min_{i \neq i^\star}(\mu_{i^\star} - \mu_i)^2} F_\mu(y) = +\infty$. For the practical implementation of the optimal allocation, we use the approach of [17] and perform binary searches to compute the unique solution of $\psi_\mu(r) = 0$. A faster implementation based on Newton's iterates was proposed by [6] after proving that $\psi_\mu$ is convex. While this improvement holds only for Gaussian distributions, the binary searches can be used for more general distributions.

DKM [12] view $T^\star(\mu)^{-1}$ as a min-max game between the learner and the nature, and design saddle-point algorithms to solve it sequentially. At each time $n$, a learner outputs an allocation $w_n$, which is used by the nature to compute the worst alternative mean parameter $\lambda_n$. Then, the learner is updated based on optimistic gains based on $\lambda_n$.

FWS [42] alternates between forced exploration and Frank-Wolfe (FW) updates.

LUCB [24] samples and stops based on upper/lower confidence indices for a bonus function $g$. For Gaussian distributions, it rewrites for all $i \in [K]$ as

$$U_{n,i} = \mu_{n,i} + \sqrt{\frac{2c(n-1,\delta)}{N_{n,i}}} \quad \text{and} \quad L_{n,i} = \mu_{n,i} - \sqrt{\frac{2c(n-1,\delta)}{N_{n,i}}} \, .$$

At each time $n$, it samples $\hat{i}_n$ and $\arg\max_{i \neq \hat{i}_n} U_{n,i}$ and stops when $L_{n,\hat{i}_n} \geq \max_{i \neq \hat{i}_n} U_{n,i}$. The $\beta$-LUCB algorithm samples $\hat{i}_n$ with probability $\beta$, else it samples $\arg\max_{i \neq \hat{i}_n} U_{n,i}$. The stopping time is the same as for LUCB.

**Adaptive proportions**  [44] propose IDS, whcih is an update mechanism for $\beta$ based on the optimality conditions for the problem underlying $T^\star(\mu)$. For Gaussian with known homoscedastic variance, IDS can be written as

$$\beta_n = \frac{N_{n,B_n} d_{\text{KL}}(\mu_{n,B_n}, u_n(B_n, C_n))}{N_{n,B_n} d_{\text{KL}}(\mu_{n,B_n}, u_n(B_n, C_n)) + N_{n,C_n} d_{\text{KL}}(\mu_{n,C_n}, u_n(B_n, C_n))} = \frac{N_{n,C_n}}{N_{n,B_n} + N_{n,C_n}} \, ,$$

where the second equality is obtained by direct computations which uses that

$$u_n(i,j) = \inf_{x \in \mathbb{R}} \, [N_{n,i} d_{\text{KL}}(\mu_{n,i}, x) + N_{n,j} d_{\text{KL}}(\mu_{n,j}, x)] = \frac{N_{n,i}\mu_{n,i} + N_{n,j}\mu_{n,j}}{N_{n,i} + N_{n,j}} \, .$$

**Reproducibility**  Our code is implemented in `Julia 1.7.2`, and the plots are generated with the `StatsPlots.jl` package. Other dependencies are listed in the `Readme.md`. The `Readme.md` file also provides detailed julia instructions to reproduce our experiments, as well as a `script.sh` to run them all at once. The general structure of the code (and some functions) is taken from the tidnabbil library.[1]

Table 4: CPU running time in seconds on random Gaussian instances ($K = 10$).

|         | TTUCB | EB-TCI | T3C  | TTTS | TaS   | FWS  | DKM  | LUCB | Uniform |
|---------|-------|--------|------|------|-------|------|------|------|---------|
| Average | 0.14  | 0.10   | 0.06 | 0.82 | 78.38 | 7.10 | 0.40 | 0.06 | 0.14    |
| Std     | 0.11  | 0.30   | 0.05 | 0.65 | 50.34 | 9.6  | 0.30 | 0.09 | 0.10    |

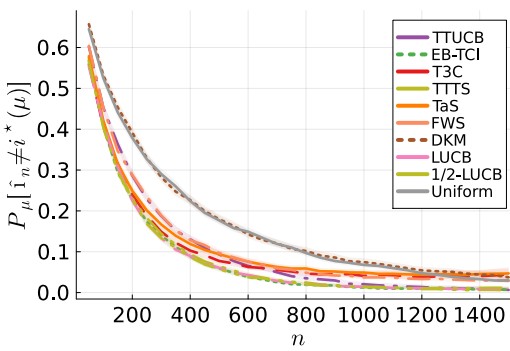

Figure 3: Empirical errors at time $n < \tau_\delta$ on random Gaussian instances ($K = 10$).

## G.2  Supplementary experiments

**Running time**  The CPU running time corresponding to the experiment displayed in Figure 1(a) are reported in Table 4. They match our discussion on computational cost detailed in Appendix G.1. The slowest algorithm is TaS, followed closely by FWS and TTTS. All remaining algorithms have similar computational cost: TTUCB, $\beta$-EB-TCI, T3C, LUCB and uniform sampling. It is slightly higher for DKM.

We emphasize that this is a coarse empirical comparison of the CPU running time in order to grasp the different orders of magnitude. More efficient implementation could (and should) be used by practitioners. As an example, the computational cost of TaS can be improved for Gaussian distributions by using the algorithm from [6] based on Newton's iterates. However, we doubt that the faster implementation of TaS will match the computational cost of DKM or FWS.

**Empirical errors before stopping**  At the exception of LUCB, all the considered algorithms are anytime algorithms (see [23] for a definition) since they are not using $\delta$ in their sampling rule. While all those algorithms are $\delta$-correct, none enjoy theoretical guarantees on the probability of error before stopping, i.e. upper bounds on $\mathbb{P}_\mu(\hat{\imath}_n \neq i^\star(\mu))$ for $n < \tau_\delta$. In Figure 3, we display their averaged empirical errors at time $n < \tau_\delta$ ( i.e. $\mathbb{1}(\hat{\imath}_n \neq i^\star)$) corresponding to the experiment displayed in Figure 1(a), with their associated Wilson Score Intervals [43]. To avoid an unfair comparison between algorithms having different stopping time, we restrict our plots to the median of the observed empirical stopping time. Therefore, even the fastest algorithm will average its empirical errors on at least 2500 instances.

Based on Figure 3, we see that uniform sampling and DKM perform the worst in terms of empirical error. At all times, the smallest empirical errors are achieved by $\beta$-EB-TCI, TTTS and LUCB. While TTUCB is at first as bad as FWS, its empirical error tends to match the one of the best algorithms for larger time. For TaS and T3C, the trend is reversed.

---

[1]This library was created by [12], see https://bitbucket.org/wmkoolen/tidnabbil. No license were available on the repository, but we obtained the authorization from the authors.

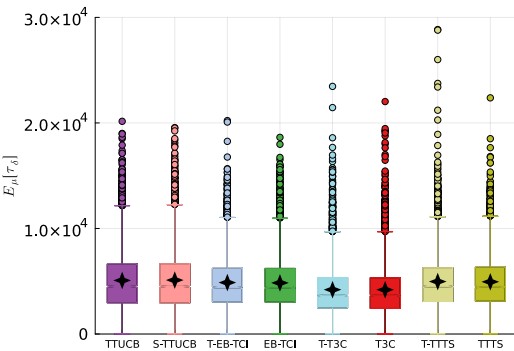

Figure 4: Empirical stopping time on random Gaussian instances ($K = 10$): tracking (T-) versus sampling (S-).

**Tracking versus sampling**    The TTUCB sampling rule uses tracking instead of sampling. Since both approaches aim at doing the same with either a deterministic or a randomized approach, it is interesting to assess whether they lead to different empirical performance. Therefore, we compare both approaches for four Top Two sampling rules. Figure 4 reveals that the algorithmic choice of tracking or sampling has a negligible impact on the empirical stopping time.

In light of this experiment, the choice of tracking or sampling is mostly a question of theoretical analysis. Since the analysis of a randomized sampling requires to control of the randomness of the allocation, we choose a deterministic tracking for analytical simplicity. Moreover, this choice is natural when both the leader and challenger are deterministic.

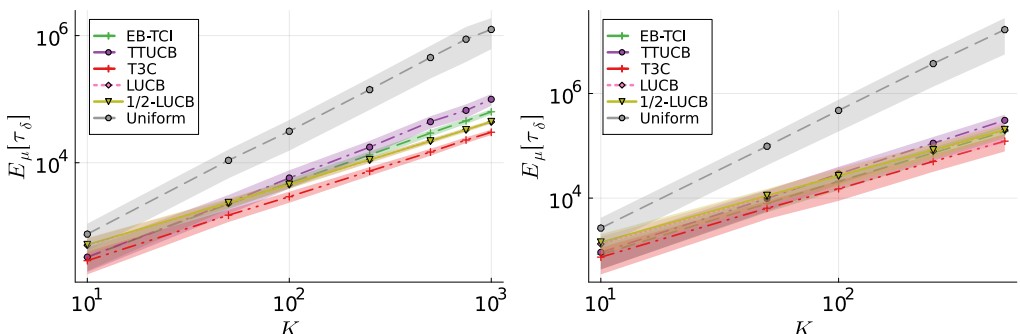

Figure 5: Influence of the dimension $K$ on the average empirical stopping time ($\pm$ standard deviation) for the Gaussian benchmark (a) "$\alpha = 0.3$" and (b) "$\alpha = 0.6$".

**Larger sets of arms**    In addition to the results presented in Section 4, we also evaluate the performance of our algorithm on the two other benchmarks used in [20]. The "$\alpha = 0.3$" scenarios consider $\mu_i = 1 - \left(\frac{i-1}{K-1}\right)^\alpha$ for all $i \in [K]$, with hardness $H(\mu) \approx 3K$. The "$\alpha = 0.6$" scenarios consider $\mu_i = 1 - \left(\frac{i-1}{K-1}\right)^\alpha$ for all $i \in [K]$, with hardness $H(\mu) \approx 12K^{1.2}$. The observations from Figure 5 are consistent with the ones in Figure 1(b). Overall, T3C performs the best for larger sets of arms.

### G.2.1    Adaptive proportions

The ratio $T^\star_{1/2}(\mu)/T^\star(\mu)$ seems to reach its highest value $r_K = 2K/(1 + \sqrt{K-1})^2$ for "equal means" instances (Lemma C.6), i.e. $\mu_i = \mu_{i^\star} - \Delta$ for all $i \neq i^\star$ with $\Delta > 0$. To best observe differences between Top Two algorithms with $\beta = 1/2$ and their adaptive version, we consider such instances with $K = 35$ ($r_K \approx 3/2$). Figure 6 reveals that adaptive proportions yield better empirical performance, with an empirical speed-up close to $r_K \approx 3/2$. We also compare them with three asymptotically optimal BAI algorithms (TaS, FWS and DKM). Even on those hard instances, Top

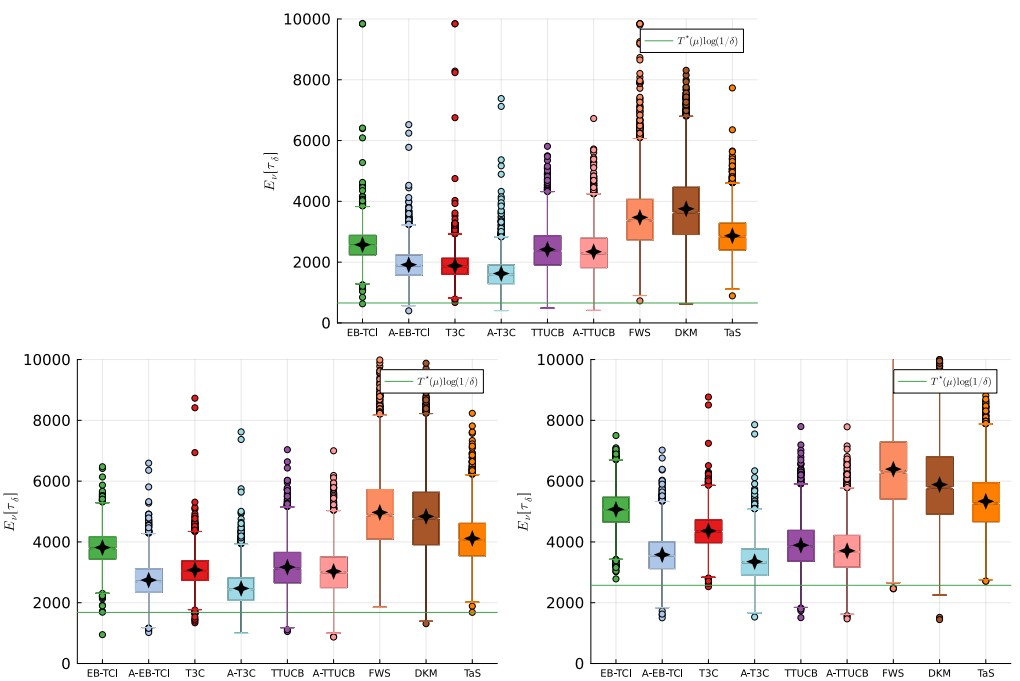

Figure 6: Empirical stopping time on "equal means" instances $(K, \mu_{i^\star}, \Delta) = (35, 0, 0.5)$ for (top) $\delta = 0.1$ and (bottom) $\delta \in \{0.01, 0.001\}$: constant $\beta = 1/2$ and adaptive (A-).

Two algorithms with fixed $\beta = 1/2$ are outperforming the asymptotically optimal algorithms for all the values $\delta \in \{0.1, 0.01, 0.001\}$. While being only $1/2$ asymptotically optimal, Top Two algorithms (with fixed $\beta = 1/2$) can obtain significantly better empirical performances compared to existing asymptotically optimal in the finite-confidence regime. The gap between the empirical performance of the adaptive and the fixed $\beta$ Top Two algorithms is increasing with $\delta$ decreasing. Interestingly, TTUCB appears to be slightly more robust to decreasing confidence $\delta$ compared to other Top Two algorithms.

We also compare Top Two algorithms using a fixed proportion $\beta = 1/2$ with their adaptive counterpart using $\beta_n = N_{n,C_n}/(N_{n,B_n} + N_{n,C_n})$ at time $n$ on random instances.

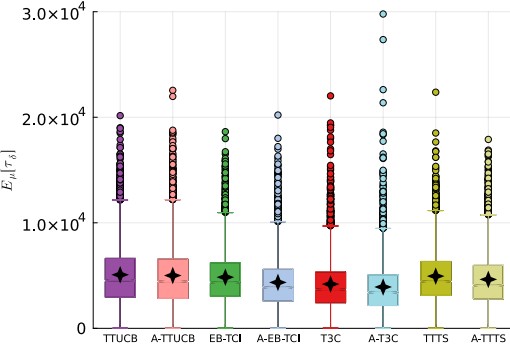

Figure 7: Empirical stopping time on random Gaussian instances ($K = 10$): constant $\beta = 1/2$ and adaptive (A-).

Experiments are conducted on 5000 random Gaussian instances with $K = 10$ such that $\mu_1 = 0.6$ and $\mu_i \sim \mathcal{U}([0.2, 0.5])$ for all $i \neq 1$. Figure 7 shows that their performances are highly similar, with a slim advantage for adaptive algorithms. For instances with multiple close competitors, the same phenomenon appears (see Figure 8 below). This is expected as $T^\star_{1/2}(\mu)/T^\star(\mu)$ is close to one when sub-optimal arms have significantly distinct means (see Figure 2).

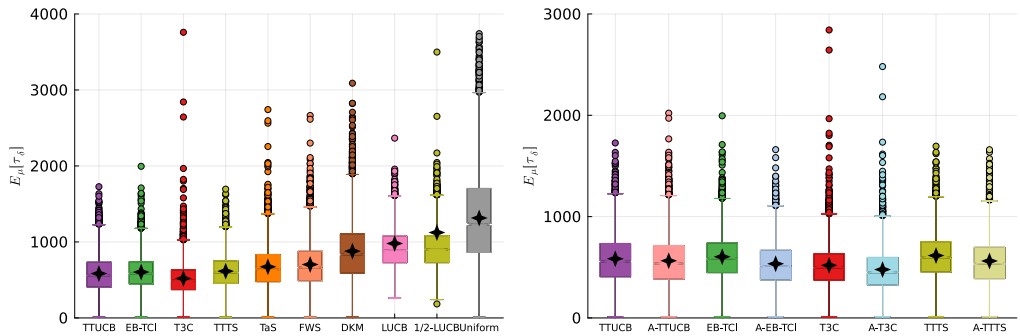

Figure 8: Empirical stopping time on random Gaussian instances ($K = 10$) with multiple close competitors.

Then, we assess the performance on 5000 random Gaussian instances with $K = 10$ such that $\mu_1 = 0.6$ and $\mu_i = \mu_1 - \Delta_i$ for $i \neq 1$ where $\Delta_i = \frac{1}{20}\left(\frac{995}{1000} + \frac{u_i}{100}\right)$ for all $i \in \{2, 3, 4, 5, 6\}$ and $\Delta_i = \frac{1}{10}\left(\frac{995}{1000} + \frac{u_i}{100}\right)$ for all $i \in \{7, 8, 9, 10\}$ with $u_i \sim U([0, 1])$. Numerically, we observe $w^\star(\mu)_{i^\star} \approx 0.28276 \pm 0.0003$ (mean $\pm$ std). Both plots in Figure 8 show the same phenomena than the ones observed in Figures 1(b) and 7.

