# OpenReview forum: "Non-Asymptotic Analysis of a UCB-based Top Two Algorithm"
_NeurIPS.cc/2023/Conference — NeurIPS 2023 spotlight_

### Official Review · Reviewer_YhuH · 2023-06-28

**Soundness:** 3 good
**Presentation:** 3 good
**Contribution:** 3 good
**Rating:** 5
**Confidence:** 4

**Summary:**

In this paper, the authors proposed a new UCB-based Top-Two algorithm (TTUCB) for the best-arm identification problem (BAI) and derived a non-asymptotic upper bound on the expected sample complexity holding for any instance having a unique best arm. The authors also demonstrated that the TTUCB algorithm achieves competitive empirical performance compared to other algorithms, including some existing Top-Two methods.

**Strengths:**

There have been some existing works that proposed top-two algorithms and analyzed the expected sample complexity for vanishing error level $\delta$, i.e., $\delta \to 0$ (asymptotic upper bound). However, this work proposed a new top-two algorithm (TTUCB) and provided a non-asymptotic upper bound on the sample complexity, i.e., a bound that holds for any error level $\delta$. The main (and new) contribution of this paper is to provide analytical techniques to obtain a non-asymptotic bound for the expected sample complexity for the TTUCB.

**Weaknesses:**

However, this paper contains quite many weaknesses as follows.

+  The tightness of this bound is not verified in this paper, especially for non-vanishing $\delta$.
+  The obtained non-asymptotic upper bound in Theorem 2.4. does not match the lower bound (Lemma 1.1) as $\delta \to 0$, which shows that the non-asymptotic upper bound is not tight enough at small $\delta$.  Note that Theorem 2.3 shows that TTUCB is $\beta$-optimal as $\delta \to 0$.
+  In the moderate regime $(\delta=0.1)$, the experiment (see Fig. 1) shows that the TTUCB algorithm can be better or worse than other existing ones. For example, in Fig. 1a, TTUCB looks worse than T3C although it is better than the others.  Hence, it seems to me that although other existing algorithms are developed for $\delta \to 0$, they can be better than your algorithm (i.e., TTUCB) as moderate $\delta$.


**Questions:**

+ Why don't you compare the performance of TTUCB with the non-asymptotic upper bound in Theorem 2.4 to check whether this bound is tight or not at different values of $\delta$?
+  From my second comment in the Originality section, what is the motivation for developing TTUCB for non-vanishing $\delta$? Why don't we perform non-asymptotic analysis for the existing top-two algorithms? Did you mean that you would like to develop a new top-two algorithm (more specifically, TTUCB) such that the non-asymptotic analysis for the upper bound can be performed easier than the others?
+ What is the main challenge when changing an asymptotic upper bound to a non-asymptotic one? More specifically, why is Theorem 2.3 optimal, but Theorem 2.4 not optimal as $\delta \to 0$?

**Limitations:**

This is a theoretical work. The authors already finished the checklist as well as stated the limitations of theorems and results.

---

> ### Author Rebuttal · Authors · 2023-08-07
>
> We thank the reviewer for the feedback and questions. We hope to address them in the following.
>
> We first answer the points raised in the “weaknesses” section.
>
> **Tightness of the finite-confidence upper bound of Theorem 2.4.**
> The lack of a tight non-asymptotic lower bound is the main open problem in fixed-confidence best-arm identification. Unfortunately, we don’t solve this open problem in this work. Given the lack of a tight lower bound, we did the best we could, which is to compare to many other existing upper bounds as well as existing lower bounds. Those comparisons are discussed extensively in Section 2.1 and Table 1.
>
> **Asymptotic behavior of the finite-confidence upper bound of Theorem 2.4.**
> We only managed to get within a factor 2 of the asymptotically optimal constant in the non-asymptotic bound. There is definitely room for improvement here, but our result is the first non-asymptotic bound for any top-two algorithm and we had to introduce a new proof technique to obtain it. That technique will surely be refined in the future. While Theorem 2.3 assumes that the means are distinct, Theorem 2.4 is the first guarantee on the expected sample complexity of a Top Two algorithm which doesn’t make this assumption.
>
> **Empirical performance of TTUCB compared to other Top Two algorithms.**
> The objective of this work was not to propose a new algorithm that beat every other Top-Two method. We would have been happy with a non-asymptotic bound for T3C for example. TTUCB is very similar to T3C (it only swaps a TS leader for a UCB leader) and it is expected to have similar empirical performance. However, by replacing a TS leader with a UCB leader and using tracking instead of sampling, we obtained a deterministic algorithm amenable to finite-confidence analysis. Deriving finite-confidence guarantees for randomized Top Two algorithms such as T3C is an interesting direction for future research.
>
> We now answer your other questions.
>
> **Motivation behind introducing TTUCB and analyzing it in the finite-confidence regime.**
> Our motivation is to analyze the family of Top-Two algorithms in general in a non-asymptotic regime. We would love to understand all of them, in particular the variants with the best empirical performance such as T3C. For some of them like EB-TC, experiments suggest that no good non-asymptotic bound is possible. Before our work, none of them had any non-asymptotic guarantees. We managed to prove a sample complexity upper bound for TTUCB. Since TTUCB is very similar to T3C (only the regret minimization algorithm used for the leader changes), we believe that our results are informative for other Top Two algorithms.
>
> **Conversion of an asymptotic upper bound towards a finite-confidence one.**
> We would like to emphasize that our main contribution is the analysis of Top Two algorithms in a new way, which is very different from the asymptotic analysis. Therefore, we are not converting the existing asymptotic analysis into a non-asymptotic one, instead we provide new tools to analyze this class of algorithm. Theorem 2.4 is asymptotically suboptimal compared to Theorem 2.3 by a multiplicative factor $2$ because it is a different proof technique and there is one technical difficulty that we could not overcome (see lines 263-265 in Section 3, and lines 629-632 in Appendix D.2).
> The main goal of existing asymptotic analyses is to exhibit a $\delta$-independent time $T_{\mu}$ after which the empirical proportions $N_{n}/(n-1)$ will stay $\epsilon$-close to the ($\beta$-)optimal allocation $w^{\star}(\mu)$. Intuitively, this property can only hold after a long time. If we were to express the dependencies of $T_{\mu}$ in terms of gaps and $K$, we would obtain highly suboptimal terms which are significantly larger than the ones obtained in Theorem 2.4. As $T_{\mu}/\log(1/\delta) \to 0$, an asymptotic analysis can disregard such terms since they are simply describing the complexity of a transitory regime. However, the transitory regime is precisely what matters for a finite-confidence upper bound. Inherently, the asymptotic analysis requires too much control on the empirical means and proportions to yield any meaningful information in the finite-confidence regime. As a consequence, we had to derive a new proof strategy to obtain an insightful finite-confidence upper bound.
>
> **Comparison between empirical performance and theoretical upper bound.**
> Comparisons between the empirical performance of the theoretical upper bound show that there is still room for improvement (as is the case for all the bounds of the algorithms we compare to). Our proof method gave a first non-asymptotic bound, but we expect that it could be refined in the future to catch finer details of the algorithm behavior. Our experiments on large sets of arms provide some insights on whether our finite-confidence upper bound is tight. Empirically, Figure 1(b) and Figure 5 in Appendix G.2 hints that the empirical performance of TTUCB scales linearly with $H(\mu)$ and not in $H(\mu)^{\alpha}$ as suggested by Theorem 2.4. Therefore, we are hopeful that better finite-time confidence guarantees can be shown.

---

> > ### Comment · Reviewer_YhuH · 2023-08-16
> > **Reply to authors' rebuttal**
> >
> > Thank you very much for your answering my questions.

---

### Official Review · Reviewer_nKF8 · 2023-07-07

**Soundness:** 3 good
**Presentation:** 4 excellent
**Contribution:** 3 good
**Rating:** 7
**Confidence:** 3

**Summary:**

This paper presents a non-asymptotic upper bound on the expected sample complexity of a TT sampling rule-based algorithm for any error level. Numerical results also verify the performance of TTUCB. This is nice work

**Strengths:**

1. The paper is well-written and organized. The insights behind the main results are well-presented.
2. The tracking rule is simple and easy to implement.
3. I appreciate that the authors include several simulations.

**Weaknesses:**

One weakness is that there is still room for further improvement in the bound.

**Questions:**

1. Any comments on how to choose the function $g_u(n)$? Are there any required properties?
2. Can you provide any suggestions on how to choose $\beta$ in practice?"

**Limitations:**

Yes

---

> ### Author Rebuttal · Authors · 2023-08-07
>
> We thank the reviewer for the feedback and questions. We hope to address them in the following.
>
> **Further improvement in the finite-confidence upper bound of Theorem 2.4.**
> Compared to our asymptotically $\beta$-optimal result in Theorem 2.3, we only managed to get within a factor $2$ of the $\beta$-characteristic time $T^\star_{\beta}(\mu)$ in our non-asymptotic bound (Theorem 2.4). There is definitely room for improvement here, but our result is the first non-asymptotic bound for any top-two algorithm and we had to introduce a new proof technique to obtain it. That technique will surely be refined in the future. While Theorem 2.3 assumes that the means are distinct, Theorem 2.4 is the first guarantee on the expected sample complexity of a Top Two algorithm which doesn’t make this assumption.
> As regards the finite-confidence regime, we provide a detailed discussion on each term (lines 186-200) and compare them with existing upper bounds and lower bounds (lines 201-229). While there is still room for further improvements in future work, Table 1 shows that we achieve the best known finite-confidence upper bounds among the class of asymptotically ($\beta$-)optimal algorithms. The lack of a tight non-asymptotic lower bound is one of the main open problems in fixed-confidence best-arm identification.
>
> **Choice of $g(n)$ in the bonus defining the UCB leader in (3).**
> The only constraint on $g(n)$ is that it should yield valid confidence intervals. Mathematically, we need to have a property similar to Lemmas E.2 and E.3 in Appendix E, for all $t \in [n^{1/\alpha}, n]$ and all arms $i \in [K]$, $\mu_{i} \in [\mu_{t,i} \pm \sqrt{g(t)/ N_{t,i}}]$. Therefore, the choice of $g(n)$ is constrained by the concentration results used to control the deviation of the empirical mean to its expected value. $g_{u}(n)$ was obtained with a union bound over time (Lemma E.2 in Appendix E), $g_{m}(n)$ relies on a mixture of martingales (Lemma E.3).
> Given two theoretically validated choices (e.g. $g_{u}(n)$ and $g_{m}(n)$), the lowest $g(n)$ will yield better empirical performance since larger $g(n)$ means more conservative UCB. That is the reason why we conduct our experiments with $g_{m}(n)$ instead of $g_{u}(n)$ since it is possible to show that $g_{m}(n) \le g_{u}(n)$ for $n \ge 50$ when choosing $s=\alpha =1.2$. Intuitively, $g(n)$ should be chosen small enough, but not too small. For example, in addition to not yielding valid confidence intervals, taking $g(n) = 0$ recovers the EB-TC algorithm which suffers from poor finite-time performances.
>
> **Choice of the fixed proportion $\beta$.**
> For fixed $\beta$, we highly recommend to use $\beta = 1/2$ without prior knowledge on the unknown mean parameters. This recommendation is supported theoretically with the worst-case inequality $T^\star_{1/2}(\mu) \le 2 T^\star(\mu)$, which shows that the expected sample complexity of an asymptotically $1/2$-optimal algorithm is at worst twice higher than that of any asymptotically optimal algorithm. In Appendix C, we show that the ratio $T^\star_{1/2}(\mu) / T^\star(\mu)$ is significantly smaller than $2$ for most instances. Based on Lemma C.6, we conjecture that the ratio reaches its maximum $r_{K} = 2K/(1+ \sqrt{K-1})^2$ for “equal means'' instances (all suboptimal arms having the same mean), i.e. $T^\star_{1/2}(\mu) \le r_{K} T^\star(\mu)$. This conjecture is validated empirically for $K=3$ and $K=4$ in Figure 2, and has a partial theoretical proof (see Lemma C.6).
> Ideally, we would choose $\beta$ adaptively (e.g. IDS [44]). While this would yield asymptotic optimality, the “price” lies in the fact that finite-confidence guarantees are harder to derive for these fully adaptive algorithms. Deriving them is an interesting direction for future research.

---

> > ### Comment · Reviewer_nKF8 · 2023-08-12
> >
> > I am satisfied with the authors' response. Thanks!

---

### Official Review · Reviewer_4xLs · 2023-07-07

**Soundness:** 4 excellent
**Presentation:** 4 excellent
**Contribution:** 4 excellent
**Rating:** 8
**Confidence:** 3

**Summary:**

The main objective of the paper is to establish the first non-asymptotic bound on the expected sample complexity for a Top two sampling policy in the context of best arm identification. To achieve this, the paper introduces a novel algorithm called TTUCB, which is proven to be $\beta$-optimal in asymptotic scenarios and also satisfies a non-asymptotic upper bound. This upper bound is compared with the existing upper bounds of established non-Top two algorithms. The paper supports its claims with synthetic simulation experiments.

One notable contribution of the paper is the development of a generic framework that enables regret minimization algorithms to effectively tackle the best arm identification problem while providing optimal guarantees. Additionally, the paper offers guidelines on how other regret minimization algorithms can be adapted to choose a leader at each round. Overall, the paper provides significant insights into the problem of best-arm identification and presents a promising algorithmic solution along with empirical evidence to support its efficacy.

**Strengths:**

The paper demonstrates exceptional writing, with clear claims and well-defined objectives. It successfully fills a significant gap in the literature by addressing the challenge of providing non-asymptotic guarantees for Top-two algorithms, resulting in original and valuable contributions to the broader bandit literature.

The problem formulation is thoroughly justified, establishing a solid foundation for the subsequent theoretical results. The discussions on the theoretical guarantees of the proposed TTUCB algorithm are insightful and comprehensive, effectively identifying areas for potential improvement. Additionally, the paper highlights the low computational complexity of the approach, enhancing its practical applicability.

The upper bounds achieved by the proposed algorithm are comparable to those of well-established methods in the field, as discussed aptly within the paper. By providing relevant comparisons, the paper strengthens the credibility and applicability of its results.

Furthermore, the paper successfully contextualizes its work within the broader scope of the bandit literature, illustrating the connections between the theoretical analysis and the use of regret-minimizing algorithms in the best arm identification problem. This contextualization is particularly valuable as it addresses a scenario that differs from the traditional exploitation-exploration trade-off associated with regret minimization.

Overall, the paper stands out for its clarity, strong theoretical discussions, well-justified approach, and relevant connections, making it a valuable contribution to the field of bandit problems.

**Weaknesses:**

One potential weakness is the tightness of the upper bound provided in the paper. The proposed upper bound incorporates various parameters and constants. While there is a comparison with the upper bounds of other algorithms, it remains uncertain whether this upper bound represents the best achievable result for TTUCB or Top-two algorithms in general. Further exploration into the tightness of the upper bound would contribute to a more comprehensive understanding of the algorithm's performance.

Another concern is the exclusive use of synthetic data in the experimental setup, without considering real-world datasets. It would be beneficial to address the potential challenges and limitations of applying the proposed approach to real-world datasets. Discussing any potential issues related to data collection, noise, or other factors specific to real-world settings would enhance the paper's practical applicability and broaden its impact.

**Questions:**

1) Could you provide more details on the challenges encountered within the proof technique that led to the decision of adopting a tracking approach instead of randomization in TTUCB? It would be helpful to understand the specific difficulties that arose and how they influenced this decision.

2) Additionally, has there been any simulation-based evidence to demonstrate the comparative performance of TTUCB using tracking versus a randomization-based approach? It would be valuable to assess the effectiveness of TTUCB in comparison to algorithms utilizing random

**Limitations:**

The authors talk about the limitations and places of potential improvement of the work. Societal impact is not discussed.

---

> ### Author Rebuttal · Authors · 2023-08-07
>
>
> We thank the reviewer for the feedback and questions. We hope to address them in the following.
>
> We first answer the points raised in the “weaknesses” section.
>
> **Tightness of the upper bound.**
> Our asymptotic bound is the best achievable for an algorithm that samples the leader with a fixed proportion $\beta$. Using the adaptive choice of $\beta$ named IDS [44] instead of fixed $\beta$ will yield an asymptotically optimal algorithm.
> In the finite-confidence regime. the $H(\mu)^\alpha$ dependency stems from the way we define our upper confidence indices. We doubt that better guarantees (with $\alpha = 1$) can be obtained for TTUCB. This is likely due to the shape of the optimistic index of the UCB part. Designing a TTUCB variant which does not suffer from this limitation is an interesting question for future work.
> About other Top Two algorithms, we can say that an algorithm that does not enforce extra exploration like EB-TC will not get good non-asymptotic bounds and has bad stopping time in some experiments. Top Two algorithms can add exploration in various ways: in the leader with a UCB method (ours) or using Thompson Sampling (T3C), or in the challenger like EB-TCI. Since T3C is the same as TTUCB but with UCB replaced by TS, we think based on the general bandit literature that the theoretical guarantees we could prove should be similar.
> We hope that the tools we are developing will allow us to investigate the non-asymptotic performance of these variants.
>
> **Synthetic vs real data.**
> Our results hold for sub-Gaussian random variables, hence in particular for bounded distributions, although they imply the strong instance-dependent property of $\beta$-optimality only for Gaussians.  Therefore, we could assess the performance of TTUCB on benchmarks that don’t use Gaussian rewards. However, since the main focus of our paper is a new theoretical analysis and since the performance guarantees we derive are more expressive for Gaussian distributions, we chose to use this in our experiments. We seek to understand Top Two algorithms in this admittedly restricted setting before generalizing to more real world settings.
>
> We now answer your other questions.
>
> **Tracking vs Randomization**
> It is easier to deal with the arm counts coming from a tracking procedure than with their random counterpart. It removes the need for martingales arguments to bound the deviations of the samples, hence it simplifies the analysis.
> Tracking also has a better convergence to the target compared to the randomization approach. Lemma 2.2 shows that the speed of convergence is at least $O(1/n)$ for tracking, while we would obtain a speed of $O(1/\sqrt{n})$ for randomization. In a related field, worse convergence rate for random approaches compared to tracking approaches has been observed empirically in section 7 of Mutný et al (2023, Active Exploration via Experiment Design in Markov Chains).
> It is not clear however whether this faster convergence is really useful, since the noise in the observations of the algorithm can be the dominant source of uncertainty.
> For all the experiments presented in Section 4 and Appendix G.2, we compared tracking vs sampling even though we didn’t include the plot. There is no significant difference for the stopping time.
> Overall, tracking or sampling makes no difference experimentally and no significant difference in the theoretical results we can obtain, and using tracking allows us to skip the step of relating the random counts to their mean.

---

> > ### Comment · Reviewer_4xLs · 2023-08-16
> >
> > Thank you for addressing my queries and providing an explanation for my better understanding. I am satisfied with the author's responses.

---

### Official Review · Reviewer_WPkC · 2023-07-25

**Soundness:** 3 good
**Presentation:** 4 excellent
**Contribution:** 3 good
**Rating:** 7
**Confidence:** 3

**Summary:**

The authors consider the problem of best-arm identification in the unstructured multi-armed bandit problem. More precisely, they study the problem in the fixed-budget setting in the moderate confidence regime. They propose a top two algorithm based on UCB which they call TTUCB and establish that the algorithm is asymptotically $\beta$-optimal. Here, by $\beta$-optimal, the authors mean that the algorithm samples sub-optimal arms optimally under the constraint that the best arm is sampled a proportion of times that is fixed at $\beta$. More importantly, they also provide a non-asymptotic guarantee for their algorithm that is valide for all confidence levels. The authors complement their theoretical findings with empirical results.


**Strengths:**

- Overall, the paper is well written and easy to follow. The related work is well discussed and the proofs appear to be sound to the best of my knowledge.
- The non-asymptotic study of best arm identification in the fixed confidence setting is a fascinating problem and remains largely unexplored. The authors provide a non-trivial contribution to this problem by shedding some  insight into the non-asymptotic behaviour of a UCB-based TOP TWO algorithm.
-  The proposed algorithm TTUCB seems to enjoy the best non-asymptotic guarantee up to universal constants universal constants.

**Weaknesses:**

- The theoretical results are limited to the case of Gaussian arms with known variances. Moreover, The authors only provide a algorithm that is asymptotically $\beta$-optimal but not asymptotically optimal. This is somewhat unsatisfactory!
- The authors address the non-asymptotic behaviour of best-arm identification but do not provide any non-asymptotic lower bound. As such it is unclear to me what is the best non-asymptotic bound. In fact, when the authors compare the different bounds, I believe they should also clarify (in the table) that TTUCB has an additional linear dependence on the number of arms $K$ at the finite confidence behaviour.
-  The authors could provide experiments which exhibit more the finite confidence behaviour of their algorithm in contrast with other algorithms. For instance, experiments that show how the sample complexity varies as $\delta$ varies.

**Questions:**

- In Lemma 3.1, the definition of $D_0$ is not given which, according to Appendix D.2, is equal to $(1+\varepsilon^2)/\beta^2$. My first comment is that this needs to be precised in the statement of Lemma 3.1. Secondly, if you were to adaptively seek to achieve the optimal allocation with $\beta^\star$, then $D_0$ is not anymore a universal constant and might depend in a non-trivial way on the problem instance $\mu$, in which case, your proof might not work to obtain a non-asymptotic guarantee. Would this be a limitation of your proof techniques when trying to attain the optimal constant $T_\star$? I am not sure using IDS might work either. Can you comment on this?
-  The result of Lemma 3.1, appear to be very specific to the Gaussian case, can the author comment on how can this generalize to the general case?
-  The authors mention that $ T_{1/2}^\star(\mu)\le  2T^\star(\mu)$ in the worst case, can you comment on how tight is this bound? Also according to your experiment it seems TTUCB outperform asymptotically optimal algorithms like track-and-stop and FWS. Does this mean that in the experiment you considered,  $T_{1/2}^\star(\mu)$ and $T^\star(\mu)$ are roughly the same? Can you run your experiments on a more challenging example where $T_{1/2}^\star(\mu)\approx 2 T^\star(\mu)$?
-  In the experiments section, the authors present plots in a scenario where $\delta=0.1$. Why is this value of $\delta$ considered to be in the moderate regime? Is there a clear definition if what we refer to as moderate regime? shouldn't this depend on  the number of arms $K$?

**Limitations:**

I believe the authors have discussed the limitations of their work accordingly.

---

> ### Author Rebuttal · Authors · 2023-08-07
>
> Thank you for your feedback and questions. We hope to address them in the following.
>
> **Only for Gaussian arms with known variances.**
> All our results hold for sub-Gaussian distributions as discussed in section 3.2. The algorithm can be said to be $\beta$-optimal in the distribution-dependent sense only for Gaussian distributions, but its sample complexity is bounded for all sub-Gaussians. We discuss distribution-dependent extension of the asymptotic analysis for single-parameter exponential families or bounded distributions in section 3.2.
>
> **Only asymptotically $\beta$-optimal.**
> As discussed in Section 3.4, achieving asymptotic optimality with TTUCB is possible by replacing the fixed $\beta$ by an adaptive choice (e.g. IDS [44]).
>
> **No non-asymptotic lower bound.**
> The lack of tight non-asymptotic lower bound is one of the main open problems in fixed confidence identification. We don’t solve it in this work.
>
> **Experiments on finite-confidence behavior and varying $\delta$.**
> Our choice of $\delta = 0.1$ was made to showcase the performance of the algorithm in the finite-confidence regime. We performed additional experiments for $\delta \in \{0.1, 0.01, 0.001\}$ on the “equal means'' instances with $(K, \mu^\star, \Delta) = (35,0,0.5)$, see the plots in the  “global” response. We compare three asymptotically optimal BAI algorithms (TaS, FWS and DKM) with three Top Two algorithms (EB-TCI, T3C and TTUCB) for fixed $\beta=1/2$ or adaptive proportions with IDS. Overall, most algorithms have the same scaling with the confidence $\delta$. The gap between the empirical performance of the adaptive and the fixed $\beta$ Top Two algorithms is increasing with $\delta$ decreasing. Interestingly, TTUCB appears to be slightly more robust to decreasing confidence $\delta$ compared to other Top Two algorithms.
>
> **Extension of Lemma 3.1 to adaptive choice of $\beta$, e.g. IDS.**
> When seeking to achieve asymptotic optimality with an adaptive choice of $\beta$ like IDS [44], it is harder to obtain a finite-confidence equivalent to Lemma 3.1. If we were to write the asymptotic analysis of TTUCB with IDS, we would obtain convergence towards the optimal allocation. While this result is stronger than Lemma 3.1, it only holds asymptotically. Obtaining a non-asymptotic result would require controlling the rate of convergence of the empirical allocation towards the optimal allocation. For fixed $\beta$, tracking yields a $\mathcal O(1/n)$ convergence of $N_{n,i^\star} / (n-1)$ towards $\beta$. Intuitively, having a fixed $\beta$ allows us to have an anchor and finite-confidence guarantees can then be obtained by reasoning only about the sub-optimal arms.
>
> **Extension of Lemma 3.1 beyond Gaussian distributions.**
> Our proof relies on the specificity of Gaussian distributions. In particular, we use the fact that the transportation costs can be written as the product of a function of the mean gap and a function of the arm allocations. This Gaussian-specific property allows us to decouple the concentration of the mean and the control of the empirical counts. Adapting our proof strategy to other distributions is an interesting direction for future work.
>
> **Tightness to the worst-case inequality for using $\beta = 1/2$.**
> In Appendix C, we provide a detailed discussion on the tightness of the inequality $T^\star_{1/2}(\mu) \le 2 T^\star(\mu)$ for Gaussian distributions. Lemma C.6 shows that the function $\mu \to T^\star_{1/2}(\mu) / T^\star(\mu)$ has gradient 0 when all the sub-optimal arms have the same mean (which we call “1-sparse” instances). At those points the ratio is $r_{K} = 2K/(1+ \sqrt{K-1})^2$. Therefore, we conjecture that $T^\star_{1/2}(\mu) \le r_{K} T^\star(\mu)$ for Gaussian distributions. In Figure 2 of Appendix C, this conjecture is numerically validated for $K=3$ and $K=4$.
> $r_{K}$ is greatly smaller than $2$ for a small number of arms $K$. For example $r_{5} \approx 1.1$, $r_{20}\approx 1.4$, $r_{100} \approx 1.7$.
> Figure 2 also suggests that the ratio $T^\star_{1/2}(\mu) / T^\star(\mu)$  is close to $1$ when all the means are different. On the random instances ($K=10$) considered in Figure 1(a), we observed empirically a small ratio $T^\star_{1/2}(\mu) / T^\star(\mu)$. This explains why TTUCB can perform on par with or outperform asymptotically optimal algorithms such as TaS and FWS. Figures 7 and 8 in Appendix G.2.1 shows that the benefit of choosing $\beta$ adaptively (i.e. IDS) is mild on those random instances.
> In Figure 6 in Appendix G.2.1, we show that using IDS yields better empirical performance than using a fixed $\beta$ for “equal means'' instances. Taking $K=35$ (i.e. $r_{K} \approx 3/2$), we empirically observe a similar ratio between the empirical stopping time of the Top Two algorithms. We performed additional experiments on this “equal means'' instance with $(K, \mu^\star, \Delta) = (35,0,0.5)$, see the plots in the  “global” response. We compare three asymptotically optimal BAI algorithms (TaS, FWS and DKM) with three Top Two algorithms (EB-TCI, T3C and TTUCB) with either fixed $\beta=1/2$ or adaptive proportions with IDS. Even on those hard instances, Top Two algorithms with fixed $\beta = 1/2$ are outperforming the asymptotically optimal algorithms for all the values $\delta \in \{0.1,0.01,0.001\}$. While being only $1/2$ asymptotically optimal, Top Two algorithms (with fixed $\beta=1/2$) can obtain significantly better empirical performances compared to existing asymptotically optimal in the finite-confidence regime.
>
> **The moderate regime definition.**
> The finite-confidence or moderate regime has no formal mathematical definition. Such a definition should indeed depend on $K$ and other problem-dependent quantities. It describes a regime where the $\delta$-independent term is not dominated by $T^\star \log(1/\delta)$.
> We took $\delta = 0.1$ to highlight the performance when $\delta$ was not too small.
>
> **Miscellaneous.**
> We will add the definition of $D_{0}$ in lemma 3.1.

---

> > ### Comment · Reviewer_WPkC · 2023-08-11
> > **Thanks for your response.**
> >
> > Thanks for addressing my questions and confirming my understanding of your results. I have now revised the score I gave from 6 to 7.
> >
> > Regarding the moderate regime definition, it would be nice to provide further discussion and intuition on this in your paper.  At an intuitive level, it seems to me that if $K$ is too large, than the limit that defines the moderate regime (whatever that is) becomes smaller. For the new  numerical results you provided, $K = 35$, which to my intuition means that $\delta = 0,001$ is still considered within the moderate confidence regime. This would explain why TTUCB is performing better than track-and-stop. It would be interesting to find a regime of $K$ and $\delta$ where track-and-stop outperforms your algorithm TTUCB, if it's at all possible. This would mean, at least experimentally, that an algorithm that is both asymptotically optimal and enjoys a non-trivial non-asymptotic guarantess is still an open question. I would be happy to hear your thoughts on this.

---

> > > ### Author Response · Authors · 2023-08-13
> > >
> > > When $K$ becomes large, the moderate regime indeed extends to smaller $\delta$. We will add further discussion of what that regime means in the paper. Thank you for the suggestion.
> > > Our non-asymptotic guarantees only hold for TTUCB with fixed $\beta$, which is asymptotically only $\beta$-optimal: if we perform experiments with small $K$ and $\delta$ very small, Track-and-Stop outperforms that algorithm in instances where $\beta=1/2$ is not optimal. TTUCB paired with an adaptive leader/challenger allocation however looks never worse than Track-and-Stop in our experiments. But we don’t have non-asymptotic results for that. An algorithm that is both asymptotically optimal and enjoys a non-trivial non-asymptotic guarantee is hence still an open question, but Top Two algorithms with adaptive $\beta$ are good candidates.

---

### Author Rebuttal · Authors · 2023-08-07

We attach additional experiments to better answer the questions of the reviewer WPkC.

---

### Decision · Program_Chairs · 2023-09-21

**Decision:**

Accept (spotlight)

**Comment:**

This paper has two novelties: (i) novel algorithm and (ii) novel analysis (finite-time analysis of a top-two-style algorithm for the first time). While it is not clear if the analysis is tight, these are meaningful contributions and worth being known in the NeurIPS community.